# Credal Bayesian Deep Learning

**Michele Caprio** *michele.caprio@manchester.ac.uk*
*Department of Computer Science,*
*University of Manchester*

**Souradeep Dutta** *souradeep@ece.ubc.ca*
*Department of Computer Science,*
*University of British Columbia*

**Kuk Jin Jang** *jangkj@seas.upenn.edu*
*Department of Computer and Information Science,*
*University of Pennsylvania*

**Vivian Lin** *vilin@seas.upenn.edu*
*Department of Computer and Information Science,*
*University of Pennsylvania*

**Radoslav Ivanov** *ivanor@rpi.edu*
*Department of Computer Science,*
*Rensselaer Polytechnic Institute*

**Oleg Sokolsky** *sokolsky@seas.upenn.edu*
*Department of Computer and Information Science,*
*University of Pennsylvania*

**Insup Lee** *lee@seas.upenn.edu*
*Department of Computer and Information Science,*
*University of Pennsylvania*

**Reviewed on OpenReview:** *https://openreview.net/forum?id=4NHF9AC5ui*

## Abstract

Uncertainty quantification and robustness to distribution shifts are important goals in machine learning and artificial intelligence. Although Bayesian Neural Networks (BNNs) allow for uncertainty in the predictions to be assessed, different sources of predictive uncertainty cannot be distinguished properly. We present Credal Bayesian Deep Learning (CBDL). Heuristically, CBDL allows to train an (uncountably) infinite ensemble of BNNs, using only finitely many elements. This is possible thanks to prior and likelihood finitely generated credal sets (FGCSs), a concept from the imprecise probability literature. Intuitively, convex combinations of a finite collection of prior-likelihood pairs are able to represent infinitely many such pairs. After training, CBDL outputs a set of posteriors on the parameters of the neural network. At inference time, such posterior set is used to derive a set of predictive distributions that is in turn utilized to distinguish between (predictive) aleatoric and epistemic uncertainties, and to quantify them. The predictive set also produces either (i) a collection of outputs enjoying desirable probabilistic guarantees, or (ii) the single output that is deemed the best, that is, the one having the highest predictive lower probability – another imprecise-probabilistic concept. CBDL is more robust than single BNNs to prior and likelihood misspecification, and to distribution shift. We show that CBDL is better at quantifying and disentangling different types of (predictive) uncertainties than single BNNs and ensemble of BNNs. In addition, we apply CBDL to two case studies to demonstrate its downstream tasks capabilities: one, for motion prediction in autonomous driving scenarios,

and two, to model blood glucose and insulin dynamics for artificial pancreas control. We show that CBDL performs better when compared to an ensemble of BNNs baseline.

# 1 Introduction

One of the greatest virtues an individual can have is arguably being aware of their own ignorance, and acting cautiously as a consequence. Similarly, an autonomous system using neural networks (NNs) would greatly benefit from understanding the probabilistic properties of the NN's output (for example, its robustness to distribution shift), in order to incorporate them into any further decision-making. This paper collocates on the path of giving a machine such a desirable quality.

In the last few years, there has been a proliferation of work on calibrating (classification) NNs, in order to estimate the confidence in their outputs (Guo et al., 2017) or to produce conformal sets that are guaranteed to contain the true label, in a probably approximately correct (PAC) sense (Park et al., 2020). While such methods are a promising first step, they require a calibration set (in addition to the original training set) and cannot be directly used on Out-Of-Distribution data without further examples.

Bayesian Neural Networks (BNNs) offer one approach to overcome the above limitations. The Bayesian paradigm provides a rigorous framework to analyze and train uncertainty-aware neural networks, and more generally to support the development of learning algorithms (Jospin et al., 2022). In addition, it overcomes some of the drawbacks of deep learning models, namely that they are prone to overfitting, which adversely affects their generalization capabilities, and that they tend to be overconfident about their predictions when they provide a confidence interval. BNNs, though, are trained using a single prior, which may still suffer from miscalibration and robustness issues (Lenk & Orme, 2009).

In this work we introduce Credal Bayesian Deep Learning (CBDL), a procedure that draws on concepts from the imprecise probability (IP) literature (Augustin et al., 2014; Troffaes & de Cooman, 2014; Walley, 1991). Unlike other techniques in the fields of artificial intelligence (AI) and machine learning (ML) involving imprecise probabilities – that typically only focus on classification problems – CBDL can be used for both classification and regression. It captures the ambiguity the designer faces when selecting which prior to choose for the parameters of a neural network, and which likelihood distribution to choose for the training data at hand.

CBDL can be thought of as a NN trained using prior and likelihood finitely generated credal sets (FGCSs), $\mathcal{P}_{\text{prior}}$ and $\mathcal{P}_{\text{lik}}$, respectively.[1] They are convex sets of probability measures having finitely many extreme elements (the elements that cannot be written as a convex combination of one another), $\text{ex}\mathcal{P}_{\text{prior}}$ and $\text{ex}\mathcal{P}_{\text{lik}}$, respectively (see also Remark 2). A very simple example of a prior FGCS $\mathcal{P}_{\text{prior}}$ is the collection of all the convex combinations of two one-dimensional Normal distributions $\mathcal{N}(\mu_1, \sigma_1^2)$ and $\mathcal{N}(\mu_2, \sigma_2^2)$, i.e.,

$$\mathcal{P}_{\text{prior}} = \{P : P = \beta\mathcal{N}(\mu_1, \sigma_1^2) + (1-\beta)\mathcal{N}(\mu_2, \sigma_2^2), \text{ for all } \beta \in [0,1]\}.$$

Consequently, in this toy example we have that $\text{ex}\mathcal{P}_{\text{prior}} = \{\mathcal{N}(\mu_1, \sigma_1^2), \mathcal{N}(\mu_2, \sigma_2^2)\}$. FGCSs are further examined in section 2.2.

Given the use of finitely generated credal sets, CBDL can also be seen as a non-condensed (uncountably) infinite ensemble of BNNs – each BNN corresponding to a pair $(P, L)$ of prior $P$ from the prior FGCS $\mathcal{P}_{\text{prior}}$ and likelihood $L$ from the likelihood FGCS $\mathcal{P}_{\text{lik}}$. Such infinite ensemble, though, is carried out using only finitely many elements, that is, only pairs $(P^{\text{ex}}, L^{\text{ex}})$ of components of the sets of extreme elements $\text{ex}\mathcal{P}_{\text{prior}}$ and $\text{ex}\mathcal{P}_{\text{lik}}$. This because every element $P$ of $\mathcal{P}_{\text{prior}}$ can be obtained from a convex combinations of the elements of $\text{ex}\mathcal{P}_{\text{prior}}$, and similarly for the likelihood FGCS. After training, CBDL produces a posterior FGCS $\mathcal{P}_{\text{post}}$ on the parameters of the neural network. At inference time, $\mathcal{P}_{\text{post}}$ is used to derive a predictive FGCS $\mathcal{P}_{\text{pred}}$, that is, given a new input, a set of plausible distributions over the space of outputs.[2] In turn,

---

[1]Intuitively, the larger these credal sets, the higher prior and likelihood ambiguity the user faces.

[2]As we shall see in section 3.1, since computing the posteriors and the predictive distributions is oftentimes an intractable problem, we approximate the elements of $\mathcal{P}_{\text{post}}$ and of $\mathcal{P}_{\text{pred}}$ using Variational Inference (VI). As a consequence, we denote the VI-approximated posterior and predictive credal sets as $\check{\mathcal{P}}_{\text{post}}$ and $\hat{\mathcal{P}}_{\text{pred}}$, respectively.

$\mathcal{P}_{\text{pred}}$ generates a set of outputs – or a single output, depending on the user's needs – that enjoys desirable probabilistic guarantees, even when the elements of credal set $\mathcal{P}_{\text{pred}}$ are approximated, e.g. using variational inference. CBDL also gives a way of quantifying and disentangling different types of uncertainties within $\mathcal{P}_{\text{pred}}$.

We use a credal set approach to overcome some of the drawbacks of single BNNs. In particular, CBDL allows to counter the criticism to the practice in (standard) Bayesian statistics of (i) using a single, arbitrary prior to represent the initial state of ignorance of the agent, (ii) using non-informative priors to model ignorance, and (iii) using a single, arbitrary likelihood to represent the agent's knowledge about the sampling model.[3] As a consequence, credal sets make the analysis more robust to prior and likelihood misspecification. In addition, they make it possible to quantify and distinguish between epistemic and aleatoric uncertainties (EU and AU, respectively). This is desirable in light of several areas of recent ML research, such as Bayesian deep learning (Depeweg et al., 2018; Kendall & Gal, 2017), adversarial example detection (Smith & Gal, 2018), and data augmentation in Bayesian classification (Kapoor et al., 2022).

AU refers to the uncertainty that is inherent to the data generating process; as such, it is irreducible. Think, for example, of a coin toss. No matter how many times the coin is tossed, the stochastic variability of the experiment cannot be eliminated. EU, instead, refers to the lack of knowledge about the data generating process; as such, it is reducible. It can be lessened on the basis of additional data. For example, after only a few tosses, we are unable to gauge whether a coin is biased or not, but if we repeat the experiment long enough, this type of uncertainty vanishes. We note in passing that predictive EU cannot be adequately captured using a single BNN (Hüllermeier & Waegeman, 2021; Fellaji & Pennerath, 2024). One reason for this is that selecting a unique prior and a unique likelihood implicitly assumes perfect knowledge around the true prior and the true data generating process. Another, more subtle one, is explored in the last paragraph of section 2.1. Methods that disentangle between the two types of (predictive) uncertainties that are based on a single distribution are ad-hoc, but are not theoretically well-justified. EU can be typically reduced by retraining the model using an augmented training set (Lin et al., 2024) (e.g. via semantic preserving transformations (Kaur et al., 2023), Puzzle Mix (Kim et al., 2020), etc.). On the other hand, since AU is irreducible, there is an increasing need for ML techniques that are able to detect and flag its excess, so that the user can "proceed with caution".

**Remark 1.** *EU should not be confused with the concept of epistemic probability (de Finetti, 1974; 1975; Walley, 1991). In the subjective probability literature, epistemic probability can be captured by a single distribution. Its best definition can be found in Walley (1991, Sections 1.3.2 and 2.11.2). There, the author specifies how epistemic probabilities model logical or psychological degrees of partial belief of the agent. We remark, though, how de Finetti and Walley work with finitely additive probabilities, while in this paper we use countably additive probabilities.*

The motivation for working with credal sets is threefold: (i) it allows to be robust against prior and likelihood misspecification; (ii) unlike when using a single probability distribution, it permits to represent ignorance in the sense of lack of knowledge; (iii) it allows to quantify and disentangle between EU and AU. A more in-depth discussion can be found in Appendix A. In addition, we point out that despite a hierarchical Bayesian model (HBM) approach seems to be a viable alternative to one based on credal sets, these latter are better justified philosophically, and do not suffer from the same theoretical shortcomings of HBM procedures (Bernardo, 1979; Hüllermeier & Waegeman, 2021; Jeffreys, 1946; Walley, 1991). A more detailed explanation can be found in Appendix B. Let us mention that we only compare CBDL against Bayesian techniques because a comparison against non-Bayesian ones is either difficult to justify, or unfair. Indeed, the initial knowledge component, captured by the prior distribution (or the set of priors, in our case), has no direct comparisons with other approaches. Take conformal prediction (Vovk et al., 2022; Shafer & Vovk, 2008; Gibbs et al., 2023; Barber et al., 2023) for instance. It is a model-free method, which means that no prior knowledge around the experiment at hand is required. The only choice the user makes is which non-conformity score to use. How would we compare this approach with the choice of different priors, which conveys the idea that the user knows that a true distribution over the space of neural network parameters exists, but it is not fully known? In addition, despite the ubiquitous claim that conformal prediction (CP) is an uncertainty *quantification* tool, it is actually not. CP is an uncertainty *representation* tool. Indeed, CP *represents* uncertainty via

---

[3]Criticisms (i) and (iii) are also pointed out in Manchingal & Cuzzolin (2022, Section 2.2).

the conformal prediction region (on the other hand, in this paper we represent uncertainty via credal sets). It does not quantify it: there is no real value attached to any kind of predictive uncertainty (aleatoric or epistemic). Some claim that the diameter of the conformal prediction region quantifies the uncertainty, but even in that case, it is unable to distinguish between AU and EU. Indeed, the diameter is a positive function of both: it increases as both increase, and hence it cannot be used to distinguish between the two.

We summarize our contributions next: (1) We present CBDL, and develop the theoretical tools and the algorithm required to use it in practice. (2) We show that a CBDL approach is more robust than single BNNs to prior and likelihood misspecification, and to distribution shifts. We also explain how, during inference, the credal set of posteriors $\mathcal{P}_{\text{post}}$ on the network parameters obtained during training is used to derive a credal set of predictive distributions $\mathcal{P}_{\text{pred}}$, and in turn a set of outcomes – or a single outcome – that enjoys probabilistic guarantees.[4] (3) We show how CBDL is better at quantifying and disentangling predictive AU and EU than single BNNs and ensemble of BNNs.[5] We also apply CBDL to two safety critical systems to demonstrate its downstream tasks capabilities. One, motion prediction for autonomous driving, and two, the human insulin and blood glucose dynamics for artificial pancreas control. We demonstrate improvements in both these settings with respect to ensemble of BNNs methods, the reason being that better uncertainty quantification leads to a positive impact on the decisions made using them.

Before moving on, let us point out that while CBDL pays a computational price coming from the use of credal sets, it is able to quantify both (predictive) EU and AU, and to do so in a principled manner, unlike single and ensembles of BNNs. In addition, it requires less stringent assumptions on the nature of the prior and likelihood ambiguity faced by the agent than other imprecise-probabilities-based techniques, as explained in Appendices B and L. We also stress that if the user prioritizes computational efficiency, they should use non-Bayesian methods, as they are faster to implement than Bayesian techniques. In safety-critical situations instead – where using uncertainty-informed methods is crucial for quantifying the types of uncertainties, but also for the outcome of the analysis at hand – CBDL is a natural choice.

**Structure of the Paper.** Section 2 presents the needed preliminary concepts, followed by section 3 that introduces and discusses the CBDL algorithm, together with its theoretical properties. We present our experimental results in section 4, and we examine the related work in section 5. Section 6 concludes our work. In the appendices, we give further theoretical and philosophical arguments and we prove our claims.

**Notation Summary.** In this paper, we make extensive use of different notation and acronyms. We summarize them in Table 1 to help the reader navigate through them.

## 2 Background and Preliminaries

In this section, we present the background notions that are needed to understand our main results. In section 2.1 we introduce Bayesian Neural Networks. Section 2.2 discusses (finitely generated) credal sets, upper and lower probabilities, and imprecise highest density regions. Section 2.3 introduces the concepts of upper and lower entropy, which are used to quantify and disentangle AU and EU. The reader familiar with these concepts can skip to section 3.

### 2.1 Bayesian Neural Networks

In line with the recent survey on BNNs by Jospin et al. (2022), Bayes' theorem can be stated as $P(H \mid D) = [P(D \mid H)P(H)]/P(D) = P(D, H)/\int P(D, H')\mathrm{d}H'$, where $H$ is a hypothesis about which the agent holds some prior beliefs, and $D$ is the data the agent uses to update their initial opinion. Probability distribution $P(D \mid H)$ represents how likely it is to observe data $D$ if hypothesis $H$ were to be true, and is called *likelihood*, while probability distribution $P(H)$ represents the agent's initial opinion around the plausibility

---

[4]Once again, as we shall see in section 3.1, we approximate the elements of the posterior and the predictive credal sets using Variational Inference (VI).

[5]We note in passing how recently Mucsányi et al. (2024) show that (finite) deep ensembles are the current state-of-the-art methods for quantifying and disentangling different types of uncertainties on ImageNet. Our experiments show that CBDL improves on (finite) ensembles of BNNs, both in uncertainty quantification and disentanglement, and in downstream tasks performances. Together with the findings in Mucsányi et al. (2024), this is an additional argument in favor of the effectiveness of our methodology.

| **Notation and Acronyms** | **Meaning** |
|---|---|
| FGCS | Finitely Generated Credal Set |
| CBDL | Credal Bayesian Deep Learning |
| Upper case latin letters, such as $P$ and $L$ | Probability measures, e.g. prior or likelihood |
| Lower case latin letters, such as $p$ and $\ell$ | The pdf/pmf associated to probability measures |
| $\Pi$ | A finite collection of probability measures |
| $\text{Conv}(\cdot)$ | The convex hull operator |
| $\Pi'$ | The convex hull of $\Pi$, i.e. $\Pi' = \text{Conv}(\Pi)$ |
| $\mathcal{P}_{\text{prior}}$ | A prior FGCS |
| $\mathcal{P}_{\text{lik}}$ | A likelihood FGCS |
| $\mathcal{P}_{\text{post}}$ | A posterior FGCS |
| VI | Variational Inference |
| $\check{\mathcal{P}}_{\text{post}}$ | A posterior FGCS whose extreme elements are the VI approximations of the "correct" ones |
| $\mathcal{P}_{\text{pred}}$ | A predictive FGCS |
| $\hat{\mathcal{P}}_{\text{pred}}$ | A predictive FGCS whose extreme elements are the VI approximations of the "correct" ones |
| $\text{ex}\mathcal{P}$ | The extreme elements of an FGCS $\mathcal{P}$ |
| BNN | Bayesian Neural Network |
| $D = D_{\mathbf{x}} \times D_{\mathbf{y}}$ | Training set; $D_{\mathbf{x}}$ is the set of training inputs, and $D_{\mathbf{y}}$ is the set of training outputs |
| EBNN | Ensemble of Bayesian Neural Networks |
| $x \mapsto \Phi(x)$ | Function representing the neural network architecture |
| $\underline{P}, \overline{P}$ | Lower and upper probabilities, respectively |
| HDR | Highest Density Region |
| IHDR | Imprecise Highest Density Region |
| $IR_\alpha(\mathcal{P})$ | Imprecise Highest Density Region of level $\alpha$ associated with FGCS $\mathcal{P}$ |
| $H(P)$ | Entropy of probability measure $P$ |
| $\underline{H}(P), \overline{H}(P)$ | Upper and lower entropy associated with a credal set |
| $\text{EU}(\mathcal{P})$ | Epistemic Uncertainty encoded in an FGCS $\mathcal{P}$ |
| $\text{AU}(\mathcal{P})$ | Aleatoric Uncertainty encoded in an FGCS $\mathcal{P}$ |
| $\text{TU}(\mathcal{P})$ | Total Uncertainty encoded in an FGCS $\mathcal{P}$ |
| $\#A$ | Cardinality of a generic set $A$ |
| post, pred | The act of computing a posterior and a posterior predictive distributions, respectively |
| BMA | Bayesian Model Averaging |

Table 1: Notation and acronyms that we use throughout the paper, together with their meaning.

of hypothesis $H$, and is called *prior*. The *evidence* available is encoded in $P(D) = \int P(D, H')\mathrm{d}H'$, while *posterior* probability $P(H \mid D)$ represents the agent's updated opinion. Using Bayes' theorem to train a predictor can be understood as learning from data $D$: the Bayesian paradigm offers an established way of quantifying uncertainty in deep learning models.

BNNs are stochastic artificial neural networks (ANNs) trained using a Bayesian approach (Caprio & Mukherjee, 2023b; Goan & Fookes, 2020; Jospin et al., 2022; Lampinen & Vehtari, 2001; Titterington, 2004; Wang & Yeung, 2021). The goal of ANNs is to represent an arbitrary function $y = \Phi(x)$. Let $\theta$ represent the parameters of the network, and call $\Theta$ the space $\theta$ belongs to. Stochastic neural networks are a type of ANN built by introducing stochastic components to the network. This is achieved by giving the network either a stochastic activation or stochastic weights to simulate multiple possible models with their associated probability distribution. This can be summarized as $\theta \sim p(\theta)$, $y = \Phi_\theta(x) + \varepsilon$, where $\Phi$ depends on $\theta$ to highlight the stochastic nature of the neural network, $p$ is the density of a probability measure $P$ on $\Theta$,[6] and $\varepsilon$ represents random noise to account for the fact that function $\Phi_\theta$ is just an approximation. The connection between the BNN notation and the general one of the Bayes' theorem is explained in the next two paragraphs.

To design a BNN, the first step is to choose a deep neural network *architecture*, that is, functional model $\Phi_\theta$. Then, the agent specifies the *stochastic model*, that is, a prior distribution over the possible model parametrization $p(\theta)$, and a prior confidence in the predictive power of the model $p(y \mid x, \theta)$. Given the usual assumption that multiple data points from the training set are independent, the product $\prod_{(x,y) \in D} p(y \mid x, \theta)$ represents the *likelihood* of outputs $y \in D_{\mathbf{y}}$ given inputs $x \in D_{\mathbf{x}}$ and parameter $\theta$, where (a) $D = D_{\mathbf{x}} \times D_{\mathbf{y}}$ is the training set; (b) $D_{\mathbf{x}} = \{x_i\}_{i=1}^n$ is the collection of training inputs, which is a subset of the space $\mathcal{X}$ of inputs; (c) $D_{\mathbf{y}} = \{y_i\}_{i=1}^n$ is the collection of training outputs, which is a subset of the space $\mathcal{Y}$ of outputs.

The model parametrization can be considered to be hypothesis $H$. Following Jospin et al. (2022), we assume independence between model parameters $\theta$ and training inputs $D_{\mathbf{x}}$, in formulas $D_{\mathbf{x}} \perp\!\!\!\perp \theta$. Hence, Bayes' theorem can be rewritten as

$$p(\theta \mid D) = \frac{p(D_{\mathbf{y}} \mid D_{\mathbf{x}}, \theta)p(\theta)}{\int_\Theta p(D_{\mathbf{y}} \mid D_{\mathbf{x}}, \theta')p(\theta')\mathrm{d}\theta'} \propto p(D_{\mathbf{y}} \mid D_{\mathbf{x}}, \theta)p(\theta).$$

Notice that the equality comes from having assumed $D_{\mathbf{x}} \perp\!\!\!\perp \theta$. Posterior density $p(\theta \mid D)$ is typically high dimensional and highly nonconvex (Izmailov et al., 2021c; Jospin et al., 2022), so computing it and sampling from it is a difficult task. The first issue is tackled using Variational Inference (VI) procedures, while Markov Chain Monte Carlo (MCMC) methods address the second challenge. Both are reviewed – in the context of machine learning – in Jospin et al. (2022, Section V), where the authors also inspect their limitations. BNNs can be used for both regression and classification (Jospin et al., 2022, Section II); besides having a solid theoretical justification, there are practical benefits from using BNNs, as presented in Jospin et al. (2022, Section III).

Suppose now we collect a new input $\tilde{x}$ of interest, and that we want to predict its correct output $\tilde{y}$. Then, the posterior distribution $p(\theta \mid D)$ over the network parameters comes to the rescue, as it allows us to compute the so-called *posterior predictive distribution*,

$$p(\tilde{y} \mid \tilde{x}, D) = \int_\Theta p(\tilde{y} \mid \tilde{x}, \theta)p(\theta \mid D)\mathrm{d}\theta = \mathbb{E}_{\theta \sim p(\theta \mid D)}[p(\tilde{y} \mid \tilde{x}, \theta)],$$

where $p(\tilde{y} \mid \tilde{x}, \theta)$ is the model distribution we specified before. Posterior predictive $p(\tilde{y} \mid \tilde{x}, D)$ tells us "how likely" output $\tilde{y}$ is to be the "correct one" for input $\tilde{x}$, given the knowledge encapsulated in the data $D$ we collected, which enters the computation via the posterior probability $p(\theta \mid D)$. Let us add a remark here: oftentimes, scholars claim that, in a Bayesian setting, the distribution on the parameters $\theta$ captures the epistemic uncertainty (EU) faced by the agent. This is a somehow agreeable premise, akin to a second-order distribution reasoning. If we accept this assertion, though, we see how EU at the predictive level is not

---

[6]We can write $p$ as the Radon-Nikodym derivative of $P$ with respect to some $\sigma$-finite dominating measure $\mu$, that is, $p = \mathrm{d}P/\mathrm{d}\mu$.

quantifiable any more, since it gets washed away by taking the expectation $\mathbb{E}_{\theta \sim p(\theta|D)}[\cdot]$. As we shall see later, CBDL allows to overcome this conceptual shortcoming.

## 2.2 Imprecise Probabilities

As CBDL is rooted in the theory of imprecise probabilities (IPs), in this section we give a gentle introduction to the IP concepts we will use throughout the paper.

CBDL is based on the *Bayesian sensitivity analysis* (BSA) approach to IPs, that in turn is grounded in the *dogma of ideal precision* (DIP) (Berger, 1984), (Walley, 1991, Section 5.9). The DIP posits that in any problem there is an *ideal probability model* which is precise, but which may not be precisely known. We call this condition *ambiguity* (Ellsberg, 1961; Gilboa & Marinacci, 2013).

Facing ambiguity can be represented mathematically by a set $\mathcal{P}_{\text{prior}}$ of priors and a set $\mathcal{P}_{\text{lik}}$ of likelihoods that seem "plausible" or "fit" to express the agent's beliefs on the parameters of interest and their knowledge of the data generating process (DGP). Generally speaking, the farther apart the "boundary elements" of the sets (i.e., their infimum and supremum), the higher the agent's ambiguity. Of course, if $\mathcal{P}_{\text{prior}}$ and $\mathcal{P}_{\text{lik}}$ are singletons we go back to the usual Bayesian paradigm.

A procedure based on sets $\mathcal{P}_{\text{prior}}$ and $\mathcal{P}_{\text{lik}}$ yields results that are more robust to prior and likelihood mis-specification than a regular Bayesian method. In the presence of prior ignorance and indecisiveness about the sampling model, it is better to give answers in the form of intervals or sets, rather than arbitrarily select a prior and a likelihood, and then update. Sets $\mathcal{P}_{\text{prior}}$ and $\mathcal{P}_{\text{lik}}$ allow to represent *indecision*, thus leading to less informative but more robust and valid conclusions.

**Remark 2.** *Throughout the paper, we denote by $\Pi = \{P_1, \ldots, P_k\}$, $k \in \mathbb{N}$, a finite set of probabilities on a generic space $\Omega$, such that for all $j \in \{1, \ldots, k\}$, $P_j$ cannot be written as a convex combination of the other $k-1$ components of $\Pi$. We denote by $\Pi'$ its convex hull $\Pi' \equiv Conv(\Pi)$, i.e., the set of probabilities $Q$ on $\Omega$ that can be written as $Q(A) = \sum_{j=1}^{k} \beta_j P_j(A)$, for all $A \subset \Omega$, where the $\beta_j$'s are elements of $[0,1]$ that sum up to $1$. In the literature, it is referred to as a Finitely Generated Credal Set (FGCS, Levi (1980); Cozman (2000b)). Notice then that the extreme elements of $\Pi'$ correspond to the elements of $\Pi$, in formulas $ex\Pi' = \Pi$. Simple graphical representations of finitely generated credal sets are given in Figures 1 and 2.*

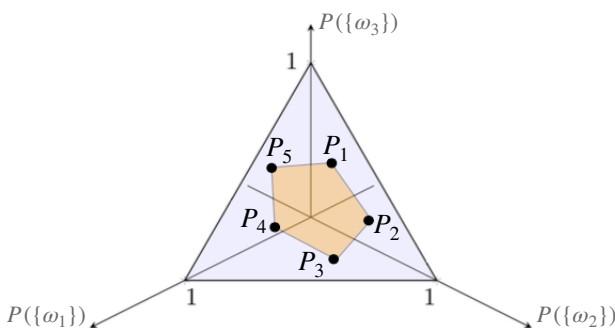

Figure 1: Suppose we are in a 3-class classification setting, so $\Omega = \{\omega_1, \omega_2, \omega_3\}$. Then, any probability measure $P$ on $\Omega$ can be seen as a probability vector. For example, suppose $P(\{\omega_1\}) = 0.6$, $P(\{\omega_2\}) = 0.3$, and $P(\{\omega_3\}) = 0.1$. We have that $P \equiv (0.6, 0.3, 0.1)^\top$. Since its elements are positive and sum up to 1, probability vector $P$ belongs to the unit simplex, the purple triangle in the figure. Then, we can specify $\Pi = \{P_1, \ldots, P_5\}$, and obtain as a consequence that $\Pi' = \text{Conv}(\Pi)$ is the orange pentagon. It is a convex polygon with finitely many extreme elements, and it is the geometric representation of a finitely generated credal set.

Let us now introduce the concepts of *lower* and *upper probabilities*. The lower probability $\underline{P}$ associated with $\Pi$ is given by $\underline{P}(A) = \inf_{P \in \Pi} P(A)$, for all $A \subset \Omega$. The upper probability $\overline{P}$ associated with $\Pi$ is defined as the conjugate to $\underline{P}$, that is, $\overline{P}(A) := 1 - \underline{P}(A^c) = \sup_{P \in \Pi} P(A)$,[7] for all $A \subset \Omega$. These definitions hold even if $\Pi$ is not finite. Then, we have the following important result.

---

[7]Here, $A^c := \Omega \setminus A$.

**Proposition 3.** $\overline{P}$ *is the upper probability for* $\Pi$ *if and only if it is also the upper probability for* $\Pi'$. *That is,* $\overline{P}(A) = \sup_{P \in \Pi} P(A) = \sup_{P' \in \Pi'} P'(A)$, *for all* $A \subset \Omega$. *The same holds for the lower probability.*

A version of Proposition 3 was proven in Dantzig (1963), while a variant for finitely additive probability measures can be found in Walley (1991, Section 3.6). A simple graphical representation of upper and lower probabilities for a set $A$ is given in Figure 2.

We now use lower probability $\underline{P}$ to define the $\alpha$-*level Imprecise Highest Density Region* (IHDR), for some $\alpha \in [0, 1]$.

**Definition 4.** *(Coolen, 1992, Section 2) Let* $\alpha$ *be any value in* $[0, 1]$. *Then, set* $IR_\alpha(\Pi') \subset \Omega$ *is called a* $(1 - \alpha)$-*Imprecise Highest Density Region (IHDR) if*

1. $\underline{P}[\{\omega \in IR_\alpha(\Pi')\}] \geq 1 - \alpha$;

2. $\int_{IR_\alpha(\Pi')} d\omega$ *is a minimum. If* $\Omega$ *is at most countable, we replace* $\int_{IR_\alpha(\Pi')} d\omega$ *with* $\#IR_\alpha(\Pi')$, *where* $\#$ *denotes the cardinality operator.*

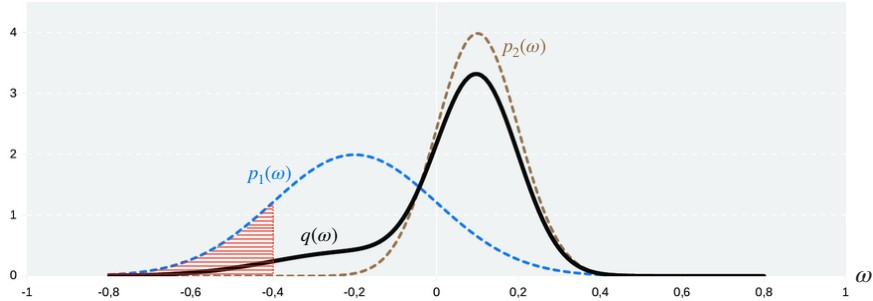

Figure 2: In this figure, a replica of Flint et al. (2017, Figure 1), $\Pi = \{P_1, P_2\}$, where $P_1$ and $P_2$ are two Normal distributions whose probability density functions (pdf's) $p_1$ and $p_2$ are given by the dashed blue and brown curves, respectively. Their convex hull is $\Pi' = \text{Conv}(\Pi) = \{Q : Q = \beta P_1 + (1 - \beta)P_2, \text{ for all } \beta \in [0, 1]\}$. The pdf $q$ of an element $Q$ of $\Pi'$ is depicted by a solid black curve. In addition, let $A = [-0.8, -0.4]$. Then, $\underline{P}(A) = \int_{-0.8}^{-0.4} p_2(\omega) d\omega \approx 0$, while $\overline{P}(A)$ is given by the red shaded area under $p_1$, that is, $\overline{P}(A) = \int_{-0.8}^{-0.4} p_1(\omega) d\omega$.

Definition 4 holds also if $\Pi = \text{ex}\Pi'$ is not finite. Notice that condition 2 is needed so that $IR_\alpha(\Pi')$ is the subset of $\Omega$ having the lowest possible cardinality, which still satisfies condition 1. By the definition of lower probability, Definition 4 implies that $P'[\{\omega \in IR_\alpha(\Pi')\}] \geq 1 - \alpha$, for all $P' \in \Pi'$. Here lies the appeal of the IHDR concept. Let us give a simple example, borrowed from Caprio et al. (2024a). Suppose $\Omega = \{\omega_1, \ldots, \omega_5\}$, $\Pi = \{P_1, P_2, P_3\}$, and $\alpha = 0.1$. The numerical values for $P_s(\{\omega_j\})$ are given in Table 2, for all $s \in \{1, 2, 3\}$ and all $j \in \{1, \ldots, 5\}$. Then, from Proposition 3 and Definition 4, we have that $IR_\alpha(\Pi') = \{\omega_1, \omega_2, \omega_3\}$.

|       | $\omega_1$ | $\omega_2$ | $\omega_3$ | $\omega_4$ | $\omega_5$ |
|-------|------|------|------|-------|-------|
| $P_1$ | 0.7  | 0.25 | 0.03 | 0.01  | 0.01  |
| $P_2$ | 0.6  | 0.2  | 0.1  | 0.05  | 0.05  |
| $P_3$ | 0.5  | 0.3  | 0.15 | 0.025 | 0.025 |

Table 2: Numerical values for our example. It is easy to see that the smallest subset of $\Omega$ that is assigned a probability of at least 0.9 by all the elements of $\Pi$ is $\{\omega_1, \omega_2, \omega_3\}$.

An operative way of building the IHDR is to consider the union of the (precise) Highest Density Regions (HDRs) of the elements of $\Pi = \text{ex}\Pi'$.[8] Let us define them formally.

---

[8]Here, by "operative" we mean that this procedure to compute the IHDR is easy to carry out in practice.

**Definition 5.** *(Coolen, 1992, Section 1) Pick any $P_j \in \Pi$, $j \in \{1, \ldots, k\}$. Let $\alpha$ be any value in $[0, 1]$. Then, set $R_\alpha(P_j) \subset \Omega$ is called a $(1 - \alpha)$-Highest Density Region (HDR) for $P_j$ if*

$$P_j[\{\omega \in R_\alpha(P_j)\}] \geq 1 - \alpha \quad and \quad \int_{R_\alpha(P_j)} d\omega \ is \ a \ minimum.$$

*If $\Omega$ is at most countable, we replace $\int_{R_\alpha(P_j)} d\omega$ with $\#R_\alpha(P_j)$. Equivalently (Hyndman, 1996),*

$$R_\alpha(P_j) = \{\omega \in \Omega : p_j(\omega) \geq p_j^\alpha\} \subset \Omega,$$

*where $p_j$ is the pdf or the probability mass function (pmf) of $P_j$, and $p_j^\alpha$ is a constant value. In particular, it is the largest constant such that $P_j[\{\omega \in R_\alpha(P_j)\}] \geq 1 - \alpha$.*

In dimension 1, $R_\alpha(P_j)$ can be interpreted as the smallest collection of elements of $\Omega$ (interval or union of intervals) to which distribution $P_j$ assigns probability of at least $1 - \alpha$. As we can see, HDRs are a Bayesian counterpart of confidence intervals.[9] We give a simple visual example in Figure 3.

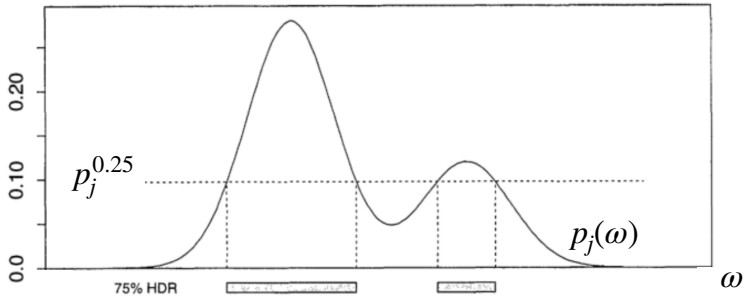

Figure 3: The 0.25-HDR from a Normal Mixture density. This picture is a replica of Hyndman (1996, Figure 1). The geometric representation of "75% probability according to $P_j$" is the area between the pdf curve $p_j(\omega)$ and the horizontal bar corresponding to $p_j^{0.25}$. A higher probability coverage (according to $P_j$) would correspond to a lower constant, so $p_j^\alpha < p_j^{0.25}$, for all $\alpha < 0.25$. In the limit, we recover 100% coverage at $p_j^0 = 0$.

As we mentioned earlier, an operative way of obtaining IHDR $IR_\alpha(\Pi')$ is by putting $IR_\alpha(\Pi') = \cup_{j=1}^k R_\alpha(P_j)$. Thanks to Proposition 3, by taking the union of the HDRs, we ensure that all the probability measures in the credal set $\Pi' = \text{Conv}(\Pi)$ assign probability of at least $1 - \alpha$ to the event $\{\omega \in IR_\alpha(\Pi')\}$. In turn, this implies that $\underline{P}[\{\omega \in IR_\alpha(\Pi')\}] = \min_{j \in \{1, \ldots, k\}} P_j[\{\omega \in IR_\alpha(\Pi')\}] \geq 1 - \alpha$. We also have that the difference between the upper and lower probabilities of $\{\omega \in IR_\alpha(\Pi')\}$ is bounded by $\alpha$. To see this, notice that $\overline{P}[\{\omega \in IR_\alpha(\Pi')\}] \leq 1$, so $\overline{P}[\{\omega \in IR_\alpha(\Pi')\}] - \underline{P}[\{\omega \in IR_\alpha(\Pi')\}] \leq \alpha$.

### 2.3 Quantifying and Disentangling Aleatoric and Epistemic Uncertainties

Recall that, given a probability measure $P$ on a generic space $\Omega$, the (Shannon) entropy of $P$ is defined as $H(P) := \mathbb{E}[-\log p] = -\int_\Omega \log[p(\omega)]P(d\omega)$ if $\Omega$ is uncountable, where $p$ denotes the pdf of $P$. If $\Omega$ is at most countable, we have that $H(P) = -\sum_{\omega \in \Omega} P(\{\omega\}) \log[P(\{\omega\})]$. As pointed out by Dubois & Hüllermeier (2007); Hüllermeier & Waegeman (2021), the entropy primarily captures the shape of the distribution, namely its "peakedness" or non-uniformity, and hence informs about the predictability of the outcome of a random experiment: the higher its value, the lower the predictability. Then, we can define the imprecise versions of the Shannon entropy as proposed by Abellán et al. (2006); Hüllermeier & Waegeman (2021), $\overline{H}(P) := \sup_{P \in \Pi'} H(P)$ and $\underline{H}(P) := \inf_{P \in \Pi'} H(P)$, called the *upper* and *lower Shannon entropy*, respectively.[10] Notice that these definitions hold for all sets of probabilities, not just for (finitely generated) credal sets. The upper entropy is a measure of total uncertainty since it represents the minimum level of predictability associated with the elements of $\Pi'$. In Abellán et al. (2006); Hüllermeier & Waegeman (2021), the authors

---

[9]Hence standard choices for the value of $\alpha$ are 0.1, 0.05, 0.01.

[10]In Appendix E, we provide bounds to the values of upper and lower entropy.

posit that it can be decomposed as a sum of aleatoric and epistemic uncertainties, and that this latter can be specified as the difference between upper and lower entropy, thus obtaining

$$\underbrace{\overline{H}(P')}_{\text{TU}(\Pi')} = \underbrace{\underline{H}(P')}_{\text{AU}(\Pi')} + \underbrace{\left[\overline{H}(P') - \underline{H}(P')\right]}_{\text{EU}(\Pi')},$$

where $\text{TU}(\Pi')$ denotes the total uncertainty associated with set $\Pi'$, $\text{AU}(\Pi')$ is the AU associated with $\Pi'$, and $\text{EU}(\Pi')$ represents the EU associated with $\Pi'$. As we can see, if $\Pi'$ is a singleton, then $\text{TU}(\Pi') = \text{AU}(\Pi')$, and $\text{EU}(\Pi') = 0$. This captures the idea that, in general, a single distribution can only gauge aleatoric uncertainty.[11] We have the following proposition.

**Proposition 6.** *Let $\Pi, \Pi'$ be sets of probability measures as the ones considered in Remark 2. Then, $\sup_{P \in \Pi} H(P) = \overline{H}(P) \leq \overline{H}(P') = \sup_{P' \in \Pi'} H(P')$ and $\inf_{P \in \Pi} H(P) = \underline{H}(P) = \underline{H}(P') = \inf_{P' \in \Pi'} H(P')$.*

Proposition 6 tells us that the upper entropy of the extreme elements in $\Pi = \text{ex}\Pi'$ is a lower bound for the upper entropy of the whole credal set $\Pi'$, and that the lower entropy of the extreme elements in $\Pi$ is equivalent to the lower entropy of the whole credal set $\Pi'$. These facts imply that $\text{AU}(\Pi') = \text{AU}(\Pi)$, and that $\text{EU}(\Pi') \geq \text{EU}(\Pi)$. In addition, as a consequence of (Smieja & Tabor, 2012, Theorem III.1), we have that $\text{EU}(\Pi') \leq \text{EU}(\Pi) + \log(\#\Pi)$. In turn, we have that $\text{EU}(\Pi') \in [\text{EU}(\Pi), \text{EU}(\Pi) + \log(\#\Pi)]$.

We briefly mention that other uncertainty measures based on credal sets are also available (see Bronevich & Rozenberg (2021); Hofman et al. (2024); Hüllermeier & Waegeman (2021) or Appendix E for a few examples) and they can be used in place of upper and lower entropy[12] to quantify EU and AU within our credal region $\mathcal{P}$, as long as the measure chosen for the total uncertainty is bounded.

Let us also add a small remark. Although EU can be reduced with an increasing amount of data, it is very seldom the case that it goes to zero when a finite amount of data is collected. When that happens, it means that the initial uncertainty was very low in the first place. Typically, EU goes to zero only asymptotically, as a finite amount of data is almost never enough to overcome initial uncertainty (Walley, 1991; Wimmer et al., 2023).

## 3 Our Procedure and Its Properties

This is the main portion of the paper. In section 3.1, we present and discuss the CBDL algorithm. Its theoretical properties are derived in section 3.2.

### 3.1 CBDL algorithm

Recall that $D = D_{\mathbf{x}} \times D_{\mathbf{y}}$ denotes the training set, where $D_{\mathbf{x}} = \{x_i\}_{i=1}^n \subset \mathcal{X}$ is the collection of training inputs, $D_{\mathbf{y}} = \{y_i\}_{i=1}^n \subset \mathcal{Y}$ is the collection of training outputs, and $\mathcal{X}$ and $\mathcal{Y}$ denote the input and output spaces, respectively. We then denote by $P$ a generic prior on the parameters $\theta \in \Theta$ of a BNN having pdf $p$, and by $L \equiv L_{x,\theta}$ a generic likelihood on the space $\mathcal{Y}$ of outputs having pdf $\ell \equiv \ell_{x,\theta}$. The act of computing the posterior from prior $P$ and likelihood $L$ using a BNN is designated by $\mathsf{post}[P, L]$, and the act of deriving the posterior predictive from posterior $P(\cdot \mid D)$ and likelihood $L$ is designated by $\mathsf{pred}[P(\cdot \mid D), L]$. The CBDL procedure is presented in Algorithm 1, and it is discussed in the following paragraphs.

During training, in **Step 1** the user specifies $K$ priors on the parameters of the neural network, that constitute the extrema $\text{ex}\mathcal{P}_{\text{prior}}$ of the prior FGCS $\mathcal{P}_{\text{prior}}$. Similarly, in **Step 2** they elicit $S$ likelihoods capturing the possible architectures of the neural network, that correspond to the extrema $\text{ex}\mathcal{P}_{\text{lik}}$ of the likelihood FGCS $\mathcal{P}_{\text{lik}}$. Let us give an example. Outlined in Jospin et al. (2022, Sections IV-B and IV-C1), for classification, the standard process for BNNs involves

---

[11]It can gauge EU, though, if it is a second-order distribution, or if it is the result of an ensemble of probabilities.

[12]Which, although easy to compute, can sometimes suffer from shortcomings coming from lacking the monotonicity property. That is, there are credal sets that, although nested into each other, have the same upper and/or lower entropy (Hüllermeier & Waegeman, 2021).

---

**Algorithm 1** Credal Bayesian Deep Learning (CBDL) – Training and Inference

---

*During Training*
**Step 1** Specify $K$ priors $\mathrm{ex}\mathcal{P}_{\mathrm{prior}} = \{P_k^{\mathrm{ex}}\}_{k=1}^K$
**Step 2** Specify $S$ likelihoods $\mathrm{ex}\mathcal{P}_{\mathrm{lik}} = \{L_s^{\mathrm{ex}}\}_{s=1}^S$
**Step 3** Compute $P_{k,s}(\cdot \mid D) = \mathsf{post}[P_k^{\mathrm{ex}}, L_s^{\mathrm{ex}}]$, for all $k$ and all $s$        ▷ % Approximated via VI by $\breve{P}_{k,s}$%
*During Inference*
    **Inputs:** New input $\tilde{x} \in \mathcal{X}$
    **Parameters:** Confidence parameter $\alpha \in [0, 1]$
    **Outputs:** Predictive Aleatoric and Epistemic Uncertainties, $\alpha$-level IHDR
**Step 4** Compute $P_{k,s}^{\mathrm{pred}} = \mathsf{pred}[\breve{P}_{k,s}, L_s^{\mathrm{ex}}]$, for all $k$ and all $s$        ▷ % Approximated via VI by $\hat{P}_{k,s}^{\mathrm{pred}}$%
**Step 5** Compute and return $\mathrm{AU}(\hat{\mathcal{P}}_{\mathrm{pred}})$ and the bounds for $\mathrm{EU}(\hat{\mathcal{P}}_{\mathrm{pred}})$
**Step 6** Compute and return the $(1 - \alpha)$-IHDR $IR_\alpha(\hat{\mathcal{P}}_{\mathrm{pred}})$

---

- A Normal prior with zero mean and diagonal covariance $\sigma^2 I$ on the coefficients of the network, that is, $p(\theta) = \mathcal{N}(0, \sigma^2 I)$. In the context of CBDL, we could specify, e.g. $\mathrm{ex}\mathcal{P}_{\mathrm{prior}} = \{P : p(\theta) = \mathcal{N}(\mu, \sigma^2 I), \mu \in \{\mu_-, \mathbf{0}, \mu_+\}, \sigma^2 \in \{3, 7\}\}$. That is, the extreme elements of the prior credal set are five independent Normals having different levels of "fatness" of the tails, and centered at a vector $\mu_+$ having positive entries, a vector $\mu_-$ having negative entries, and a vector $\mathbf{0}$ having entries equal to 0. They capture the ideas of positive bias, negative bias, and no bias of the coefficients, respectively. This is done to hedge against possible prior misspecification.

- A Categorical likelihood, $p(y \mid x, \theta) = \mathrm{Cat}(\Phi_\theta(x))$, whose parameter is given by the output of a functional model $\Phi_\theta$. In the context of CBDL, we could specify the set of extreme elements of the likelihood credal set as $\mathrm{ex}\mathcal{P}_{\mathrm{lik}} = \{L : \ell_{x,\theta}(y) = \mathrm{Cat}(\Phi_{s,\theta}(x)), s \in \{1, \ldots, S\}\}$. Specifying set $\mathrm{ex}\mathcal{P}_{\mathrm{lik}}$, then, corresponds to eliciting a finite number $S$ of possible (parametrized) architectures for the neural network $\Phi_{s,\theta}$, $s \in \{1, \ldots, S\}$, and obtain, as a consequence, $S$ categorical distributions $\{\mathrm{Cat}(\Phi_{s,\theta}(x))\}_{s=1}^S$. This captures the ambiguity around the true data generating process faced by the agent, and allows them to hedge against likelihood misspecification.

More in general, we can use the priors and likelihoods that better fit the type of analysis we are performing.[13] For example, for the choice of the priors we refer to Fortuin et al. (2021), where the authors study the problem of selecting the right type of prior for BNNs.

**Step 3** performs an element-wise application of Bayes' rule for all the elements of $\mathrm{ex}\mathcal{P}_{\mathrm{prior}}$ and $\mathrm{ex}\mathcal{P}_{\mathrm{lik}}$. Each posterior is approximated using the Variational Inference (VI) method (Jospin et al., 2022, Section V). That is, we project every posterior $P_{k,s}(\cdot \mid D)$ onto a set $\mathbb{S}$ of "well-behaved" distributions (e.g. Normals) using the KL divergence. In formulas, $\breve{P}_{k,s} = \arg\min_{Q \in \mathbb{S}} KL[Q \| P_{k,s}(\cdot \mid D)]$. By "well-behaved", we mean that they have to satisfy the conditions in Zhang & Gao (2020, Sections 2 and 3).[14] This ensures that as the sample size goes to infinity, the approximated posteriors converge to the true data generating process. We also point out how despite using a VI approximation for the exact posteriors, as we shall see in the next section, the credal set of approximated posteriors is closer to the "oracle" posterior $P^o(\cdot \mid D)$ than any of its elements taken individually. As a consequence, working with credal sets leads to VI posterior approximations that are better than the ones resulting from a single BNN, or an ensemble of BNNs, where several BNNs are combined into one.

**Remark 7.** *Although highly unlikely in practice, it is theoretically possible that the VI approximation of the (finite) set $\{P_{k,s}(\cdot \mid D)\}_{k,s}$ of posteriors is a singleton, see Figure 4. While the conditions in Zhang & Gao (2020) guarantee that asymptotically the approximated posteriors coincide with the true data generating process, this typically does not happen with finite-dimensional datasets. As a consequence, obtaining a singleton when projecting $\{P_{k,s}(\cdot \mid D)\}_{k,s}$ onto $\mathbb{S}$ may result in an underestimation of the uncertainties faced by the user. In that case, we either consider a different set – whose elements still satisfy the conditions in Zhang*

---

[13]Choosing 2 to 5 priors and likelihoods is usually enough to safely hedge against prior and likelihood misspecification.
[14]We assume that the conditions on the priors and the likelihoods given in Zhang & Gao (2020) are satisfied.

*& Gao (2020, Sections 2 and 3) – on which to project $\{P_{k,s}(\cdot \mid D)\}_{k,s}$ according to the KL divergence, or we use a different "projection operator", that is, a divergence different from the KL. For example, Rényi and $\chi^2$ divergences, or Hellinger and total variation metrics are suggested by Zhang & Gao (2020). Alternatively, we can consider a different approximation strategy altogether, for instance the Laplace approximation (Ritter et al., 2018).*

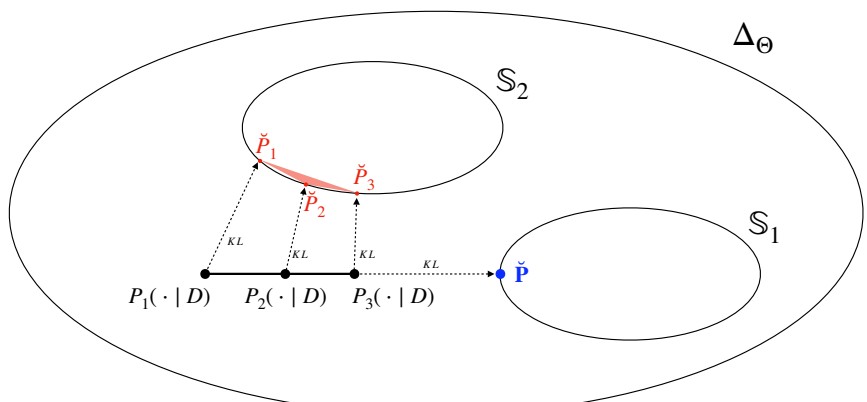

Figure 4: Let $\Delta_\Theta$ denote the space of probability measures on $\Theta$. Suppose that in the analysis at hand we specified three priors and only one likelihood, so $S = 1$ and we can drop the $s$ index. Let $\{P_k(\cdot \mid D)\}_{k=1}^3$ be the collection of exact posteriors, so that the black segment represents the exact posterior FGCS. Then, if we project the elements of $\{P_k(\cdot \mid D)\}_{k=1}^3$ onto $\mathbb{S}_1$ via the KL divergence, we obtain the same distribution $\check{\mathbf{P}}$. This is detrimental to the analysis because such an approximation underestimates the (posterior) epistemic (and possibly also aleatoric) uncertainty faced by the agent. Then, the user could specify a different set $\mathbb{S}_2$ of "well-behaved" distributions onto which project the elements of $\{P_k(\cdot \mid D)\}_{k=1}^3$. In the figure, we see that they are projected onto $\mathbb{S}_2$ via the KL divergence to obtain $\check{P}_1$, $\check{P}_2$, and $\check{P}_3$. The convex hull of these latter, captured by the red shaded triangle, represents the variational approximation of the exact posterior FGCS.

After **Step 3**, we obtain a finite set $\{\check{P}_{k,s}\}_{k,s}$ of VI approximation of the posteriors on the network parameters, whose cardinality is $K \times S$. Its convex hull constitutes $\check{\mathcal{P}}_{\text{post}}$, that is, the VI-approximated posterior FGCS. We assume that $\text{ex}\check{\mathcal{P}}_{\text{post}} = \{\check{P}_{k,s}\}_{k,s}$. This is an assumption because it may well be that – due to the approximation procedure – some of the elements of $\{\check{P}_{k,s}\}_{k,s}$ are not independent of one another. We defer to future work the design of a procedure that finds the elements of $\{\check{P}_{k,s}\}_{k,s}$ that cannot be written as a convex combination of one another.

Being a combinatorial task, **Step 3** is a computational bottleneck of Algorithm 1. We have to calculate $K \times S$ VI approximations to as many posteriors, but this allows us to forego any additional assumptions on the nature of the lower and upper probabilities that are oftentimes required by other imprecise-probabilities-based techniques.[15] Clearly, CBDL is simplified if either $\mathcal{P}_{\text{prior}}$ or $\mathcal{P}_{\text{lik}}$ are singletons. Notice that in the case that $\mathcal{P}_{\text{prior}}$ and $\mathcal{P}_{\text{lik}}$ are both not singletons, for all $A \subset \Theta$ the interval $[\underline{\check{P}}(A), \overline{\check{P}}(A)]$ is wider than the case when one or the other is a singleton. In the limiting case where both are singletons, we retrieve the usual Bayesian updating, so the interval shrinks down to a point.

Before moving to inference time, let us remark a difference between Bayesian Model Averaging (BMA) and CBDL. In BMA, the user specifies a distribution on the models. Translated in the notation we use in this work, this means having a discrete distribution $Q$ over the $K \times S$ prior-likelihood combinations, that is, over the elements of $\text{ex}\check{\mathcal{P}}_{\text{post}}$. Such $Q$ is then used to select an element $\check{P}^\star$ from the (VI-approximated) posterior FGCS $\check{\mathcal{P}}_{\text{post}}$ as

$$\check{\mathcal{P}}_{\text{post}} \ni \check{P}^\star = \sum_{k,s} Q(\{\check{P}_{k,s}\})\check{P}_{k,s}, \tag{1}$$

---

[15]If we are willing to make such assumptions, Theorem 10 in Appendix D shows how to compute the upper posterior using only upper prior and upper likelihood.

where $Q(\{\breve{P}_{k,s}\}) \in [0,1]$ for all $k$ and all $s$, and $\sum_{k,s} Q(\{\breve{P}_{k,s}\}) = 1$ (Caprio & Mukherjee, 2023a). For example, if $Q$ is the discrete uniform distribution on $\{1, \ldots, K \times S\}$, then $\breve{P}^\star$ is the so-called *center of gravity* of the credal set $\breve{\mathcal{P}}_{\text{post}}$ (Miranda & Montes, 2023, Section 3.2). Instead, CBDL does not select a unique distribution from $\breve{\mathcal{P}}_{\text{post}}$. Rather, distributions are kept separate so to be able to derive a predictive FGCS in **Step 4** of Algorithm 1, that is in turn used to quantify and disentangle predictive uncertainties, and to compute the predictive IHDR, that is, a collection of outputs having a high probability of being the correct ones for a new input $\tilde{x}$.

During inference, a new input $\tilde{x}$ is provided. In **Step 4**, every element of $\text{ex}\breve{\mathcal{P}}_{\text{post}}$ is used to derive a predictive distribution $P_{k,s}^{\text{pred}} = \text{pred}[\breve{P}_{k,s}, L_s^{\text{ex}}]$ on the output space $\mathcal{Y}$. In particular, for every $k$ and every $s$, the pdf $p_{k,s}^{\text{pred}}$ of $P_{k,s}^{\text{pred}}$ is obtained as

$$p_{k,s}^{\text{pred}}(\tilde{y} \mid \tilde{x}, x_1, y_1, \ldots, x_n, y_n) = \int_\Theta \ell_s^{\text{ex}}(\tilde{y} \mid \tilde{x}, \theta) \cdot p_{k,s}(\theta \mid x_1, y_1, \ldots, x_n, y_n)\mathrm{d}\theta$$
$$\approx \int_\Theta \ell_s^{\text{ex}}(\tilde{y} \mid \tilde{x}, \theta) \cdot \breve{p}_{k,s}(\theta)\mathrm{d}\theta,$$

where $\ell_s^{\text{ex}}$ is the pdf of likelihood $L_s^{\text{ex}}$, $p_{k,s}$ is the pdf of the true posterior $P_{k,s}(\cdot \mid D)$, $\breve{p}_{k,s}$ is the pdf of the VI-approximated posterior $\breve{P}_{k,s}$, and $\tilde{y}$ is the output associated to the new input $\tilde{x}$ (see Appendix F for more details). Each predictive distribution $P_{k,s}^{\text{pred}}$ is approximated to $\hat{P}_{k,s}^{\text{pred}}$ (e.g. using Normals) via Variational Inference. The convex hull of the collection $\{\hat{P}_{k,s}^{\text{pred}}\}_{k,s}$ having cardinality $K \times S$ constitutes the VI-approximated predictive FGCS $\hat{\mathcal{P}}_{\text{pred}}$. Similarly to what we did for $\text{ex}\breve{\mathcal{P}}_{\text{post}}$, we assume that $\text{ex}\hat{\mathcal{P}}_{\text{pred}} = \{\hat{P}_{k,s}^{\text{pred}}\}_{k,s}$.

In **Step 5**, building on the results in section 2.3, and in particular on Proposition 6, we compute and return $\text{AU}(\hat{\mathcal{P}}_{\text{pred}})$ and the bounds for $\text{EU}(\hat{\mathcal{P}}_{\text{pred}})$. In particular, we have

$$\text{AU}(\mathcal{P}_{\text{pred}}) \approx \text{AU}(\hat{\mathcal{P}}_{\text{pred}}) = \underline{H}(\hat{P}^{\text{pred}}) \tag{2}$$

and

$$\text{EU}(\mathcal{P}_{\text{pred}}) \approx \text{EU}(\hat{\mathcal{P}}_{\text{pred}}) \in [\overline{H}(\hat{P}^{\text{pred}}) - \underline{H}(\hat{P}^{\text{pred}}), \overline{H}(\hat{P}^{\text{pred}}) - \underline{H}(\hat{P}^{\text{pred}}) + \log(K \times S)], \tag{3}$$

where (i) $\underline{H}(\hat{P}^{\text{pred}}) = \min_{k,s} H(\hat{P}_{k,s}^{\text{pred}})$; (ii) $\overline{H}(\hat{P}^{\text{pred}}) = \max_{k,s} H(\hat{P}_{k,s}^{\text{pred}})$; and (iii) $\mathcal{P}_{\text{pred}}$ is the "true" predictive FGCS, that is, the one we would have obtained had we been able to compute the exact posteriors in **Step 3** and the exact predictive distributions in **Step 4**. Let us point out a salient feature of CBDL. The AU and the (bounds for) the EU associated with the VI-approximated predictive FGCS $\hat{\mathcal{P}}_{\text{pred}}$ embed uncertainty comparable to an uncountably infinite ensemble of BNNs, i.e., an ensemble of BNNs of cardinality $\aleph_1$, despite the simple and intuitive mathematics over the finite set $\text{ex}\hat{\mathcal{P}}_{\text{pred}}$. They are not merely pessimistic results on the uncertainties associated with a finite ensemble of BNNs.

We observe that we compute the AU and EU associated with the (VI-approximated) predictive credal set $\hat{\mathcal{P}}_{\text{pred}}$, rather than those related to the (VI-approximated) posterior credal set $\breve{\mathcal{P}}_{\text{post}}$. We do so because we are ultimately interested in reporting the uncertainty around the predicted outputs given a new input in the problem at hand, more than the uncertainty on the parameters of the NN.

Before commenting on the next step, let us pause here and add a remark. While a "bad choice" of priors and likelihoods may lead to large values of upper and lower entropy – $\overline{H}(\hat{P}^{\text{pred}})$ and $\underline{H}(\hat{P}^{\text{pred}})$, respectively – this is not a risk confined to our procedure. Poor modeling choices are an unavoidable risk in model-based techniques. This gave rise to the famous adage by George Box "essentially, all models are wrong but some are useful" (Box, 1976).[16] We maintain that our method is indeed useful, since it overcomes some of the shortcomings of traditional Bayesian techniques – as explained in Appendices A and B. As for "regular" Bayesian methods, though, for our approach too the designer will need to make "plausible" choices for priors

---

[16]Curiously, a similar motivation was brought forward by Pseudo-Dionysius the Areopagite in favor of the use of sacred images in the Christian tradition (Migne, 1857). While they do not capture the essence of God, these defective approximations help the believer's thought to elevate.

and likelihoods. A further positive aspect of working with credal sets is that they are able to "self-regulate". That is, in the case of prior-likelihood conflict – which happens e.g. if the prior set is ill-specified (Reimherr et al., 2021) – the posterior credal set, and in turn the predictive credal set too, will be wider. This will be reflected in the uncertainty measures which will register an excess of posterior, and in turn predictive, epistemic uncertainties.

Finally, in **Step 6**, we compute and return the $\alpha$-level Imprecise Highest Density Region $IR_\alpha(\hat{\mathcal{P}}_{\mathrm{pred}})$ for the VI-approximated predictive FGCS $\hat{\mathcal{P}}_{\mathrm{pred}}$, which approximates the IHDR for the "true" predictive FGCS $\mathcal{P}_{\mathrm{pred}}$. It is the smallest subset of $\mathcal{Y}$ such that $\hat{P}^{\mathrm{pred}}[\{\tilde{y} \in IR_\alpha(\hat{\mathcal{P}}_{\mathrm{pred}})\}] \geq 1 - \alpha$, for all $\hat{P}^{\mathrm{pred}} \in \hat{\mathcal{P}}_{\mathrm{pred}}$, and for some $\alpha \in [0,1]$. It can be interpreted as the smallest collection of outputs $\tilde{y}$ that have a high probability of being the correct ones for the new input $\tilde{x}$, according to all the distributions in $\hat{\mathcal{P}}_{\mathrm{pred}}$. Or equivalently, we can say that the correct output for the new input $\tilde{x}$ belongs to IHDR $IR_\alpha(\hat{\mathcal{P}}_{\mathrm{pred}})$ with lower probability of at least $1 - \alpha$, a probabilistic guarantee for the set of outputs generated by our procedure. Notice also that the size of $IR_\alpha(\hat{\mathcal{P}}_{\mathrm{pred}})$ is an increasing function of both (predictive) AU and EU. As a consequence, it is related, but it is not equal, to the (predictive) AU the agent faces. If we want to avoid to perform the procedure only to discover that $IR_\alpha(\hat{\mathcal{P}}_{\mathrm{pred}})$ is "too large", then we can add an "AU check" after **Step 5**. This, together with computing $IR_\alpha(\hat{\mathcal{P}}_{\mathrm{pred}})$ in a classification setting, is explored in Appendices G and H.

As we have seen in section 2.2, we find $IR_\alpha(\hat{\mathcal{P}}_{\mathrm{pred}})$ by taking the union of the $K \times S$ HDRs $R_\alpha(\hat{P}^{\mathrm{pred}}_{k,s})$ of the extreme elements $\mathrm{ex}\hat{\mathcal{P}}_{\mathrm{pred}}$ of $\hat{\mathcal{P}}_{\mathrm{pred}}$. The HDR of a well-known distribution (for instance, a Normal) can be routinely obtained in R, e.g. using package HDInterval (Juat et al., 2022).

We conclude this section with two remarks. First, we point out how CBDL does not depend on the method used to approximate the posterior and the predictive distributions: Variational Inference can be substituted by other approaches. CBDL can also be easily adapted to other TU, AU, and EU measures, as long as the measure chosen for the total uncertainty is bounded. Second, **Step 6** can be effortlessly modified so that CBDL produces the collection of outputs having the highest lower density. This is in line with the Naive Credal Classifier theory (Cozman, 2000a; Zaffalon, 2002). In this case, we forego control on the accuracy level $1 - \alpha$ of the output region produced by CBDL. It is implemented as follows. In the modified version of **Step 6**, we compute

$$\underset{\tilde{y} \in \mathcal{Y}}{\arg\max} \min_{k,s} \hat{p}^{\mathrm{pred}}_{k,s}(\tilde{y}),$$

where $\hat{p}^{\mathrm{pred}}_{k,s}$ is the pdf of the VI-approximated predictive distribution $\hat{P}^{\mathrm{pred}}_{k,s}$. If such arg max is not a singleton, and the user is set on CBDL outputting a unique value $\tilde{y}$ for the new input $\tilde{x}$, they can select one element uniformly at random from the arg max.

## 3.2 Theoretical Properties of CBDL

Working with credal sets makes CBDL more robust to distribution misspecification and shifts than single BNNs. To see this, we present the following general result, and then we apply it to our case.

Let $\Pi'$ be an FGCS as in Remark 2, and consider a probability measure $\Psi$ such that $\Psi \notin \Pi'$.

**Proposition 8.** *Call d any metric and div any divergence on the space of probability measures of interest. Let $d(\Pi', \Psi) := \inf_{P' \in \Pi'} d(P', \Psi)$ and $div(\Pi'\|\Psi) := \inf_{P' \in \Pi'} div(P'\|\Psi)$. Then, for all $P' \in \Pi'$, $d(\Pi', \Psi) \leq d(P', \Psi)$ and $div(\Pi', \Psi) \leq div(P'\|\Psi)$.*

Proposition 8 holds if $\Pi'$ is any set of probabilities, not just an FGCS.[17] In Appendix I, we show that the above result still holds if the elements of $\Pi'$ and $\Psi$ are defined on Euclidean spaces having different dimensions (Cai & Lim, 2022; Caprio, 2022).

Let us now apply Proposition 8 to CBDL. Suppose that, when designing a single BNN, an agent chooses likelihood **L**, while when implementing CBDL, they specify in **Step 2** a finite set of likelihoods $\mathrm{ex}\mathcal{P}_{\mathrm{lik}} = \{L^{\mathrm{ex}}_s\}^S_{s=1}$, $S \geq 2$, and then let the induced credal set $\mathcal{P}_{\mathrm{lik}} = \mathrm{Conv}(\mathrm{ex}\mathcal{P}_{\mathrm{lik}})$ represent their uncertainty around

---

[17]More in general, Proposition 8 holds for any type of function $f$, not just metrics or divergences, because Proposition 8 essentially only applies the definition of the infimum.

the sampling model. Assume that $\mathbf{L} \in \mathrm{ex}\mathcal{P}_{\mathrm{lik}}$. This means that when designing the single BNN, the agent chooses arbitrarily which of the elements of $\mathrm{ex}\mathcal{P}_{\mathrm{lik}}$ to use. Suppose also that the "oracle" data generating process $L^o$ is different from $\mathbf{L}$, $L^o \neq \mathbf{L}$, so that we are actually in the presence of likelihood misspecification. Then, we have two cases. (1) If the true sampling model $L^o$ belongs to $\mathcal{P}_{\mathrm{lik}}$, then the distance – measured via a metric or a divergence – between $\mathcal{P}_{\mathrm{lik}}$ and $L^o$ is 0, while that between $\mathbf{L}$ and $L^o$ is positive. (2) If $L^o \notin \mathcal{P}_{\mathrm{lik}}$, then the distance between $\mathcal{P}_{\mathrm{lik}}$ and $L^o$ is no larger than the distance between $\mathbf{L}$ and $L^o$, no matter (i) which metric or distance we use (Proposition 8), (ii) whether or not $L^o$ and the elements of $\mathcal{P}_{\mathrm{lik}}$ are defined on the same Euclidean space (Appendix I, Lemma 17). A visual representation is given in Figure 5. A similar argument holds if part of the training dataset $D$ is generated by one distribution $L_1^o$, and the remaining part is generated by $L_2^o$, a situation that we can categorize as distribution shift. Even if – in the case of a single BNN – the user were able to elicit exactly $\mathbf{L} = L_1^o$, the likelihood would still be misspecified for the data that come from $L_2^o$. This shortcoming is overcome by a credal set approach, in which either both oracle distributions $L_1^o$ and $L_2^o$ belong to the credal set $\mathcal{P}_{\mathrm{lik}}$, or the distance between $\mathcal{P}_{\mathrm{lik}}$ and either of them is no larger than that between the single chosen one $\mathbf{L}$, and $L_1^o$ and $L_2^o$.

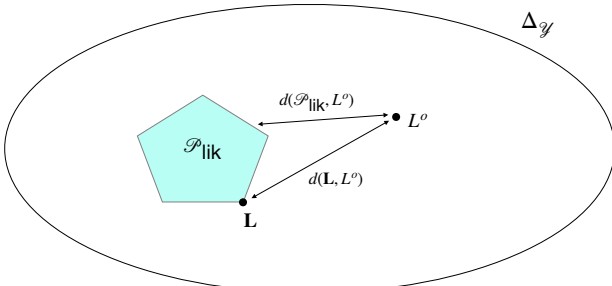

Figure 5: CBDL is more robust to distribution shifts than single BNNs. Here $\mathcal{P}_{\mathrm{lik}}$ is the convex hull of five plausible likelihoods, and $d$ denotes a generic metric on the space $\Delta_{\mathcal{Y}}$ of probabilities on $\mathcal{Y}$. We see how $d(\mathcal{P}_{\mathrm{lik}}, L^o) < d(\mathbf{L}, L^o)$; if we replace metric $d$ by a generic divergence div, the inequality would still hold.

Let us add a remark here. Assume that the "oracle" prior $P^o$ is in the prior credal set $\mathcal{P}_{\mathrm{prior}}$ and that the "oracle" likelihood $L^o$ is in the likelihood credal set $\mathcal{P}_{\mathrm{lik}}$. Then, it is immediate to see that the "oracle" posterior $P^o(\cdot \mid D)$ belongs to the posterior credal set $\mathcal{P}_{\mathrm{post}}$. Naturally, this does not imply that the posterior credal set collapses to $P^o(\cdot \mid D)$. In general, it is unlikely that a finite amount of data is able to completely annihilate all the epistemic uncertainty faced by the agent. What may happen is that if the training set is large enough, $\mathcal{P}_{\mathrm{post}}$ may be inscribed in a ball of small radius around $P^o(\cdot \mid D)$. This does not mean that we suffer from under-confidence due to larger-than-necessary epistemic uncertainty. Rather, the relative epistemic uncertainty (measured by the difference between prior and posterior uncertainty, divided by the prior uncertainty) drops significantly. In addition, working with sets of prior and likelihoods allows us to hedge against prior and likelihood misspecification, a consequence of Proposition 8.

## 4 Experiments

From the previous sections, it is clear that CBDL improves on the uncertainty quantification capabilities of single BNNs.[18] CBDL allows a better quantification of predictive AU because of its robustness to misspecification stemming from Proposition 8. In a sense, the predictive AU quantified by a single BNN is a function of the choices of prior and likelihood made by the user. In addition, as we pointed out before (e.g. in the last paragraph of section 2.1), predictive EU cannot be obtained in a theoretically principled manner from a single BNN (Hüllermeier & Waegeman, 2021; Fellaji & Pennerath, 2024).

Here, we show that these theoretical arguments are also backed by empirical evidence. In particular, in section 4.1 we compare CBDL to the method proposed in Krishnan & Tickoo (2020), in which a single Bayesian Neural Network's output is utilized to estimate both predictive epistemic and aleatoric uncertainties. We also show that CBDL improves on the ensemble of BNNs (EBNN) proposed in Cobb et al. (2019), that we

---

[18]This includes empirical Bayes methodologies (Krishnan et al., 2020).

treat as an ablation that is a naive extension of (Krishnan & Tickoo, 2020) to multiple BNNs. In section 4.2, we analyze the downstream task performance of CBDL. We show that it performs better than an ensemble of BNNs (EBNN) (Cobb et al., 2019). To demonstrate the utility of our method, we study the behavior of certain safety-critical settings under distribution shifts and its ramifications. One, for motion prediction in autonomous driving scenarios, and two, to model blood glucose and insulin dynamics for artificial pancreas control.

## 4.1  (Predictive) Uncertainty Quantification

Distribution shifts can introduce uncertainties in a system, which in turn can render the predictions meaningless. This can be due to naturally occurring corruptions, as introduced in Hendrycks & Dietterich (2019) for image datasets. The authors introduced 18 different noise types, which can be varied across 5 different severity levels, ranging from low (severity = 1), to medium (severity = 2, 3), and high (severity = 4, 5). The intuition is that in the current context, increasing the noise severity should generally result in higher uncertainty. We evaluate our CBDL method on 4 standard image datasets, CIFAR-10 (Krizhevsky et al., 2009), SVHN (Netzer et al., 2011), Fashion-MNIST (Xiao et al., 2017), and MNIST (Lecun et al., 1998). We use a slightly different set of perturbations than those introduced in Mu & Gilmer (2019) for gray-scale images like MNIST and Fashion-MNIST. Additionally, we perform cross domain testing for each dataset, where we expect the predictive uncertainties to be higher. We implement and train a Resnet-20 Bayesian Neural Network model inside the library Bayesian-torch (Krishnan et al., 2019). For each dataset, we train 4 different networks initialized with different seeds on the prior and with the same architecture. This corresponds to eliciting a prior FGCS $\mathcal{P}_{\mathrm{prior}}$ such that $\mathrm{ex}\mathcal{P}_{\mathrm{prior}} = \{P_1^{\mathrm{ex}}, \ldots, P_4^{\mathrm{ex}}\}$, so that $K$ in **Step 1** of Algorithm 1 is equal to 4, and a likelihood FGCS that is a singleton, $\mathcal{P}_{\mathrm{lik}} = \mathrm{ex}\mathcal{P}_{\mathrm{lik}} = \{L\}$, so that $S$ in **Step 2** of Algorithm 1 is equal to 1. We use a learning-rate of 0.001, batch-size of 128, and train the networks using Mean-Field Variational Inference for 200 epochs. The inference is carried out by performing multiple forward passes through parameters drawn from the posterior distribution. We used 20 Monte-Carlo samples in the experiments.

**Baselines.** In Krishnan & Tickoo (2020), for a single BNN output, the predictive distribution is obtained through multiple stochastic forward passes on the network while sampling from the weight posteriors using Monte Carlo estimators. In their work, they define the overall entropy and the entropy of the predictive distribution as the *predictive entropy* (Krishnan & Tickoo, 2020, (C.2)). This quantity captures a combination of predictive aleatoric and epistemic uncertainties. Additionally, they define the *mutual information* between weight posterior and predictive distribution as the predictive epistemic uncertainty. Finally, the predictive aleatoric uncertainty is defined as the *expected entropy*. The predictive epistemic and aleatoric uncertainties sum to the predictive entropy. In our experiments, we use these definitions to compute the respective quantities.

In Cobb et al. (2019), the authors consider different BNNs, but instead of keeping them separate and use them to build a predictive credal set, they average them out. Similar to theirs, we elicit the following procedure, that we call ensemble of BNNs (EBNN).

Consider $R \in \mathbb{N}_{\geq 2}$ different BNNs, and compute the posterior distribution on the parameters. They induce $R$ predictive distributions on the output space $\mathcal{Y}$, each having mean $\mu_r$ and variance $\sigma_r^2$, $r \in \{1, \ldots, R\}$. We call *EBNN distribution* $P_{\mathrm{ens}}$ a Normal having mean $\mu_{\mathrm{ens}} = 1/R \sum_{r=1}^R \mu_r$ and covariance matrix $\sigma_{\mathrm{ens}}^2 I$, where $\sigma_{\mathrm{ens}}^2 = 1/R \sum_{r=1}^R \sigma_r^2 + 1/(R-1) \sum_{r=1}^R (\mu_r - \mu_{\mathrm{ens}})^2$. In section 4.2, we use the $\alpha$-level HDR $R_\alpha(P_{\mathrm{ens}})$ associated with $P_{\mathrm{ens}}$ as a baseline for the IHDR $IR_\alpha(\hat{\mathcal{P}}_{\mathrm{pred}})$ computed at **Step 6** of Algorithm 1.

Following Cobb et al. (2019), for EBNN we posit that $1/R \sum_{r=1}^R \sigma_r^2$ captures the (predictive) aleatoric uncertainty associated with $P_{\mathrm{ens}}$, and $1/(R-1) \sum_{r=1}^R (\mu_r - \mu_{\mathrm{ens}})^2$ captures the (predictive) epistemic uncertainty associated with $P_{\mathrm{ens}}$.[19] For CBDL, we look at the value of $\mathrm{AU}(\hat{\mathcal{P}}_{\mathrm{pred}}) = \underline{H}(\hat{P}^{\mathrm{pred}})$ from equation 2, and at the lower bound $\overline{H}(\hat{P}^{\mathrm{pred}}) - \underline{H}(\hat{P}^{\mathrm{pred}})$ for $\mathrm{EU}(\hat{\mathcal{P}}_{\mathrm{pred}})$ from equation 3. It is enough to focus on the lower

---

[19]Notice that in this case we retain the assumption that the EBNN distribution $P_{\mathrm{ens}}$ on $\mathcal{Y}$ has mean $\mu_{\mathrm{ens}}$ and covariance matrix $\sigma_{\mathrm{ens}}^2 I$, but we do not require that it is a Normal. That is because in the four image datasets that we consider, the output space $\mathcal{Y}$ is finite.

bound because the upper bound is given by $\overline{H}(\hat{P}^{\mathrm{pred}}) - \underline{H}(\hat{P}^{\mathrm{pred}}) + \log(4)$, and $\log(4) \approx 0.6$ is a fixed value. Hence, the trend of both lower and upper bounds for $\mathrm{EU}(\hat{\mathcal{P}}_{\mathrm{pred}})$ as the severity of corruption changes is the same.

**Summary of our Results.** Two scenarios are evaluated: an In-Distribution (ID) scenario with corruption by noises of increasing severity, and an Out-Of-Distribution (OOD) assessment. The following main points summarize our findings:

1. (ID Evaluation) For both our proposed method and that of Krishnan & Tickoo (2020), for increasing severities of noise corruption, predictive epistemic and aleatoric uncertainties increase. In comparison, the baseline EBNN method demonstrates a counterintuitive result, where the predictive aleatoric uncertainty decreases as the severity of corruption increases. The table of values can be found in Appendix N.

2. (ID Evaluation) To assess the utility of our proposed method in downstream tasks, we analyze the accuracy versus rejection rate for all methods. Table 3 summarizes the results. Across MNIST-C and Fashion-MNIST-C, CBDL exhibits a higher average accuracy across the range of rejection rates. For the remaining datasets, it demonstrates results comparable to the baseline (see section 4.1.1 for evaluation details)

3. (OOD Evaluation) To evaluate OOD detection, models trained on a particular dataset are tested on the other three datasets with the goal of analyzing the behavior of the predictive uncertainties. Tables 4 and 5 summarize the results. Across all the evaluated datasets, CBDL had the most consistent behavior of exhibiting a large increase in predictive AU, relative to the increase in predictive EU when tested on an OOD dataset. In comparison, EBNN shows a decrease in predictive AU, and the single BNN shows inconsistent behavior depending on the dataset.

| Severity | CIFAR-10C | | | MNIST-C | | | Fashion MNIST-C | | | SVHN | | |
|---|---|---|---|---|---|---|---|---|---|---|---|---|
| | BNN | Ensemble | CBDL | BNN | Ensemble | CBDL | BNN | Ensemble | CBDL | BNN | Ensemble | CBDL |
| 1 | 0.754 | 0.983 | 0.982 | 0.645 | 0.919 | 0.929 | 0.525 | 0.730 | 0.748 | 0.698 | 0.913 | 0.909 |
| 2 | 0.761 | 0.947 | 0.946 | 0.680 | 0.879 | 0.896 | 0.537 | 0.655 | 0.641 | 0.703 | 0.843 | 0.835 |
| 3 | 0.749 | 0.906 | 0.907 | 0.617 | 0.804 | 0.841 | 0.442 | 0.498 | 0.529 | 0.669 | 0.764 | 0.758 |
| 4 | 0.733 | 0.838 | 0.836 | 0.639 | 0.713 | 0.737 | 0.332 | 0.421 | 0.489 | 0.608 | 0.586 | 0.588 |
| 5 | 0.661 | 0.734 | 0.734 | 0.404 | 0.495 | 0.539 | 0.300 | 0.351 | 0.391 | 0.455 | 0.378 | 0.385 |
| Mean | 0.732 | **0.882** | 0.881 | 0.597 | 0.762 | **0.788** | 0.433 | 0.531 | **0.560** | 0.627 | **0.697** | 0.695 |

Table 3: In-Distribution evaluation on noise corruptions of increasing severity. Average area under accuracy vs rejection rate curve. Higher numbers indicate overall better accuracy over a range of rejection rates.

### 4.1.1 In-distribution Evaluation

Overall, CBDL demonstrates the most consistent behavior in terms of increasing levels of predictive epistemic and aleatoric uncertainties as the severity of noise corruption increases. The full table of results can be found in Appendix N. In our experiments, for CIFAR-10 the single BNN also exhibits similarly consistent behavior. In MNIST and Fashion-MNIST, the BNN seems to be more consistent than CBDL, whereas for SVHN it seems to show a more inconsistent behavior. These results allow us to conclude that CBDL is comparable if not better than BNN in terms of In-Distribution behavior.

**Accuracy vs Rejection Rate.** As the quantities pertaining to different approaches are not comparable, we perform an additional evaluation comparing the accuracy vs the rejection rate of a given method. In this task, for a given uncertainty threshold, if the uncertainty quantity exceeds such a threshold, then the prediction is rejected. Of those not rejected, the accuracy is computed. A range of uncertainty thresholds is tested for each method and plotted. Figure 6 depicts an example of the plot. There, for lower severities (e.g. 1-2) of Gaussian blur noise, CBDL exhibits better performance and converges to 100% accuracy at a lower rejection rate. Intuitively, for the same range of rejection rates, a higher curve signifies better performance, as less samples are rejected while achieving a higher accuracy. However, for higher severities (above 3), the single BNN seems to show better performance. Even so, the overall rejection rate is high to achieve such accuracy. The entirety of the results can be found in Appendix P. To quantify the differences, the areas

| | | CBDL | | Ensemble | | BNN | |
|---|---|---|---|---|---|---|---|
| | | Epistemic | Aleatoric | Epistemic | Aleatoric | Epistemic | Aleatoric |
| CIFAR | In Dist-Clean | 0.102 | 0.031 | 0.008 | 0.076 | 0.151 | 0.179 |
| | MNIST | 0.184 (1.801) | 0.142 (4.626) | 0.021 (2.723) | 0.044 (0.572) | 0.445 (2.95) | 0.772 (4.312) |
| | Fashion MNIST | 0.183 (1.791) | 0.147 (4.808) | 0.022 (2.822) | 0.042 (0.557) | 0.431 (2.861) | 0.811 (4.53) |
| | SVHN | 0.183 (1.789) | 0.141 (4.606) | 0.019 (2.421) | 0.046 (0.608) | 0.394 (2.612) | 0.647 (3.616) |
| | | Epistemic | Aleatoric | Epistemic | Aleatoric | Epistemic | Aleatoric |
| MNIST | In Dist-Clean | 0.188 | 0.057 | 0.012 | 0.063 | 0.198 | 0.071 |
| | CIFAR | 0.185 (0.986) | 0.167 (2.94) | 0.023 (1.948) | 0.042 (0.67) | 0.821 (4.155) | 0.372 (5.124) |
| | Fashion MNIST | 0.162 (0.866) | 0.187 (3.287) | 0.019 (1.618) | 0.035 (0.564) | 0.919 (4.654) | 0.461 (6.496) |
| | SVHN | 0.190 (1.015) | 0.168 (2.963) | 0.022 (1.847) | 0.043 (0.69) | 0.923 (4.674) | 0.297 (4.183) |
| | | Epistemic | Aleatoric | Epistemic | Aleatoric | Epistemic | Aleatoric |
| Fashion MNIST | In Dist-Clean | 0.121 | 0.029 | 0.007 | 0.075 | 0.197 | 0.351 |
| | CIFAR | 0.205 (1.705) | 0.104 (3.561) | 0.022 (3.178) | 0.050 (0.667) | 0.408 (2.073) | 0.221 (0.629) |
| | MNIST | 0.165 (1.367) | 0.182 (6.192) | 0.021 (3.069) | 0.033 (0.443) | 0.909 (4.618) | 0.328 (0.932) |
| | SVHN | 0.218 (1.81) | 0.109 (3.742) | 0.026 (3.827) | 0.046 (0.618) | 0.598 (3.036) | 0.305 (0.867) |
| | | Epistemic | Aleatoric | Epistemic | Aleatoric | Epistemic | Aleatoric |
| SVHN | In Dist-Clean | 0.061 | 0.01 | 0.004 | 0.083 | 0.097 | 0.107 |
| | CIFAR | 0.181 (2.992) | 0.121 (11.778) | 0.026 (6.029) | 0.040 (0.479) | 0.655 (6.765) | 0.716 (6.69) |
| | MNIST | 0.198 (3.272) | 0.063 (6.154) | 0.024 (5.495) | 0.056 (0.675) | 0.329 (3.393) | 0.336 (3.139) |
| | Fashion MNIST | 0.199 (3.288) | 0.113 (10.984) | 0.026 (6.002) | 0.041 (0.491) | 0.103 (6.375) | 0.125 (6.178) |

Table 4: Out-Of-Distribution evaluation. Across all datasets, when tested on datasets outside of the training dataset, CBDL demonstrated the most consistent behavior in terms of predictive uncertainty quantification. The quantity in the parenthesis is the magnitude of increase relative to the quantity when tested on the clean dataset.

| | | CBDL | | Ensemble | | BNN | |
|---|---|---|---|---|---|---|---|
| | | Epi. AUROC | Alea. AUROC | Epi. AUROC | Alea. AUROC | Epi. AUROC | Alea. AUROC |
| CIFAR | MNIST | **0.905** | 0.826 | 0.805 | 0.104 | 0.860 | **0.929** |
| | Fashion MNIST | **0.920** | 0.836 | 0.816 | 0.093 | 0.808 | **0.937** |
| | SVHN | **0.851** | 0.823 | 0.789 | 0.117 | 0.826 | **0.909** |
| | | Epi. AUROC | Alea. AUROC | Epi. AUROC | Alea. AUROC | Epi. AUROC | Alea. AUROC |
| MNIST | CIFAR | **0.930** | **0.912** | 0.827 | 0.040 | 0.835 | 0.882 |
| | Fashion MNIST | **0.956** | 0.935 | 0.804 | 0.030 | 0.929 | **0.960** |
| | SVHN | **0.971** | **0.926** | 0.808 | 0.033 | 0.874 | 0.898 |
| | | Epi. AUROC | Alea. AUROC | Epi. AUROC | Alea. AUROC | Epi. AUROC | Alea. AUROC |
| Fashion MNIST | CIFAR | **0.933** | 0.724 | 0.862 | 0.121 | 0.703 | **0.770** |
| | MNIST | 0.925 | 0.881 | 0.770 | 0.085 | **0.962** | **0.933** |
| | SVHN | **0.972** | 0.830 | 0.865 | 0.066 | 0.855 | **0.904** |
| | | Epi. AUROC | Alea. AUROC | Epi. AUROC | Alea. AUROC | Epi. AUROC | Alea. AUROC |
| SVHN | CIFAR | **0.979** | 0.941 | 0.890 | 0.014 | 0.964 | **0.952** |
| | MNIST | **0.781** | 0.679 | 0.777 | 0.205 | 0.728 | **0.709** |
| | Fashion MNIST | **0.967** | 0.881 | 0.899 | 0.030 | 0.921 | **0.917** |

Table 5: Out-Of-Distribution evaluation. AUROC for OOD detection is computed using both aleatoric and epistemic predictive uncertainty measures of the different approaches. Each model is trained on the training partition of the dataset and tested on the respective testing set of the OOD dataset. When using predictive AU, a single BNN performs relatively better than CBDL for OOD detection, though CBDL is still comparable and outperforms the single BNN when trained on Fashion-MNIST. When using predictive EU, CBDL outperforms the other approaches on most datasets.

under the curve are computed and averaged for a given noise severity level. Finally, this quantity is averaged across all severity levels. The results are shown in Table 3. As we can see, when EBNN outperforms CBDL in specific instances, the difference is minor, and when the opposite happens, the difference is significant. Over all datasets, CBDL outperforms a single BNN.

Figure 6: Accuracy vs Rejection Rate - CIFAR10C Gaussian blur. For lower severities (e.g. 1-2) of Gaussian blur noise, CBDL exhibits better performance and converges to 100% accuracy at a lower rejection rate. For higher severities (above 3), the single BNN seems to show better performance. Even so, the overall rejection rate is high to achieve such accuracy. The ensemble method (ENS) exhibits comparable performance in this specific instance.

### 4.1.2 Out-of-distribution Evaluation

In this final evaluation, we analyze the behavior of the three approaches (CBDL, EBNN, and BNN) when tested on an OOD dataset. For a given dataset, for example CIFAR, a model is trained on the dataset, and evaluated on the remaining datasets, i.e. MNIST, SVHN, and Fashion-MNIST. The relative increase in the average predictive EU and AU is computed compared to when an approach is tested on the In-Distribution testing set. The In-Distribution uncertainty quantities for each dataset are shown in Table 4. The relative magnitude of the increase in each quantity is shown in the parenthesis for each respective OOD dataset.

The baseline EBNN exhibits a counterintuitive behavior, viz. a decrease in predictive AU. Although this behavior is consistent across the different dataset combinations, this would be an undesirable quality for OOD detection scenarios.

The single BNN exhibits inconsistent behavior in terms of the relative increase or decrease in magnitude. For CIFAR-10, the predictive AU increases by a magnitude larger than the predictive EU, while for SVHN the increase is smaller. Moreover, for Fashion-MNIST the predictive AU decreases, and for MNIST the behavior is inconsistent.

Only our proposed CBDL method demonstrates a stable, consistent increase in predictive AU, which is greater in magnitude relative to the increase in predictive EU. These results demonstrate a clear improvement in performance by our proposed approach.

We also give results pertaining AUROCs for the OOD detection performance, presented in Table 5. Each group of results shows which dataset the models are trained on, and which they are tested on. We report AUROC when using both the predictive EU and AU of each approach. In general, the baseline EBNN method performs the worst in all cases. When using predictive AU, a single BNN performs relatively better than CBDL for OOD detection, though CBDL is still comparable and outperforms it when trained on Fashion-MNIST. When using predictive EU, CBDL clearly outperforms the other approaches on most datasets. This is in line with other works where predictive EU is considered important for OOD detection (Kendall & Gal, 2017).

We conjecture that this may be due to the fact that a single BNN is not able to gauge predictive EU properly. Hence, the single BNN flags an instance as OOD when it comes from the "tail" of the distribution; this is well-captured by (predictive) aleatoric uncertainty. On the contrary, CBDL is able to gauge predictive EU properly, and hence it is able to capture when OOD happens by looking at the "disagreement" between the elements of the predictive credal set, captured by the difference between upper and lower entropy.

## 4.2 Downstream Tasks Performance

As we have shown in the previous section, CBDL is better than single BNNs and ensemble of BNNs at quantifying and disentangling predictive AU and EU. In this section, we show with two applications – motion prediction in autonomous driving scenarios, and blood glucose and insulin dynamics for artificial pancreas control – that CBDL has better downstream tasks capability than EBNN. We do not compare CBDL against belief tracking techniques because these latter require extra assumptions that CBDL do not, as we further expand on in Appendix K.

### 4.2.1 Motion Prediction for Autonomous Racing

In this case study, we demonstrate the utility of CBDL for motion prediction in autonomous driving scenarios. An important challenge in autonomous driving is understanding the intent of other agents and predicting their future trajectories to allow for safety-aware planning. In autonomous racing, where control is pushed to the dynamical limits, accurate and robust predictions are even more essential for outperforming opponent agents while assuring safety. CBDL provides a straightforward method for quantifying uncertainty and deriving robust prediction regions for anticipating an agent's behavior.

We use the problem settings in Tumu et al. (2023) to define the problem of obtaining prediction sets for future positions of an autonomous racing agent. Our results show that the prediction regions have improved coverage when compared to EBNN. These results hold in both In-Distribution and Out-Of-Distribution settings, which are described below.

**Problem.** Let $O^i(t, l) \equiv O^i = \{\pi^i_{t-l}, \ldots, \pi^i_t\}$ denote the $i$-th trajectory instance of an agent at time $t$, consisting of the observed positions from time $t - l$ up to time $t$. Let then $C^i$ be a time-invariant context variable. Let also $F^i(t, h) \equiv F^i = \{\pi^i_{t+1}, \ldots, \pi^i_{t+h}\}$ be the collection of the next $h$ future positions. We wish to obtain a model $M$ that predicts region $\mathcal{R}_\alpha$ with probabilistic guarantees. In particular, for EBNN $\mathcal{R}_\alpha$ is the $\alpha$-level HDR $R_\alpha(P_{\text{ens}})$ of $P_{\text{ens}}$, so that $P_{\text{ens}}[F^i \in R_\alpha(P_{\text{ens}})] \geq 1 - \alpha$, while for CBDL $\mathcal{R}_\alpha = IR_\alpha(\hat{\mathcal{P}}_{\text{pred}})$, so that $\underline{\hat{P}}^{\text{pred}}[F^i \in IR_\alpha(\hat{\mathcal{P}}_{\text{pred}})] \geq 1 - \alpha$, or equivalently, $\hat{P}^{\text{pred}}[F^i \in IR_\alpha(\hat{\mathcal{P}}_{\text{pred}})] \geq 1 - \alpha$, for all $\hat{P}^{\text{pred}} \in \hat{\mathcal{P}}_{\text{pred}}$.

The dataset consists of instances of $(O^i, F^i)$ divided into a training set $D_{\text{train}}$ and a testing set $D_{\text{test}}$. We train an uncertainty-aware model on $D_{\text{train}}$ that computes the triplet $(F^i_l, F^i_m, F^i_u) = M(O^i, C^i)$ where $F^i_l$, $F^i_u$, $F^i_m$ are the lower, upper, and mean predictions of the future positions, respectively.

The dataset $D_{\text{all}}$ is created by collecting simulated trajectories of autonomous race cars in the F1Tenth-Gym (O'Kelly et al., 2020); for details, see Tumu et al. (2023). As shown in Figure 7, different racing lines were utilized including the center, right, left, and optimal racing line for the Spielberg track.

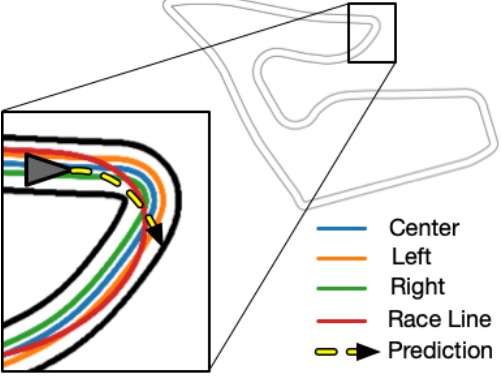

Figure 7: Motion Prediction for F1Tenth-Gym Environment (O'Kelly et al., 2020). Data is collected by simulating various racing lines on the Spielberg Track.

| In-Distribution Results | | | | | | |
| --- | --- | --- | --- | --- | --- | --- |
| | Ensemble | | | CBDL | | |
| $1 - \alpha$ | 0.9 | 0.95 | 0.99 | 0.9 | 0.95 | 0.99 |
| One-step | 0.962 | 0.980 | 0.992 | **0.992** | **0.995** | **0.997** |
| Multi-step | 0.638 | 0.826 | 0.937 | **0.914** | **0.948** | **0.979** |
| Out-of-Distribution Results | | | | | | |
| | Ensemble | | | CBDL | | |
| $1 - \alpha$ | 0.9 | 0.95 | 0.99 | 0.9 | 0.95 | 0.99 |
| One-step | 0.919 | 0.950 | 0.980 | **0.979** | **0.988** | **0.995** |
| Multi-step | 0.532 | 0.703 | 0.860 | **0.825** | **0.884** | **0.943** |

Table 6: F1Tenth coverage results. We report one-step coverage and multi-step coverage across 3 different values of $\alpha$. CBDL exceed coverage of EBNNs in all settings.

We denote these by $D_{\text{center}}$, $D_{\text{right}}$, $D_{\text{left}}$, and $D_{\text{race}}$, respectively. Position $\pi$ is a vector $\pi = (a, b, \vartheta, v)^{\top}$, where $a$ and $b$ are coordinates in a 2-dimensional Euclidean space, and $\vartheta$ and $v$ are the heading and speed, respectively. In total, the $D_{\text{all}}$ consists of 34686 train instances, 4336 validation instances, and 4336 test instances.

**In-Distribution vs Out-Of-Distribution.** We consider the prediction task to be In-Distribution when $D_{\text{train}}, D_{\text{test}} \subset D_{\text{all}}$. It is Out-Of-Distribution (OOD) when $D_{\text{train}} \subset D_{\text{center}} \cup D_{\text{right}} \cup D_{\text{left}}$ and $D_{\text{test}} \subset D_{\text{race}}$.

**Metrics.** We train the ensemble of BNNs and the CBDL models, $M_{\text{ens}}$ and $M_{\text{CBDL}}$ respectively, using the same architecture and different seeds. As for section 4.1, for CBDL this corresponds to having a non-singleton prior FGCS, and a singleton likelihood FGCS. We compare the performance with respect to the test set by computing the single-step coverage, where each prediction time-step is treated independently, and the multi-step coverage, which considers the entire $h$-step prediction.

Figure 8.(a) depicts a sample of the In-Distribution evaluation for each of the models. For a given trajectory, the red boxes indicate when the prediction region did not cover the actual trajectory at that time-step. Qualitatively, $M_{\text{CBDL}}$ has less missed time steps when compared to $M_{\text{ens}}$. Table 6 shows that CBDL performs better in terms of both one-step and multi-step coverage. Similar results can be observed for the OOD scenario. There, all models were trained on racing lines which are predominantly parallel to the track curvature. As a consequence, when the test set consists of instances with higher curvatures, the overall coverage of all models degrades. This can be seen in Figure 8.(b), where the prediction of the models (orange) tends to be straight while the actual trajectory is more curved (green). Despite this, the figure and the coverage metrics in Table 6 show how CBDL exhibits a more robust behavior.

### 4.2.2 Artificial Pancreas Control

**Overall Setup.** In this next case study we consider the problem of data-driven control of human blood glucose-insulin dynamics, using an artificial pancreas system, see Figure 9.

External insulin delivery is accomplished by using an insulin pump controlled by the artificial pancreas software, which attempts to regulate the blood-glucose (BG) level of the patient within the euglycemic range of $[70, 180]$ mg/dl (Kushner et al., 2018). Levels below 70 mg/dl lead to hypoglycemia, which can lead to loss of consciousness, coma or even death. On the other hand, levels above 300 mg/dl lead to a condition called ketoacidosis, where the body can break down fat due to lack of insulin, and lead to build up of ketones. In order to treat this situation, patients receive external insulin delivery through insulin pumps. Artificial Pancreas (AP) systems can remedy this situation by measuring the blood glucose level,

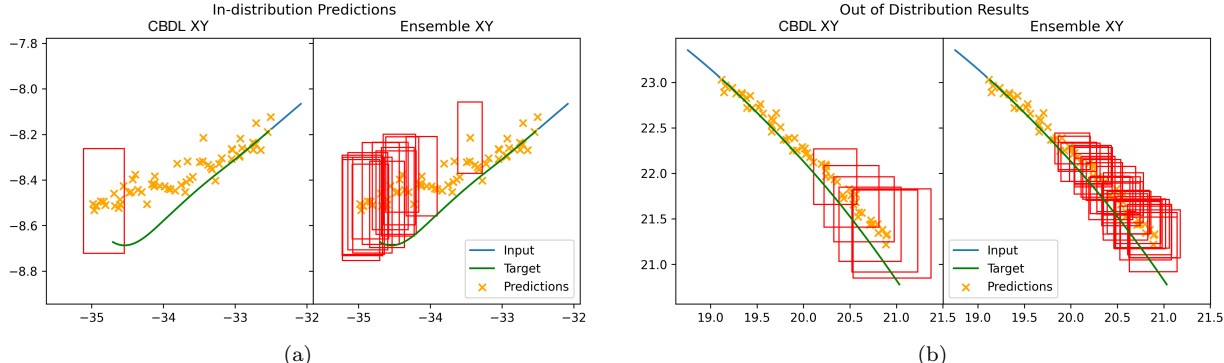

(a)  (b)

Figure 8: In both pictures, the red boxes indicate when the prediction region did not cover the actual trajectory at that time-step. **Left:** F1Tenth In-Distribution results. Given an input of past observations, CBDL exhibits better coverage of the future target trajectory. Predictions which do not cover the target within the desired $1-\alpha$ level are indicated in red. **Right:** F1Tenth Out-Of-Distribution (OOD) results. Robust performance is exhibited by CBDL when compared to EBNN in OOD settings.

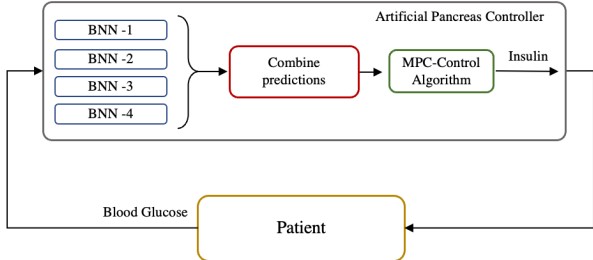

Figure 9: The Bayesian Neural Networks predict a future blood glucose value. These individual predictions are combined to get a robust estimate of the true value as an interval. This is used by the Model Predictive Control (MPC) algorithm to recommend insulin dosage for the patient. The patient block in our experiment is simulated using the virtual patient models from the UVa-Padova simulator.

and automatically injecting insulin into the blood stream. Thus, we define the *unsafe regions* of the space as $G(t) \in (-\infty, 70) \cup (300, \infty)$, where $G(t)$ is the BG value at time $t$. This is the shaded region in Figure 10.

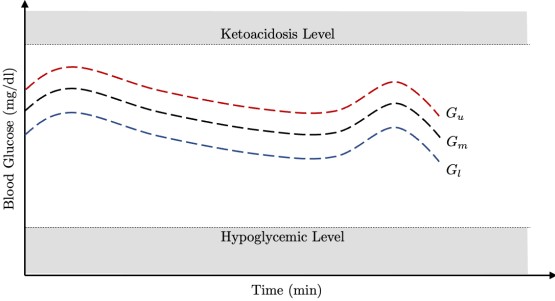

Figure 10: Starting from an initial glucose value, the task of the artificial pancreas controller is to maintain the blood glucose value within safe operating limits using insulin as a mode of control.

**Neural Network Models and Controller.** Deep Neural Networks are effective in capturing the BG-insulin dynamics for personalized medical devices (Kushner et al., 2018). This allows for improved device performance. Even though standard Feedforward Neural Networks can be used, Bayesian Neural Networks

(BNNs), and especially a collection of multiple BNNs, offer a better alternative towards uncertainty aware predictions. Here, we test the ramifications of these prediction sets, when used inside an online receding horizon control scheme for insulin delivery. We use the standard MPC control scheme for this purpose, well-known in the literature (Dutta et al., 2018). More formally, let $G(t)$ and $I(t) \equiv I_t$ be the blood-glucose and insulin values at time $t$, respectively. We denote the finite length trajectory of length $H$ as $\overleftarrow{G}_H(t) := [G(t - H + 1), \dots, G(t)]$, and $\overleftarrow{I}_H(t) := [I(t - H + 1), \dots, I(t)]$. An uncertainty aware model $M$ computes the triplet $(G_l(t+l), G_m(t+l), G_u(t+l)) = M(\overleftarrow{G}_H(t), \overleftarrow{I}_H(t))$, where $G_m$ is the mean prediction output, and $G_l, G_u$ are the lower and upper predictions of the glucose value, respectively. By design, it is true that $G_l \leq G_m \leq G_u$. A model predictive control algorithm – whose cost function we denote by $J$ – solves $\arg\min_{I_0, I_1, \dots, I_{k-1}} \sum_{i=0}^{k-1} J(M(\overleftarrow{G}_H(t+i), \overleftarrow{I}_H(t+i)))$.

After every time step, the control algorithm picks the first insulin input $I_0$ as the insulin bolus for the patient, and discards the rest. Cost function $J$ takes into account three factors, (i) Distance of the mean prediction level $G_m$ at each time step from a target value of 120 mg/dl, (ii) Distance of upper and lower predictions ($G_u$ and $G_l$) from the unsafe regions of the state space $G(t) > 300$ and $G(t) < 70$, and (iii) Total insulin injected $\sum_{t=0}^{k-1} I_t$.

Starting with some initial glucose value $G(0)$, we measure the performance of the artificial pancreas controller as the fraction of time it spends in the unsafe regions,

$$t_{\text{unsafe}} = \frac{1}{T} \sum_{t=1}^{T} \mathbb{1}\left\{G(t) \in (-\infty, 70) \cup (300, \infty)\right\},$$

where $\mathbb{1}\{\cdot\}$ denotes the indicator function. A lower value is more desirable. We compare EBNN and CBDL as different realizations of the model $M$.

**Distribution Shift using Meals.** A well known problem with learned models is distribution shift. Bayesian Neural Networks can address this issue by apprising the end user of the increased uncertainty. For regression models of the type described above, this appears as larger prediction intervals $[G_l, G_u]$. The artificial pancreas controller can run into this situation in the following way: the insulin-glucose time series data collected for training the data-driven model $M$ can be without meals, while at test time the patient can have meals. This creates a distribution shift between the training and test time data. Fortunately, the UVa-Padova simulator (Dalla Man et al., 2013) allows us to create datasets with and without meal inputs. In this case study, the training data was obtained by randomly initializing the BG value in the range $[120, 190]$, and simulating the patient for 720 minutes. The controller was executed at 5 minutes intervals. At test time the patient was supplied meals at specific time intervals (for details, see Appendix J). This creates a significant distribution shift since meals are effectively an unknown variable which can affect the system state. However, from the controller's perspective this is practical, since patients can have unannounced meals.

**Results and Discussion.** To capture the difference in performance between EBNN and CBDL, we compute $\text{Perf}_{\text{diff}} := (t_{\text{unsafe}}^{\text{EBNN}} - t_{\text{unsafe}}^{\text{CBDL}})/t_{\text{unsafe}}^{\text{EBNN}}$. Both $t_{\text{unsafe}}^{\text{EBNN}}$ and $t_{\text{unsafe}}^{\text{CBDL}}$ depend on interval $[G_l, G_u]$; for EBNN, this latter corresponds to the $\alpha$-level HDR $R_\alpha(P_{\text{ens}})$ associated with EBNN distribution $P_{\text{ens}}$, while for CBDL it corresponds to the IHDR $IR_\alpha(\hat{\mathcal{P}}_{\text{pred}})$. We consider one case in which CBDL is trained using a credal prior set and only one likelihood (we choose different seeds which initialize the prior distributions but we keep the same architecture for the BNNs), and another case in which we do the opposite (we use the same seed and different architectures).

We report $\text{Perf}_{\text{diff}}$, across different choices in Table 7. We observe that for lower values of $\alpha$ (or equivalently, for larger values of $1 - \alpha$, e.g. 0.95 and 0.99) the gains of CBDL are more pronounced. This means that when larger significance levels need to be ensured, CBDL is to be preferred to ensemble of BNNs. As discussed before, the CBDL procedure considers all the infinitely many possible priors that can be expressed as a convex combination of the priors that the user specifies at the beginning of the analysis. The same holds for the likelihoods. While this results in a more conservative estimate as compared to the EBNN framework, CBDL produces controllers which respect the safety limits better. To see that CBDL is more conservative than EBNN, notice that when combining predictive distributions from multiple BNNs, CBDL combines the predictions via a finitely generated credal set (FGCS) whose extrema are the individual distributions. On

the contrary, an EBNN takes an average of the individual distributions to compute the ensemble distribution $P_{\text{ens}}$. The union of the HDRs of the predictive distributions is more conservative (i.e., broader) than the HDR of the single ensemble distribution. While the approach by EBNN seems like a reasonable choice on the surface, it falls short in capturing the uncertainty necessary for the downstream task. For more details on this case study, see Appendix J.

| $1 - \alpha$ | 0.9 | 0.95 | 0.99 |
|---|---|---|---|
| Varying Seeds | -5.8% | **5.5**% | **0.6**% |
| Varying Architectures | -6.6% | **0.9**% | **0.3**% |

Table 7: We report the performance improvements when using IBNNs as compared to EBNNs across 3 different values of $\alpha$. Row 1 corresponds to the case where the individual BNNs are trained with different seeds for the prior distribution; and Row 2 is the case when the BNNs have different architectures.

## 5 Related Work

In Corani et al. (2012), the authors introduce credal classifiers (CCs) as a generalization of classifiers based on Bayesian networks. Unlike CCs, CBDL does not require independence assumptions between non-descendant, non-parent variables. In addition, CBDL avoids NP-hard complexity issues of searching for optimal structure in the space of Bayesian networks (Chickering et al., 2004). In Manchingal & Cuzzolin (2022), an epistemic convolutional neural network (ECNN) is developed that explicitly models the epistemic uncertainty induced by training data of limited size and quality. A clear distinction is that ECNNs measure uncertainty in target-level representations whereas CBDL identifies the uncertainty measure on the output space $\mathcal{Y}$. Despite the merit of their work, we believe CBDL achieves greater generality, since it is able to quantify both aleatoric and epistemic predictive uncertainties, and is applicable to problems beyond classification. For a review of the state of the art concerning the distinction between EU and AU we refer to Hüllermeier & Waegeman (2021) and to Manchingal & Cuzzolin (2022). We also point out how CBDL has been recently used to solve prior-likelihood conflicts in Bayesian statistics (Marquardt et al., 2023). Further references can be found in Appendix L.

We also point out how there exist other efficient methods which perform approximate Variational Inference via dropouts in deep neural networks (Kendall & Gal, 2017; Gal & Ghahramani, 2016). As mentioned at the end of section 3.1, we can easily adapt CBDL to use such dropout approximations in **Steps 3-4** of Algorithm 1. Since our contribution is centered around how different predictions can be combined via an FGCS, we used the de-facto standard for performing inference on BNNs, which is based on off-the-shelf VI techniques. In the future, we plan to study the effect on computational complexity and uncertainty quantification capability of a CBDL procedure that approximates posterior and predictive distributions via dropout.

We do not consider Bayesian Model Averaging (BMA) as a baseline for CBDL for two main reasons. First, BMA applied to deep learning needs to implement full batch Hamiltonian Monte Carlo in order to get to the true posterior (Izmailov et al., 2021b). Given the number of parameters in modern deep learning architectures – in the order of millions – this is realistically possible at an experimental level only to labs with access to industry scale computational resources. In order to be practically relevant, we limit our experiments to the more well-understood realm of Variational Inference on BNNs. In addition, Izmailov et al. (2021a) show the pitfalls of BMA in the context of Bayesian Neural Networks, a further reason not to use a Highest Density Region resulting from BMA as a baseline for CBDL. Such pitfalls are related to the fact that BMA can be seen as a model featuring second-order distributions, i.e. distributions over distributions. In particular, the distribution $Q$ in equation 1 is a second order distribution. These types of models have been recently shown to suffer from major pitfalls when used to quantify predictive EU due to their sensitivity to regularization parameters, and to underestimate predictive AU (Bengs et al., 2022; Pandey & Yu, 2023; Juergens et al., 2024).

## 6    Conclusion

We presented CBDL, a procedure that can be seen as a non-condensed, uncountably infinite ensemble of BNNs, carried out using only finitely many elements. It allows to distinguish between predictive AU and EU, and to quantify them. We showed how it can be used to specify a set of outputs – the IHDR – that enjoys probabilistic guarantees. We showed empirically that it improves on the Bayesian state of the art at gauging predictive AU and EU, and we also demonstrated its downstream tasks capabilities.

We point out how a region that improves on $IR_\alpha(\hat{\mathcal{P}}_{\text{pred}})$, meaning that it would be tighter, is

$$IR'_\alpha(\hat{\mathcal{P}}_{\text{pred}}) = \{y \in \mathcal{Y} : \underline{\hat{p}}^{\text{pred}}(y) \geq \underline{\hat{p}}^\alpha\},$$

where $\underline{\hat{p}}^{\text{pred}} := \min_{k,s} \hat{p}^{\text{pred}}_{k,s}$, and $\underline{\hat{p}}^\alpha$ is the largest constant such that $\underline{\hat{P}}^{\text{pred}}[y \in IR'_\alpha(\hat{\mathcal{P}}_{\text{pred}})] \geq 1 - \alpha$. The problem with $IR'_\alpha$ is that, while the highest density regions $R_\alpha(\hat{P}^{\text{pred}}_{k,s})$ associated with the predictive distributions in $\text{ex}\hat{\mathcal{P}}_{\text{pred}}$ can be computed using off-the-shelf tools, calculating $\underline{\hat{p}}^{\text{pred}}$ and $\underline{\hat{p}}^\alpha$ would have been much more computationally expensive. In addition, it would have required to come up with a new technique to find $\underline{\hat{p}}^{\text{pred}}$ and $\underline{\hat{p}}^\alpha$. We defer studying this to future work.

We also plan to apply CBDL to continual learning to overcome the curse of dimensionality and to capture an agent's preference over the tasks to perform, similarly to Lu et al. (2023), and to active learning, to be able to sample from the regions of the state space exhibiting the highest epistemic uncertainty, similarly to Dutta et al. (2023).

Furthermore, we intend to relate CBDL to Bayesian Model Selection (BMS) (Ghosh et al., 2019). This latter suffers from the same problem as "regular" Bayesian inference. That is, while it tries to come up with a sophisticate prior that induces shrinkage, it still relies on the "correctness" of that prior, i.e. on correctly specifying the prior's parameters. In the future, an interesting way of combining CBDL with BMS will be to use a finite number of regularized horseshoe priors, as suggested by Ghosh et al. (2019, Section 3.2), as extreme elements of the prior credal set.

We also call attention to the fact that CBDL is a model-based approach. The relationship with model-free approaches such as conformal prediction (Shafer & Vovk, 2008) will be the object of future studies. In particular, we are interested in finding in which cases IHDRs are narrower than conformal regions, and vice versa, and which credal sets give rise to IHDRs enjoying the same probabilistic guarantees as conformal regions.

Finally, we point out how one possible way of easing the burden of the combinatorial task in **Step 3** of Algorithm 1 is to specify a prior credal set whose size strikes the perfect balance between being "vague enough" so that we do not underestimate the EU, and being "small enough" so that CBDL is actually implementable. We suspect conjugacy of the priors may play a key role in this endeavor. Because of its centrality, we defer the study of "optimal prior credal sets" to future work. We also point out how an evidential approach (Amini et al., 2020; Charpentier et al., 2020; Denœux, 2022; 2023; Sensoy et al., 2018) – at least in classification problems – could allows us to bypass the bottleneck in Algorithm 1 by merging an imprecise probabilistic approach, with some tricks resulting from smartly choosing our priors.

A CBDL-adjacent research question of great interest pertaining ensemble learning, then, is how do single components of an ensemble contribute to the quantification of the "global predictive EU" faced by the agent. When the uncertainty captured by an ensemble is distilled into only one distribution, there could indeed be "uncertainty spills", like when one pours water in a glass too hastily, and some finishes on the table instead of in the glass. This remains an unexplored venture in the uncertainty quantification community.

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

## A  Why do we need IPs?

The main motivations for working with credal sets are two. Let $(\Omega, \mathcal{F})$ be the measurable space of interest.

(i) A single probability distribution does not suffice to represent ignorance in the sense of lack of knowledge; this is well documented in the literature, see e.g. Hüllermeier & Waegeman (2021) and references therein. Consider the example of complete ignorance (CI) in the case of a finite state space $\Omega$ (Hüllermeier & Waegeman, 2021, Section 3.3). In standard Bayesian analysis, CI is modeled in terms of the uniform distribution $\text{Unif}(\Omega)$; this is justified by Laplace's "principle of indifference". Then, however, it is not possible to distinguish between precise probabilistic knowledge about a random event – called *prior indifference*; think of the tossing of a fair coin – and a complete lack of knowledge due to an incomplete description of the experiment – called *prior ignorance*. Another problem is given by the additive nature of probability distributions. Consider again the example of a uniform distribution. First, let us observe that it is not invariant under reparametrization. In addition, if we model the ignorance about the length $x$ of the side of a cube in $\mathbb{R}^3$ via a uniform measure on the interval $[l, u] \subset \mathbb{R}$, then this does not yield a uniform distribution of $x^3$ on $[l^3, u^3]$, which suggests some degree of informedness about the cube's volume. Finally, as pointed out in Walley (1991), if we ask a subject – even an expert – about their opinion regarding some events, it is much more likely that they will report interval of probabilities rather than single values.

(ii) Working with credal sets allows to achieve prior and likelihood *robustness*: realistically large sets $\mathcal{P}_{\text{prior}}$ of priors and $\mathcal{P}_{\text{lik}}$ of likelihoods are elicited. Using credal sets, the agent recognizes that prior beliefs and knowledge about the sampling model are limited and imprecise. Combining each pair of functions in $\mathcal{P}_{\text{prior}}$ and $\mathcal{P}_{\text{lik}}$ using Bayes' rule, a class of posterior distributions – reflecting the updated state of uncertainty – is formed. If the available information is not sufficient to identify a unique posterior distribution, or a set of posteriors whose diameter is small, credal sets allow to represent *indecision*, thus leading to a less informative but more robust conclusions.[20]

## B  On the use of credal sets

Let us address a critique raised against the use of credal sets. Lassiter (2020) argues against the use of sets of probabilities to model an agent's prior beliefs and their knowledge of the sampling model, while debating in favor of using hierarchical Bayesian models. As reported in Hüllermeier & Waegeman (2021, Secton 4.6.2), the argument against credal sets that is more cogent for the machine learning literature is that modeling a lack of knowledge in a set-based manner may hamper the possibility of inductive inference, up to a point where learning from empirical data is not possible any more. With this, we mean the following. As Pericchi (1998) points out, the natural candidate for a class of priors to represent complete ignorance is the class $\mathcal{P}_{\text{all}}$ of all distributions. When this class leads to non-vacuous and useful conclusions, these are quite compelling and uncontroversial. It turns out that the posterior probabilities obtained from this class are vacuous, that is, their lower and upper bounds are 0 and 1: no finite sample is enough to annihilate a sufficiently extreme prior belief. There is then a compromise to be made, and this is the compromise of *near-ignorance*. The near-ignorance class should be vacuous a priori in some respects, typically the ones that are the most important for the analysis at hand. This way of proceeding is labeled as arbitrary by Lassiter (2020), who instead advocates for the use of hierarchical Bayesian procedures. We find this critique not compelling, as during the analysis the job of the agent is to model reality: as pointed out in Hüllermeier & Waegeman (2021, Secton 5), statistical inference is not possible without underlying assumptions, and conclusions drawn from data are always conditional on those assumptions. If we were to work every time with the maximum level of generality, we would hardly be able to reach any conclusions. For example, in a statistical analysis we never consider the state $\Omega$ of *apparently possible states* (Walley, 1991, section 2.1.2), that is, the one that contains all the states $\omega$ that are logically consistent with the available information. If we consider a coin toss, we let the state space be $\Omega = \{\text{heads, tails}\}$, certainly not $\Omega = \{\text{heads, tails, coin landing on its edge, coin braking into pieces on landing, coin disappearing down a crack in the floor}\}$. The same holds for sets

---

[20]Here "diameter" has to be understood as the distance between upper and lower probability of event $A$, for all $A \in \mathcal{F}$.

of probabilities: it makes much more sense to work with near-ignorance credal sets than to work with $\mathcal{P}_{\text{all}}$. A final reason to rebut the point in Lassiter (2020) is that the problems indicated in (i) in section A that make the use of the uniform prior distribution – often interpreted as representing epistemic uncertainty in standard Bayesian inference – at least debatable are inherited by hierarchical Bayesian modeling, as specified in Bernardo (1979); Hüllermeier & Waegeman (2021); Jeffreys (1946). Furthermore, a single distribution that is the result of a hierarchical Bayesian procedure is unable to gauge epistemic uncertainty (Hüllermeier & Waegeman, 2021).

### B.1 Further notes on credal sets

As pointed out in Corani et al. (2012, Section 3.3), there is a way of obtaining credal sets starting from sets of probability intervals; in addition, standard algorithms can compute the extreme elements of a credal set for which a probability interval has been provided (Avis & Fukuda, 1996). However, the resulting number of extrema is exponential in the size of the possibility space (Tessem, 1992).[21] For this reason we prefer to specify prior and likelihood finitely generated credal sets instead.

The way credal sets behave after conditioning on new available evidence has been recently studied in Caprio & Seidenfeld (2023), and a way of using credal sets in the open-world scenario (that is, when the support of the elements of the credal set can become larger as more data become available) is explored in Caprio & Mukherjee (2023d). Furthermore, the use of credal sets in statistical learning theory has been studied in Caprio et al. (2024c), and in computer vision in Caprio (2024).

## C   A further IP concept: the core

Let again $(\Omega, \mathcal{F})$ be the measurable space of interest. Because of the conjugacy property of upper and lower probabilities, let us focus on upper probabilities only. We say that upper probability $\overline{P}$ is *concave* if $\overline{P}(A \cup B) \leq \overline{P}(A) + \overline{P}(B) - \overline{P}(A \cap B)$, for all $A, B \in \mathcal{F}$. Recall that $\Delta(\Omega, \mathcal{F})$ denotes the set of all probability measures on $(\Omega, \mathcal{F})$. Upper probability $\overline{P}$ is *compatible* with the convex set (Gong & Meng, 2021)

$$\text{core}(\overline{P}) := \{P \in \Delta(\Omega, \mathcal{F}) : P(A) \leq \overline{P}(A), \forall A \in \mathcal{F}\}$$
$$= \{P \in \Delta(\Omega, \mathcal{F}) : \underline{P}(A) \leq P(A) \leq \overline{P}(A), \forall A \in \mathcal{F}\}$$

where the second equality is a characterization (Cerreia-Vioglio et al., 2015, Page 3389). Notice that the core is convex (Marinacci & Montrucchio, 2004, Section 2.2). We assume it is nonempty. Then, it is weak$^\star$-compact as a result of Marinacci & Montrucchio (2004, Proposition 3).[22]

Since the core is convex, the set $\text{ex}[\text{core}(\overline{P})]$ of extreme points of the core is well defined. It contains all the elements of the core that cannot be written as a convex combination of one another. The following important result is a consequence of Walley (1991, Theorem 3.6.2).

**Theorem 9.** *Suppose core$(\overline{P})$ is nonempty. Then, the following holds.*

(a) *$ex[core(\overline{P})] \neq \emptyset$.*

(b) *core$(\overline{P})$ is the closure in the weak$^\star$ topology of the convex hull of $ex[core(\overline{P})]$.*

(c) *If $\overline{P}(A) = \sup_{P \in core(\overline{P})} P(A)$, for all $A \in \mathcal{F}$, then $\overline{P}(A) = \sup_{P \in ex[core(\overline{P})]} P(A)$, for all $A \in \mathcal{F}$.*

So in order to define an upper probability $\overline{P}$ that setwise dominates the elements of core$(\overline{P})$ it is enough to specify the extreme points of the core.

---

[21]Recall that the possibility space of a random variable is the space of the values it can take on.

[22]Recall that in the weak$^\star$ topology, a net $(P_\alpha)_{\alpha \in I}$ converges to $P$ if and only if $P_\alpha(A) \to P(A)$, for all $A \in \mathcal{F}$.

## D  A new Bayes' theorem for IPs

We present Theorem 10, a result that – although appealing – does not lend itself well to be applied to the CBDL procedure. An extension of Theorem 10 is given in Caprio et al. (2024b).

Call $\Theta$ the parameter space of interest and assume it is Polish, that is, the topology for $\Theta$ is complete, separable, and metrizable. This ensures that the set $\Delta(\Theta, \mathcal{B})$ of probability measures on $\Theta$ is Polish as well, where $\mathcal{B}$ denotes the Borel $\sigma$-algebra for $\Theta$. Let $\mathscr{X}$ be the set of all bounded, non-negative, $\mathcal{B}$-measurable functionals on $\Theta$. Call $\mathscr{D} = \mathcal{X} \times \mathcal{Y}$ the sample space endowed with the product $\sigma$-algebra $\mathcal{A} = \mathcal{A}_{\mathbf{x}} \times \mathcal{A}_{\mathbf{y}}$, where $\mathcal{A}_{\mathbf{x}}$ is the $\sigma$-algebra endowed to $\mathcal{X}$ and $\mathcal{A}_{\mathbf{y}}$ is the $\sigma$-algebra endowed to $\mathcal{Y}$. Let the agent elicit $\mathcal{L}_\theta := \{P_\theta \in \Delta(\mathscr{D}, \mathcal{A}) : \theta \in \Theta\}$. Assume that each $P_\theta \in \mathcal{L}_\theta$ has density $L(\theta) = p(D \mid \theta)$ with respect to some $\sigma$-finite dominating measure $\nu$ on $(\mathscr{D}, \mathcal{A})$; this represents the likelihood function for $\theta$ having observed data $D \subset \mathscr{D}$. We assume for now that $L \in \mathscr{X}$, for all $D \subset \mathscr{D}$.

Let the agent specify a set $\mathcal{P}$ of probabilities on $(\Theta, \mathcal{B})$. Then, compute $\overline{P}$, and consider $\mathcal{P}^{\mathrm{co}} := \mathrm{core}(\overline{P})$; it represents the agent's initial beliefs.[23] We assume that every $P \in \mathcal{P}^{\mathrm{co}}$ has density $p$ with respect to some $\sigma$-finite dominating measure $\mu$ on $(\Theta, \mathcal{B})$, that is, $p = \frac{\mathrm{d}P}{\mathrm{d}\mu}$. We require the agent's beliefs to be represented by the core for two main reasons. The first, mathematical, one is to ensure that the belief set is compatible with the upper probability. The second, philosophical, one is the following (Caprio & Gong, 2023; Caprio & Mukherjee, 2023c). A criticism brought forward by Walley (1991, Section 2.10.4.(c)) is that, given an upper probability $\overline{P}$, there is no cogent reason for which the agent should choose a specific $P_T$ that is dominated by $\overline{P}$, or – for that matter – a collection of "plausible" probabilities. Because the core considers all (countably additive) probability measures that are dominated by $\overline{P}$, it is the perfect instrument to reconcile Walley's behavioral and sensitivity analysis interpretations.

Let the agent compute $\overline{P}_\theta$, and consider $\mathcal{L}_\theta^{\mathrm{co}} := \mathrm{core}(\overline{P}_\theta)$; it represents the set of plausible likelihoods. Let

$$\mathscr{L} := \left\{ L = \frac{\mathrm{d}P_\theta}{\mathrm{d}\nu}, \, P_\theta \in \mathcal{L}_\theta^{\mathrm{co}} \right\}, \tag{4}$$

and denote by $\overline{L}(\theta) := \sup_{L \in \mathscr{L}} L(\theta)$ and by $\underline{L}(\theta) := \inf_{L \in \mathscr{L}} L(\theta)$, for all $\theta \in \Theta$. Call

$$\mathcal{P}_D^{\mathrm{co}} := \left\{ P_D \in \Delta(\Theta, \mathcal{B}) : \frac{\mathrm{d}P_D}{\mathrm{d}\mu} = p(\theta \mid D) = \frac{L(\theta)p(\theta)}{\int_\Theta L(\theta)p(\theta)\mathrm{d}\theta}, \right.$$
$$\left. p = \frac{\mathrm{d}P}{\mathrm{d}\mu}, \, P \in \mathcal{P}^{\mathrm{co}}, \, L = \frac{\mathrm{d}P_\theta}{\mathrm{d}\nu}, \, P_\theta \in \mathcal{L}_\theta^{\mathrm{co}} \right\}$$

the class of posterior probabilities when the prior is in $\mathcal{P}^{\mathrm{co}}$ and the likelihood is in $\mathcal{L}_\theta^{\mathrm{co}}$, and let $\overline{P}_D(A) = \sup_{P_D \in \mathcal{P}_D^{\mathrm{co}}} P_D(A)$, for all $A \in \mathcal{B}$. Then, the following is a generalization of Bayes' theorem in Wasserman & Kadane (1990).

**Theorem 10.** *Suppose $\mathcal{P}^{co}, \mathcal{L}_\theta^{co}$ are nonempty. Then for all $A \in \mathcal{B}$,*

$$\overline{P}_D(A) \leq \frac{\sup_{P \in \mathcal{P}^{co}} \int_\Theta \overline{L}(\theta) \mathbb{1}_A(\theta) P(d\theta)}{\mathbf{c}}, \tag{5}$$

*provided that the ratio is well defined. Here, $\mathbf{c} := \sup_{P \in \mathcal{P}^{co}} \int_\Theta \overline{L}(\theta) \mathbb{1}_A(\theta) P(d\theta) + \inf_{P \in \mathcal{P}^{co}} \int_\Theta \underline{L}(\theta) \mathbb{1}_{A^c}(\theta) P(d\theta)$, and $\mathbb{1}_A$ denotes the indicator function for $A \in \mathcal{B}$. In addition, if $\overline{P}$ is concave, then the inequality in equation 5 is an equality for all $A \in \mathcal{B}$.*

This result is particularly appealing because, given some assumptions, it allows to perform a (generalized) Bayesian update of a prior upper probability (PUP) by carrying out only one operation, even when the likelihood is ill specified so that a set of likelihoods is needed. We also have the following.

**Lemma 11.** *Suppose $\mathcal{P}^{co}, \mathcal{L}_\theta^{co}$ are nonempty. Then, if $\overline{P}$ is concave, we have that $\overline{P}_D$ is concave as well.*

---

[23]Superscript "co" stands for convex and core.

This lemma is important because it tells us that the generalized Bayesian update of Theorem 10 preserves concavity, and so it can be applied to successive iterations. If at time $t$ the PUP is concave, then the PUP at time $t + 1$ – that is, the posterior upper probability at time $t$ – will be concave too. Necessary and sufficient conditions for a generic upper probability to be concave are given in Marinacci & Montrucchio (2004, Section 5).

In the future, these results can be generalized to the case in which the elements of $\mathscr{X}$ are unbounded using techniques in Troffaes & de Cooman (2014), and to the case in which the elements of $\mathscr{X}$ are $\mathbb{R}^d$-valued, for some $d \in \mathbb{N}$, since we never used specific properties of $\mathbb{R}$ in our proofs.

Despite being attractive, the generalized Bayesian update of Theorem 10 hinges upon three assumptions, namely that $\mathcal{P}^{\mathrm{co}}$ and $\mathcal{L}_\theta^{\mathrm{co}}$ are both cores of an upper probability, that they are nonempty, and that the prior upper probability $\overline{P}$ is concave. As the proverb goes, there is no free lunch. Having to check these assumptions, together with computing a supremum, an infimum, and the integrals in equation 5, makes Theorem 10 inadequate to be applied in the context of CBDL.

# E  Bounds on upper and lower entropy

In this section we find an upper bound for $\overline{H}(P)$ and a lower bound for $\underline{H}(P)$ that are extremely interesting. We first need to introduce three new concepts.

**Definition 12.** *Consider a set $\mathscr{P}$ of probabilities on a generic measurable space $(\Omega, \mathcal{F})$. We say that lower probability $\underline{P}$ is convex if $\underline{P}(A \cup B) \geq \underline{P}(A) + \underline{P}(B) - \underline{P}(A \cap B)$, for all $A, B \in \mathcal{F}$.*

*Then, let $\mathsf{P}$ be either an upper or a lower probability, and consider a generic bounded measurable function $f$ on $(\Omega, \mathcal{F})$, that is, $f \in B(\Omega)$. We define the Choquet integral of $f$ with respect to $\mathsf{P}$ as follows*

$$\int_\Omega f(\omega) \mathsf{P}(d\omega) := \int_0^\infty \mathsf{P}\left(\{\omega \in \Omega : f(\omega) \geq t\}\right) dt + \int_{-\infty}^0 \left[\mathsf{P}\left(\{\omega \in \Omega : f(\omega) \geq t\}\right) - \mathsf{P}(\Omega)\right] dt,$$

*where the right hand side integrals are (improper) Riemann integrals. If $\mathsf{P}$ is additive, then the Choquet integral reduces to the standard additive integral.*

*Finally, if $\Omega$ is uncountable, define for all $\omega \in \Omega$*

$$\underline{\pi}(\omega) := \inf_{P \in \mathscr{P}} \frac{dP}{d\mu}(\omega) \quad and \quad \overline{\pi}(\omega) := \sup_{P \in \mathscr{P}} \frac{dP}{d\mu}(\omega).$$

*We call them lower and upper densities, respectively.*

The following theorem gives the desired bounds.

**Theorem 13.** *Consider a set $\mathscr{P}$ of probabilities on a generic measurable space $(\Omega, \mathcal{F})$. If $\Omega$ is uncountable, assume that every $P \in \mathscr{P}$ is dominated by a $\sigma$-finite measure $\mu$ and that the Radon-Nikodym derivatives $\frac{dP}{d\mu}$ are continuous and bounded, for all $P \in \mathscr{P}$. Define*

$$H(\underline{P}) := \begin{cases} -\int_\Omega \log\left[\overline{\pi}(\omega)\right] \underline{P}(d\omega) & \text{if } \Omega \text{ is uncountable} \\ -\sum_{\omega \in \Omega} \underline{P}(\{\omega\}) \log[\overline{P}(\{\omega\})] & \text{if } \Omega \text{ is at most countable} \end{cases}$$

*and similarly*

$$H(\overline{P}) := \begin{cases} -\int_\Omega \log\left[\underline{\pi}(\omega)\right] \overline{P}(d\omega) & \text{if } \Omega \text{ is uncountable} \\ -\sum_{\omega \in \Omega} \overline{P}(\{\omega\}) \log[\underline{P}(\{\omega\})] & \text{if } \Omega \text{ is at most countable} \end{cases},$$

*Then, $\overline{H}(P) \leq H(\overline{P})$ and $\underline{H}(P) \geq H(\underline{P})$. In addition, if $\overline{P}$ is concave, the first bound is tighter, and if $\underline{P}$ is convex, the second bound is tighter.*

**Remark 14.** *In Hüllermeier & Waegeman (2021, Section 4.6.1), the authors point out that Abellán & Moral (2000) presents a generalization of the Hartley measure, called generalized Hartley measure $GH(P)$, that can be used to disaggregate the total uncertainty captured by $\overline{H}(P)$ into aleatoric and epistemic uncertainties.*

*We prefer not to introduce it in the present work because $GH(P)$ is defined based on the mass function of a belief function (Gong & Meng, 2021, Definition 2.4).[24] This entails that the authors assume that lower probability $\underline{P}$ associated with the set of probabilities of interest is a belief function, and so that for every collection $\{A, A_1, \ldots, A_k\}$ such that $A \subset A_i$, the following holds*

$$\underline{P}(A) \geq \sum_{\emptyset \neq I \subset \{1,\ldots,k\}} (-1)^{\#I-1} \underline{P}(\cap_{i \in I} A_i),$$

*for all $k \in \mathbb{N}$. As it is immediate to see, this is assumption is not needed in the context of CBDL. Other ways of disentangling and quantifying AU and EU can be found e.g. in Sale et al. (2023; 2024)*

## F    How to derive a predictive distribution

Suppose we performed a Bayesian updating procedure so to obtain posterior pdf $p(\theta \mid x_1, y_1, \ldots, x_n, y_n)$. Recall that $\{(x_i, y_i)\}_{i=1}^n \in (\mathcal{X} \times \mathcal{Y})^n$ denotes the training set. We obtain the predictive distribution $p(\tilde{y} \mid \tilde{x}, x_1, y_1, \ldots, x_n, y_n)$ on $\mathcal{Y}$ as follows

$$
\begin{aligned}
p(\tilde{y} \mid \tilde{x}, x_1, y_1, \ldots, x_n, y_n) &= \int_\Theta p(\tilde{y}, \theta \mid \tilde{x}, x_1, y_1, \ldots, x_n, y_n) \mathrm{d}\theta \\
&= \int_\Theta p(\tilde{y} \mid \theta, \tilde{x}, x_1, y_1, \ldots, x_n, y_n) \cdot p(\theta \mid \tilde{x}, x_1, y_1, \ldots, x_n, y_n) \mathrm{d}\theta \\
&= \int_\Theta p(\tilde{y} \mid \tilde{x}, \theta) \cdot p(\theta \mid x_1, y_1, \ldots, x_n, y_n) \mathrm{d}\theta,
\end{aligned}
$$

where $p(\tilde{y} \mid \tilde{x}, \theta)$ is the likelihood used to derive the posterior. Notice that the last equality comes from output $\tilde{y}$ only depending on input $\tilde{x}$ and parameter $\theta$, and from having assumed $D_{\mathbf{x}} \perp\!\!\!\perp \theta$ (see section 2.1). From an applied point of view, a sample from $p(\tilde{y} \mid \tilde{x}, x_1, y_1, \ldots, x_n, y_n)$ is obtained as follows:

1. specify input $\tilde{x}$;

2. sample a parameter $\tilde{\theta}$ from the posterior, $\tilde{\theta} \sim p(\theta \mid x_1, y_1, \ldots, x_n, y_n)$;

3. plug $\tilde{\theta}$ in the likelihood and sample $\tilde{y} \sim p(y \mid \tilde{\theta}, \tilde{x})$.

## G    Aleatoric uncertainty check for $\alpha$-level IHDR

The diameter of $IR_\alpha(\hat{\mathcal{P}}_{\text{pred}})$ is a function of the predictive aleatoric uncertainty (AU) faced by the agent.[25] If we want to avoid to perform the computation in **Step 5** of Algorithm 1 only to discover that $IR_\alpha(\hat{\mathcal{P}}_{\text{pred}})$ is "too large", then we can add a "predictive AU check".

At the beginning of the analysis, compute the lower entropy $\underline{H}(\hat{P}^{\text{pred}}) = \min_{k,s} H(\hat{P}^{\text{pred}}_{k,s})$ associated with the set $\text{ex}\hat{\mathcal{P}}_{\text{pred}}$ of extreme elements of the VI-approximated predictive credal set $\hat{\mathcal{P}}_{\text{pred}}$. By equation 2, it is equal to the predictive aleatoric uncertainty encoded in $\hat{\mathcal{P}}_{\text{pred}}$. We then verify whether the lower entropy $\underline{H}(\hat{P}^{\text{pred}})$ is "too high". That is, if $\underline{H}(\hat{P}^{\text{pred}}) > \varphi$, for some $\varphi > 0$, we want our procedure to abstain. This means that if the predictive aleatoric uncertainty in set $\hat{\mathcal{P}}_{\text{pred}}$ is too high, then our procedure does not return any output set for input $\tilde{x}$. The value of $\varphi$ can be set equal to the entropy of the probability measures that are typically used in the context the agent works in. For example, in medical applications the agent may consider the entropy of a Normal distribution, while in financial applications the entropy of a distribution with fatter tails, such as a $t$-distribution or a Cauchy. We call these *reference $\varphi$ values*.

If we add this "predictive AU check", at inference time (that is, before **Step 4** of Algorithm 1), the agent needs to specify the pair of parameters $(\varphi, \alpha)$.

---

[24]A belief function is a mathematical concept that should not be confused with the term "belief" we used throughout the paper to address the agent's knowledge.

[25]Since the diameter is a metric concept, we assume that we can find a well-defined metric $d_{\mathbf{y}}$ on $\mathcal{Y}$. If that is not the case, we substitute the diameter with the notion of cardinality.

# H $\alpha$-level IHDR in a Classification setting

In classification problems, BNNs compute the probability vector

$$\varpi := \frac{1}{\#\Theta} \sum_{\theta \in \Theta} \Phi_{\theta|D}(x),$$

where we write $\Phi_{\theta|D}$ to highlight the fact that $\theta$ is sampled from posterior $p(\theta \mid D)$, and then select the most likely class $\hat{y} := \arg\max_j \varpi_j$, where the $\varpi_j$'s are the elements of $\varpi$.

When applied to a classification setting, the general procedure introduced in Algorithm 1 becomes the following. Recall that we denote by $K$ the cardinality of $\mathrm{ex}\mathcal{P}_{\mathrm{prior}}$, and by $S$ the cardinality of $\mathrm{ex}\mathcal{P}_{\mathrm{lik}}$. Assume that $\mathcal{Y} = \{y_1, \ldots, y_J\}$, that is, there are $J \in \mathbb{N}_{\geq 2}$ possible labels. Then, a VI-approximated predictive distribution $\hat{P}_{k,s}^{\mathrm{pred}}$ in $\mathrm{ex}\hat{\mathcal{P}}_{\mathrm{pred}}$ can be seen as $J$-dimensional probability vector $\hat{\mathbf{p}}_{k,s}^{\mathrm{pred}} = (\hat{p}_{k,s,1}^{\mathrm{pred}}, \ldots, \hat{p}_{k,s,J}^{\mathrm{pred}})^{\top}$, where $\hat{p}_{k,s,j}^{\mathrm{pred}} = \hat{P}_{k,s}^{\mathrm{pred}}(\{y_j\})$, for all $k \in \{1, \ldots, K\}$, $s \in \{1, \ldots, S\}$, and $j \in \{1, \ldots, J\}$.

Now fix any $k$ and any $s$, and define the partial order $\preceq_{k,s}$ on $\mathcal{Y}$ as $y_l \preceq_{k,s} y_i \iff \hat{p}_{k,s,l}^{\mathrm{pred}} \geq \hat{p}_{k,s,i}^{\mathrm{pred}}$ and $y_l \prec_{k,s} y_i \iff \hat{p}_{k,s,l}^{\mathrm{pred}} > \hat{p}_{k,s,i}^{\mathrm{pred}}$, where $i, l \in \{1, \ldots, J\}$, $i \neq l$. This means that we can order the labels according to the probability that $\hat{P}_{k,s}^{\mathrm{pred}}$ assigns to them: the first label will be the one having highest probability according to $\hat{P}_{k,s}^{\mathrm{pred}}$, the second label will have the second-highest probability according to $\hat{P}_{k,s}^{\mathrm{pred}}$, and so on.

Now order the label space $\mathcal{Y}$ according to $\preceq_{k,s}$ so to obtain

$$\mathcal{Y}^{k,s} := \{y_1^{k,s}, \ldots, y_J^{k,s}\}.$$

This means that $y_1^{k,s} \preceq_{k,s} y_j^{k,s}$, for all $j \in \{2, \ldots, J\}$, $y_2^{k,s} \preceq_{k,s} y_j^{k,s}$, for all $j \in \{3, \ldots, J\}$ (but $y_1^{k,s} \preceq_{k,s} y_2^{k,s}$), and so on. That is, we order the labels from the most to the least likely according to $\hat{P}_{k,s}^{\mathrm{pred}}$.

Then, we call $\alpha$-level credible set according to $\hat{P}_{k,s}^{\mathrm{pred}}$, $\alpha \in [0,1]$, the set

$$
\begin{aligned}
CS_\alpha(\hat{P}_{k,s}^{\mathrm{pred}}) := \Bigg\{ y_1^{k,s}, \ldots, y_j^{k,s} : &\sum_{i=1}^{j} \hat{P}_{k,s}^{\mathrm{pred}}(\{y_i^{k,s}\}) \in [1-\alpha, 1-\alpha+\varepsilon], \ j \leq J, \\
&\text{and } \nexists j' < j : \sum_{i=1}^{j'} \hat{P}_{k,s}^{\mathrm{pred}}(\{y_i^{k,s}\}) \in [1-\alpha, 1-\alpha+\varepsilon] \Bigg\},
\end{aligned}
\tag{6}
$$

for some $\varepsilon > 0$. It corresponds to the $\alpha$-level HDR $R_\alpha(\hat{P}_{k,s}^{\mathrm{pred}})$. Notice that we require $\sum_{i=1}^{j} \hat{P}_{k,s}^{\mathrm{pred}}(\{y_i^{k,s}\}) \in [1-\alpha, 1-\alpha+\varepsilon]$ because we may need to go slightly above level $1-\alpha$. Just as a toy example, we may have 7 labels, 3 of which would give a 0.945 coverage, while 4 would give a coverage of 0.953. If we are interested in the $\alpha = 0.05$-level credible set, we ought to include the fourth label, thus yielding a coverage slightly higher than $1 - \alpha = 0.95$. The interpretation to $CS_\alpha(\hat{P}_{k,s}^{\mathrm{pred}})$ is the following: it consists of the smallest collection of labels to which $\hat{P}_{k,s}^{\mathrm{pred}}$ assigns probability of at least $1 - \alpha$ (that is, those having the highest probability of being the correct one for the new input $\tilde{x}$).

Finally, we call $\alpha$-level imprecise credible set, $\alpha \in [0,1]$, the set

$$ICS_\alpha(\hat{\mathcal{P}}_{\mathrm{pred}}) := \bigcup_{k,s} CS_\alpha(\hat{P}_{k,s}^{\mathrm{pred}}).$$

In turn, we have that $\underline{\hat{P}}^{\mathrm{pred}}[\{\tilde{y} \in ICS_\alpha(\hat{\mathcal{P}}_{\mathrm{pred}})] \geq 1 - \alpha$.

**Remark 15.** *Notice that if a credible set of level $\approx \alpha$ is enough, then we can replace the left endpoint of the interval in equation 6 with $1 - (\alpha + \varepsilon_{k,s})$, for some $\varepsilon_{k,s} > 0$. Strictly speaking, in this case we obtain an $(\alpha + \varepsilon_{k,s})$-level credible set, which we denote by $CS_{\alpha_{k,s}}(\hat{P}_{k,s}^{pred})$, where $\alpha_{k,s} := \alpha + \varepsilon_{k,s}$. Going back to our toy example, we will have a credible set with 3 labels that yields a coverage of $1 - (\alpha + \varepsilon_{k,s}) = 0.945 \approx 0.95 = 1 - \alpha$, so $\varepsilon_{k,s} = 0.005$. In turn, this implies that the imprecise credible set will have a coverage of $1 - (\alpha + \max_{k,s} \varepsilon_{k,s})$, that is, it will have level $\alpha + \max_{k,s} \varepsilon_{k,s}$. We denote it by $ICS_{\tilde{\alpha}}(\hat{\mathcal{P}}_{pred})$, where $\tilde{\alpha} := \alpha + \max_{k,s} \varepsilon_{k,s}$.*

# I  Distance between a set of distributions $\mathcal{P}$ and a single distribution $P'$ having different dimensions

Many concepts in this section are derived from Cai & Lim (2022). Let $m, n \in \mathbb{N}$ such that $m \leq n$, and let $p \in [1, \infty]$. Call $M^p(\mathbb{R}^j)$ the set of probability measures on $\mathbb{R}^j$ having finite $p$-th moment, and $M_d(\mathbb{R}^j)$ the set of probability measures on $\mathbb{R}^j$ having density with respect to some $\sigma$-finite dominating measure $\mu$, $j \in \{m, n\}$. Let $O(m, n) := \{V \in \mathbb{R}^{m \times n} : VV^\top = I_m\}$, where $I_m$ denotes the $m$-dimensional identity matrix, and for any $V \in O(m, n)$ and any $b \in \mathbb{R}^m$, define the following function

$$\varphi_{V,b} : \mathbb{R}^n \to \mathbb{R}^m, \quad x \mapsto \varphi_{V,b}(x) := Vx + b.$$

Let $\mathcal{B}(\mathbb{R}^n)$ be the Borel $\sigma$-algebra on $\mathbb{R}^n$, and for any $Q \in \Delta(\mathbb{R}^n, \mathcal{B}(\mathbb{R}^n))$, define $\varphi_{V,b}(Q) := Q \circ \varphi_{V,b}^{-1}$, the pushforward measure. Consider then two generic probability measures $Q, S$ such that $Q \in M^p(\mathbb{R}^m)$ and $S \in M^p(\mathbb{R}^n)$, and call

$$\Phi_p^+(Q, n) := \{\alpha \in M^p(\mathbb{R}^n) : \varphi_{V,b}(\alpha) = Q, \text{ for some } V \in O(m, n), b \in \mathbb{R}^m\},$$
$$\Phi_d^+(Q, n) := \{\alpha \in M_d(\mathbb{R}^n) : \varphi_{V,b}(\alpha) = Q, \text{ for some } V \in O(m, n), b \in \mathbb{R}^m\},$$
$$\Phi^-(S, m) := \{\beta \in M(\mathbb{R}^m) : \varphi_{V,b}(S) = \beta, \text{ for some } V \in O(m, n), b \in \mathbb{R}^m\}.$$

Recall now the definition of $p$-Wasserstein metric between two generic distributions defined on the *same* Euclidean space. Let $P_1, P_2 \in M^p(\mathbb{R}^n)$, for some $n \in \mathbb{N}$ and some $p \in [1, \infty]$. Then, the $p$-Wasserstein distance between them is defined as

$$W_p(P_1, P_2) := \left[ \inf_{\gamma \in \Gamma(P_1, P_2)} \int_{\mathbb{R}^{2n}} \|x - y\|_2^p \gamma(\mathrm{d}(x, y)) \right]^{1/p},$$

where $\|\cdot\|_2$ denotes the Euclidean distance, $p = \infty$ is interpreted as the essential supremum, and $\Gamma(P_1, P_2) := \{\gamma \in \Delta(\mathbb{R}^{2n}, \mathcal{B}(\mathbb{R}^{2n})) : \mathrm{proj}_1^n(\gamma) = P_2, \mathrm{proj}_2^n(\gamma) = P_1\}$ is the set of couplings between $P_1$ and $P_2$, where $\mathrm{proj}_1^n$ is the projection onto the first $n$ coordinates, and $\mathrm{proj}_2^n$ is the projection to the last $n$ coordinates.

Recall then the definition of $f$-divergence between two generic distributions defined on the *same* Euclidean space. Let $P_1, P_2 \in \Delta(\mathbb{R}^n, \mathcal{B}(\mathbb{R}^n))$, for some $n \in \mathbb{N}$, and assume $P_1 \ll P_2$. Then, for any convex functional $f$ on $\mathbb{R}$ such that $f(1) = 0$, the $f$-divergence between $P_1$ and $P_2$ is defined as

$$\mathrm{div}_f(P_1 \| P_2) := \int_{\mathbb{R}^n} f\left( \frac{\mathrm{d}P_1}{\mathrm{d}P_2}(x) \right) P_2(\mathrm{d}x).$$

Aside from the Rényi divergence, the $f$-divergence includes just about every known divergences as special case (Cai & Lim, 2022). The following are the main results of this section.

**Lemma 16.** *Let $m, n \in \mathbb{N}$ such that $m \leq n$, and let $p \in [1, \infty]$ and $f$ be any convex functional on $\mathbb{R}$ such that $f(0) = 1$. Consider a generic $\mathscr{P} \subset M^p(\mathbb{R}^m)$ and $P' \in M^p(\mathbb{R}^n)$. Let $\Phi_p^+(\mathscr{P}, n) := \cup_{P \in \mathscr{P}} \Phi_p^+(P, n)$ and $\Phi_d^+(\mathscr{P}, n) := \cup_{P \in \mathscr{P}} \Phi_d^+(P, n)$. Define*

- *$W_p^+(P, P') := \inf_{\alpha \in \Phi_p^+(P, n)} W_p(\alpha, P')$, for all $P \in \mathscr{P}$;*

- *$div_f^+(P \| P') := \inf_{\alpha \in \Phi_d^+(P, n)} div_f(P \| P')$, for all $P \in \mathscr{P}$;*

- *$W_p^+(\mathscr{P}, P') := \inf_{\alpha \in \Phi_p^+(\mathscr{P}, n)} W_p(\alpha, P')$;*

- *$div_f^+(\mathscr{P} \| P') := \inf_{\alpha \in \Phi_d^+(\mathscr{P}, n)} div_f(P \| P')$;*

*Then, for all $P \in \mathscr{P}$ the following holds*

$$W_p^+(\mathscr{P}, P') \leq W_p^+(P, P') \quad \text{and} \quad div_f^+(\mathscr{P} \| P') \leq div_f^+(P \| P').$$

**Lemma 17.** *Let $m, n \in \mathbb{N}$ such that $m \leq n$, and let $p \in [1, \infty]$ and $f$ be any convex functional on $\mathbb{R}$ such that $f(0) = 1$. Consider a generic $\mathscr{P} \subset M^p(\mathbb{R}^n)$ and $P' \in M^p(\mathbb{R}^m)$. Let $\Phi^-(\mathscr{P}, m) := \cup_{P \in \mathscr{P}} \Phi^-(P, m)$. Define*

- $W_p^-(P, P') := \inf_{\alpha \in \Phi^-(P, m)} W_p(\alpha, P')$, *for all* $P \in \mathscr{P}$;

- $div_f^-(P\|P') := \inf_{\alpha \in \Phi^-(P, m)} div_f(P\|P')$, *for all* $P \in \mathscr{P}$;

- $W_p^-(\mathscr{P}, P') := \inf_{\alpha \in \Phi^-(\mathscr{P}, m)} W_p(\alpha, P')$;

- $div_f^-(\mathscr{P}\|P') := \inf_{\alpha \in \Phi^-(\mathscr{P}, m)} div_f(P\|P')$;

*Then, for all $P \in \mathscr{P}$ the following holds*

$$W_p^-(\mathscr{P}, P') \leq W_p^-(P, P') \quad and \quad div_f^-(\mathscr{P}\|P') \leq div_f^-(P\|P').$$

A visual representation of the application of Lemma 17 in the context of CBDL is given in Figure 11.

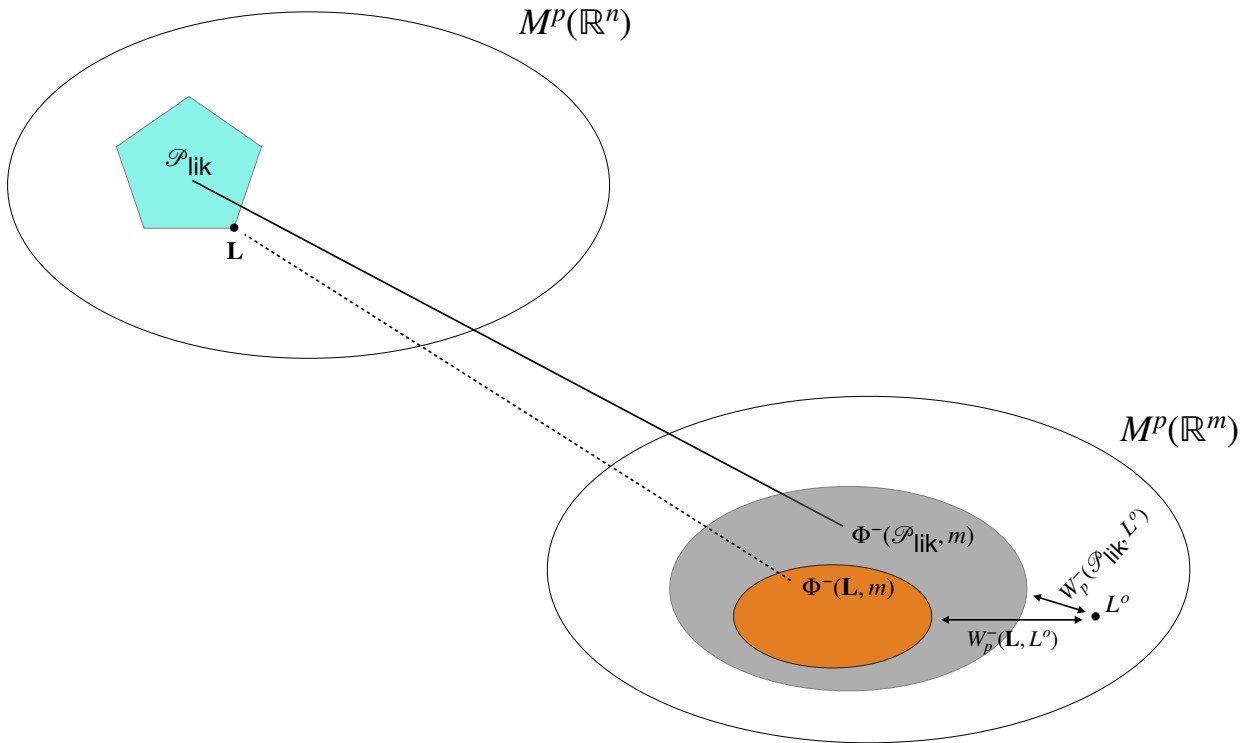

Figure 11: We assume that $n > m$ and that the oracle distribution $L^o$ belongs to $M^p(\mathbb{R}^m)$, while likelihood FGCS $\mathcal{P}_{\text{lik}}$ is a subset of $M^p(\mathbb{R}^n)$, for some finite $p \geq 1$. We also assume that $\mathbf{L}$ is one of the extreme elements of $\mathcal{P}_{\text{lik}}$. We see how $W_p^-(\mathcal{P}_{\text{lik}}, L^o) < W_p^-(\mathbf{L}, L^o)$; if we replace metric $W_p^-$ by a generic $f$-divergence $\text{div}_f$, the inequality would still hold thanks to Lemma 17.

## J Details on Artificial Pancreas Example

**Artificial Pancreas Model.** An important factor when designing the controller for an artificial pancreas is to adapt the insulin delivery algorithm to the particular details of the patient. This is because patients display a wide range of variability in their response to insulin, depending on age, Body Mass Index (BMI), and other physiological parameters. The Bayesian Neural Network models have 2 hidden layers, with 10

neurons each, for the case study with 4 different seeds. This choice was informed by the experiments in Dutta et al. (2018). For the case study with different architectures, we trained BNNs with 4 different widths: 10, 20, 30, and 40. The horizon length is $H = 10$ time steps, and the prediction horizon is 5 steps into the future. The neural networks were trained for 200 time steps, with a learning rate of 0.001, and batch size of 128 using Mean-Field Variational Inference. The training dataset consisted of 28400 training samples, recorded without meals.

**Controller.** We implemented a simple model predictive controller, using an off-the-shelf implementation of the covariance matrix adaptation evolution strategy (CMA-ES). The model predictive control planning horizon was $k = 5$, with a fixed seed for the randomized solver.

## K Other possible baselines for CBDL

Our experiments may seem like a type of belief tracking (Fortin et al., 2017; Klein et al., 2010). Two comments are in order. First, this line of literature does not use deep learning techniques. Second, in these works the authors rely on Dempster-Shafer theory, a field in imprecise probability theory where lower probabilities are assumed to be belief functions, see Remark 14. We do not rely on this assumption in our work.

## L Further related work

Modeling uncertainty has been a longstanding goal of ML/AI research and a variety of approaches have been developed for doing so (Guo et al., 2017; Park et al., 2020; Jospin et al., 2022). Recently, emphasis has been placed on discerning between aleatoric and epistemic uncertainties (Senge et al., 2014; Kull & Flach, 2014; Kendall & Gal, 2017). In Tretiak et al. (2022), the authors present an IP-based neural network which uses a regression technique based on probability intervals. Contrary to CBDL, their NN is rooted in the frequentist approach to imprecise probabilities (Huber & Ronchetti, 2009).

In Manchingal & Cuzzolin (2022, Sections 2.1, 2.3), the authors focus on belief-functions-based classification methods. CBDL cannot be directly compared with these methodologies because (i) they do not require that the user expresses their knowledge via a belief function, but rather through a credal set; (ii) they can be used for regression and classification; (iii) they are rooted in Bayesian theory, as opposed to Dempster-Shafer theory. Other works in ML using a belief function approach are those from the field of evidential machine learning (EML), see e.g. (Amini et al., 2020; Charpentier et al., 2020; Denœux, 2022; 2023; Sensoy et al., 2018) and references therein. Existing models mainly address clustering, classification, and regression problems. The reasons why CBDL cannot be directly compared with methods from the EML literature are (i) and (iii) above, the fact that EML methods are not (derived from) Bayesian ones, and that their definitions of AU and EU are slightly different from the canonical ones in ensemble deep learning.

## M Proofs

*Proof of Proposition 3.* If $\overline{P}(A) = \sup_{P' \in \Pi'} P'(A)$, for all $A \in \mathcal{F}$, then it is immediate to see that $\overline{P}(A) = \sup_{P \in \Pi} P(A) = \sup_{P' \in \text{ex}\Pi'} P'(A) = \sup_{P' \in \Pi'} P'(A)$, for all $A \in \mathcal{F}$, since $\Pi \subset \Pi'$.

Suppose now that $\overline{P}(A) = \sup_{P \in \Pi} P(A)$, for all $A \in \mathcal{F}$. Then, we have that for all $P \in \Pi$ and all $A \in \mathcal{F}$, $P(A) \leq \overline{P}(A)$. Pick now any $P' \in \Pi'$. We can write it as $P' = \sum_{j=1}^{k} \beta_j P_j$, where $\beta_j \in [0,1]$, for all $j \in \{1, \ldots, k\}$, $\sum_{j=1}^{k} \beta_j = 1$, and $\{P_j\}_{j=1}^{k} = \Pi$. Pick then any $A \in \mathcal{F}$; we have

$$P'(A) = \sum_{j=1}^{k} \beta_j P_j(A) \leq \sum_{j=1}^{k} \beta_j \overline{P}(A) = \overline{P}(A).$$

So $\overline{P}(A) \geq P'(A)$. Because this holds for all $P' \in \Pi'$ and all $A \in \mathcal{F}$, the claim is proven. $\square$

*Proof of Proposition 6.* The lower bound for the upper entropy of $\Pi'$ comes immediately from $\Pi'$ being a superset of $\Pi$.

Let us now prove the lower entropy equality. Let $\Pi = \{P_1, \dots, P_k\}$ and $\Pi' = \text{Conv}\Pi$. Pick any $P' \in \Pi'$. By the definition of $\Pi'$, there exists a collection of non-negative reals $\{\beta_j\}_{j=1}^k$ such that $\sum_{j=1}^k \beta_j = 1$ and $\sum_{j=1}^k \beta_j P_j = P'$. By the concavity of the entropy, we have that

$$H(P') = H\left(\sum_{j=1}^k \beta_j P_j\right) \geq \sum_{j=1}^k \beta_j H(P_j) \geq \sum_{j=1}^k \beta_j \underline{H}(P) = \underline{H}(P) := \inf_{P \in \Pi} H(P).$$

Since $P'$ was chosen arbitrarily and $\Pi$ is finite, this implies that

$$\inf_{P' \in \Pi'} H(P') =: \underline{H}(P') \geq \underline{H}(P).$$

In addition, we have that, since $\Pi \subset \Pi'$, $\underline{H}(P') \leq \underline{H}(P)$. Combining this with the above result, we obtain

$$\underline{H}(P') = \underline{H}(P).$$

$\square$

*Proof of Proposition 8.* Pick any metric $d$ on the space $\Delta_\Omega \equiv \Delta(\Omega, \mathcal{F})$ of probabilities on $(\Omega, \mathcal{F})$, and any $\mathbf{P}' \in \Pi'$. Because $\mathbf{P}'$ belongs to $\Pi'$, $\inf_{P' \in \Pi'} d(P', \Psi)$ can only be either equal to or smaller than $d(\mathbf{P}', \Psi)$. By the definition of $d(\Pi', \Psi)$, if $\inf_{P' \in \Pi'} d(P', \Psi) = d(\mathbf{P}', \Psi)$, then $d(\Pi', \Psi) = d(\mathbf{P}', \Psi)$. If instead $\inf_{P' \in \Pi'} d(P', \Psi) < d(\mathbf{P}', \Psi)$, then $d(\Pi', \Psi) < d(\mathbf{P}', \Psi)$. The proof is similar for a generic divergence div on $\Delta(\Omega, \mathcal{F})$. $\square$

*Proof of Theorem 10.* Assume $\mathcal{P}^{\text{co}}, \mathcal{L}_\theta^{\text{co}}$ are nonempty. Then, by Marinacci & Montrucchio (2004, Proposition 3), they are weak$^\star$-compact. Pick any $A \in \mathcal{B}$. Recall that we can rewrite the usual Bayes' updating rule as

$$P_D(A) = \frac{\int_\Theta L(\theta)\mathbb{1}_A(\theta)P(\mathrm{d}\theta)}{\int_\Theta L(\theta)\mathbb{1}_A(\theta)P(\mathrm{d}\theta) + \int_\Theta L(\theta)\mathbb{1}_{A^c}(\theta)P(\mathrm{d}\theta)}$$

$$= \frac{1}{1 + \frac{\int_\Theta L(\theta)\mathbb{1}_{A^c}(\theta)P(\mathrm{d}\theta)}{\int_\Theta L(\theta)\mathbb{1}_A(\theta)P(\mathrm{d}\theta)}},$$

which is maximized when

$$\frac{\int_\Theta L(\theta)\mathbb{1}_{A^c}(\theta)P(\mathrm{d}\theta)}{\int_\Theta L(\theta)\mathbb{1}_A(\theta)P(\mathrm{d}\theta)}$$

is minimized. But

$$\frac{\int_\Theta L(\theta)\mathbb{1}_{A^c}(\theta)P(\mathrm{d}\theta)}{\int_\Theta L(\theta)\mathbb{1}_A(\theta)P(\mathrm{d}\theta)} \geq \frac{\inf_{P \in \mathcal{P}^{\text{co}}} \int_\Theta \underline{L}(\theta)\mathbb{1}_{A^c}(\theta)P(\mathrm{d}\theta)}{\sup_{P \in \mathcal{P}^{\text{co}}} \int_\Theta \overline{L}(\theta)\mathbb{1}_A(\theta)P(\mathrm{d}\theta)},$$

which proves the inequality in equation 5. Assume now that $\overline{P}$ is concave. By Wasserman & Kadane (1990, Lemma 1), we have that there exists $\mathbf{P} \in \mathcal{P}^{\text{co}}$ such that

$$\sup_{P \in \mathcal{P}^{\text{co}}} \int_\Theta L(\theta)\mathbb{1}_A(\theta)P(\mathrm{d}\theta) = \int_\Theta L(\theta)\mathbb{1}_A(\theta)\mathbf{P}(\mathrm{d}\theta), \tag{7}$$

for all $L \in \mathcal{L}$. In addition, by Wasserman & Kadane (1990, Lemma 4), we have that for all $X \in \mathcal{X}$ and all $\epsilon > 0$, there exists a non-negative, upper semi-continuous function $h \leq X$ such that

$$\left[\sup_{P \in \mathcal{P}^{\text{co}}} \int_\Theta X(\theta)P(\mathrm{d}\theta)\right] - \epsilon < \sup_{P \in \mathcal{P}^{\text{co}}} \int_\Theta h(\theta)P(\mathrm{d}\theta)$$

$$\leq \sup_{P \in \mathcal{P}^{\text{co}}} \int_\Theta X(\theta)P(\mathrm{d}\theta). \tag{8}$$

Let now $X = \overline{L}\mathbb{1}_A$. Notice that since $\mathcal{L}_\theta^{co}$ is weak$^\star$-compact, by equation 4 so is $\mathscr{L}$. This implies that $\underline{L}, \overline{L} \in \mathscr{L}$, since a compact set always contains its boundary, so $\overline{L} \in \mathscr{X}$ as well, and in turn $\overline{L}\mathbb{1}_A \in \mathscr{X}$. Fix then any $L \in \mathscr{L}$ and put $h = L\mathbb{1}_A$. It is immediate to see that $h$ is non-negative and upper semi-continuous. Then, by equation 8, we have that for all $\epsilon > 0$

$$\left[\sup_{P\in\mathcal{P}^{co}} \int_\Theta \overline{L}(\theta)\mathbb{1}_A(\theta)P(\mathrm{d}\theta)\right] - \epsilon < \tag{9}$$
$$\sup_{P\in\mathcal{P}^{co}} \int_\Theta L(\theta)\mathbb{1}_A(\theta)P(\mathrm{d}\theta) \leq \sup_{P\in\mathcal{P}^{co}} \int_\Theta \overline{L}(\theta)\mathbb{1}_A(\theta)P(\mathrm{d}\theta).$$

Combining equation 7 andequation 9, we obtain

$$\left[\sup_{P\in\mathcal{P}^{co}} \int_\Theta \overline{L}(\theta)\mathbb{1}_A(\theta)P(\mathrm{d}\theta)\right] - \epsilon$$
$$< \int_\Theta L(\theta)\mathbb{1}_A(\theta)\mathbf{P}(\mathrm{d}\theta) \leq \sup_{P\in\mathcal{P}^{co}} \int_\Theta \overline{L}(\theta)\mathbb{1}_A(\theta)P(\mathrm{d}\theta), \tag{10}$$

for all $L \in \mathscr{L}$.

Pick now any $\epsilon > 0$ and put

$$k := \sup_{P\in\mathcal{P}^{co}} \int_\Theta \overline{L}(\theta)\mathbb{1}_A(\theta)P(\mathrm{d}\theta)$$
$$+ \inf_{P\in\mathcal{P}^{co}} \int_\Theta \underline{L}(\theta)\mathbb{1}_{A^c}(\theta)P(\mathrm{d}\theta) > 0.$$

Choose any $L \in \mathscr{L}$ and $\delta \in (0, \epsilon k)$. By equation 10 we have that $[\sup_{P\in\mathcal{P}^{co}} \int_\Theta \overline{L}(\theta)\mathbb{1}_A(\theta)P(\mathrm{d}\theta)] - \delta < \int_\Theta L(\theta)\mathbb{1}_A(\theta)\mathbf{P}(\mathrm{d}\theta)$ and that $[\inf_{P\in\mathcal{P}^{co}} \int_\Theta \underline{L}(\theta)\mathbb{1}_{A^c}(\theta)P(\mathrm{d}\theta)] + \delta > \int_\Theta L(\theta)\mathbb{1}_{A^c}(\theta)\mathbf{P}(\mathrm{d}\theta)$. Recall that $\mathbf{c} := \sup_{P\in\mathcal{P}^{co}} \int_\Theta \overline{L}(\theta)\mathbb{1}_A(\theta)P(\mathrm{d}\theta) + \inf_{P\in\mathcal{P}^{co}} \int_\Theta \underline{L}(\theta)\mathbb{1}_{A^c}(\theta)P(\mathrm{d}\theta)$, and define $\mathbf{d} := \int_\Theta L(\theta)\mathbb{1}_A(\theta)\mathbf{P}(\mathrm{d}\theta) + \int_\Theta L(\theta)\mathbb{1}_{A^c}(\theta)\mathbf{P}(\mathrm{d}\theta)$. Then,

$$\mathbf{P}_D(A) = \frac{\int_\Theta L(\theta)\mathbb{1}_A(\theta)\mathbf{P}(\mathrm{d}\theta)}{\mathbf{d}}$$
$$\geq \frac{\left[\sup_{P\in\mathcal{P}^{co}} \int_\Theta \overline{L}(\theta)\mathbb{1}_A(\theta)P(\mathrm{d}\theta)\right] - \delta}{\mathbf{c} + \delta - \delta}$$
$$= \frac{\sup_{P\in\mathcal{P}^{co}} \int_\Theta \overline{L}(\theta)\mathbb{1}_A(\theta)P(\mathrm{d}\theta)}{\mathbf{c}} - \frac{\delta}{k}$$
$$> \frac{\sup_{P\in\mathcal{P}^{co}} \int_\Theta \overline{L}(\theta)\mathbb{1}_A(\theta)P(\mathrm{d}\theta)}{\mathbf{c}} - \epsilon.$$

Since this holds for all $\epsilon > 0$, we have that

$$\sup_{P_D\in\mathcal{P}_D^{co}} P_D(A) = \frac{\sup_{P\in\mathcal{P}^{co}} \int_\Theta \overline{L}(\theta)\mathbb{1}_A(\theta)P(\mathrm{d}\theta)}{\mathbf{c}},$$

concluding the proof. □

*Proof of Lemma 11.* Walley (1981); Wasserman & Kadane (1990) show that concave upper probabilities are closed with respect to the generalized Bayes' rule. In particular, this means that, if we let $\mathbf{b} := \sup_{P\in\mathcal{P}^{co}} \int_\Theta L(\theta)\mathbb{1}_A(\theta)P(\mathrm{d}\theta) + \inf_{P\in\mathcal{P}^{co}} \int_\Theta L(\theta)\mathbb{1}_{A^c}(\theta)P(\mathrm{d}\theta)$, for any fixed $A \in \mathcal{B}$, if $\overline{P}$ is concave, then for all $L \in \mathscr{L}$

$$\overline{P}_D(A) = \frac{\sup_{P\in\mathcal{P}^{co}} \int_\Theta L(\theta)\mathbb{1}_A(\theta)P(\mathrm{d}\theta)}{\mathbf{b}} \tag{11}$$

is concave. But since $\mathcal{L}_\theta^{co}$ is weak$^\star$-compact (by our assumption and Marinacci & Montrucchio (2004, Proposition 3)), by equation 4 so is $\mathscr{L}$. This implies that $\underline{L}, \overline{L} \in \mathscr{L}$, since a compact set always contains its

boundary. Call then $L' = \overline{L}\mathbb{1}_A + \underline{L}\mathbb{1}_{A^c}$. It is immediate to see that $L' \in \mathscr{L}$. Then, by equation 11 we have that if we call $\mathbf{b}' := \sup_{P \in \mathcal{P}^{co}} \int_\Theta L'(\theta)\mathbb{1}_A(\theta)P(\mathrm{d}\theta) + \inf_{P \in \mathcal{P}^{co}} \int_\Theta L'(\theta)\mathbb{1}_{A^c}(\theta)P(\mathrm{d}\theta)$, it follows that

$$\overline{P}_D(A) = \frac{\sup_{P \in \mathcal{P}^{co}} \int_\Theta L'(\theta)\mathbb{1}_A(\theta)P(\mathrm{d}\theta)}{\mathbf{b}'}$$
$$= \frac{\sup_{P \in \mathcal{P}^{co}} \int_\Theta \overline{L}(\theta)\mathbb{1}_A(\theta)P(\mathrm{d}\theta)}{\mathbf{c}}$$

is concave, concluding the proof. □

*Proof of Theorem 13.* Suppose $\Omega$ is uncountable. First notice that, since we assumed $\frac{\mathrm{d}P}{\mathrm{d}\mu}$ to be continuous and bounded for all $P \in \mathscr{P}$, then so is $\log \circ \frac{\mathrm{d}P}{\mathrm{d}\mu}$, for all $P \in \mathscr{P}$, since composing a continuous function with a continuous and bounded one gives us a continuous and bounded function. This entails that the Choquet integrals of $\log \circ \frac{\mathrm{d}P}{\mathrm{d}\mu}$ with respect to $\underline{P}$ and $\overline{P}$ are both well defined. In addition, being continuous and bounded, both $\frac{\mathrm{d}P}{\mathrm{d}\mu}$ and $\log \circ \frac{\mathrm{d}P}{\mathrm{d}\mu}$ attain their infima and suprema thanks to Weierstrass' extreme value theorem, for all $P \in \mathscr{P}$. Hence, all the Choquet integrals used in this proof are well defined.

Then, we have the following

$$\overline{H}(P) := \sup_{P \in \mathscr{P}} H(P) = \sup_{P \in \mathscr{P}} \left( -\int_\Omega \log\left[\frac{\mathrm{d}P}{\mathrm{d}\mu}(\omega)\right] P(\mathrm{d}\omega) \right)$$

$$= \sup_{P \in \mathscr{P}} \int_\Omega (-\log)\left[\frac{\mathrm{d}P}{\mathrm{d}\mu}(\omega)\right] P(\mathrm{d}\omega)$$

$$\leq \sup_{P \in \mathscr{P}} \int_\Omega \sup_{P \in \mathscr{P}} \left\{ (-\log)\left(\frac{\mathrm{d}P}{\mathrm{d}\mu}(\omega)\right) \right\} P(\mathrm{d}\omega) \tag{12}$$

$$\leq \int_\Omega \sup_{P \in \mathscr{P}} \left\{ (-\log)\left(\frac{\mathrm{d}P}{\mathrm{d}\mu}(\omega)\right) \right\} \overline{P}(\mathrm{d}\omega) \tag{13}$$

$$= -\int_\Omega \inf_{P \in \mathscr{P}} \left\{ \log\left(\frac{\mathrm{d}P}{\mathrm{d}\mu}(\omega)\right) \right\} \overline{P}(\mathrm{d}\omega) \tag{14}$$

$$= -\int_\Omega \log\left( \inf_{P \in \mathscr{P}} \frac{\mathrm{d}P}{\mathrm{d}\mu}(\omega) \right) \overline{P}(\mathrm{d}\omega) \tag{15}$$

$$= -\int_\Omega \log[\underline{\pi}(\omega)] \overline{P}(\mathrm{d}\omega) = H(\overline{P}).$$

The inequality in equation 12 is true because for all $\omega \in \Omega$,

$$\sup_{P \in \mathscr{P}} \left\{ (-\log)\left(\frac{\mathrm{d}P}{\mathrm{d}\mu}(\omega)\right) \right\} \geq (-\log)\left(\frac{\mathrm{d}P}{\mathrm{d}\mu}(\omega)\right).$$

The inequality in equation 13 is a property of Choquet integrals taken with respect to upper probabilities (Marinacci & Montrucchio, 2004). The equality in equation 14 is true because for a generic function $f$, we have that $\sup -f = -\inf f$. Finally, the equality in equation 15 is true because the logarithm is a strictly increasing function. By Marinacci & Montrucchio (2004, Theorem 38), if $\overline{P}$ is concave, then inequality equation 13 holds with an equality, and so the bound is tighter.

The proof for $\underline{H}(P) \geq H(\underline{P})$ is similar; we use the facts that

- $\inf_{P \in \mathscr{P}} \left\{ (-\log)\left(\frac{\mathrm{d}P}{\mathrm{d}\mu}(\omega)\right) \right\} \leq (-\log)\left(\frac{\mathrm{d}P}{\mathrm{d}\mu}(\omega)\right)$, for all $\omega \in \Omega$;

- by Marinacci & Montrucchio (2004),

$$\inf_{P \in \mathscr{P}} \int_\Omega \inf_{P \in \mathscr{P}} \left\{ (-\log)\left(\frac{\mathrm{d}P}{\mathrm{d}\mu}(\omega)\right) \right\} P(\mathrm{d}\omega)$$
$$\geq \int_\Omega \inf_{P \in \mathscr{P}} \left\{ (-\log)\left(\frac{\mathrm{d}P}{\mathrm{d}\mu}(\omega)\right) \right\} \underline{P}(\mathrm{d}\omega); \tag{16}$$

- for a generic function $f$, $\inf -f = -\sup f$;

- by Marinacci & Montrucchio (2004, Theorem 38), if $\underline{P}$ is convex, then equation 16 holds with an equality.

Suppose now $\Omega$ is at most countable; in this case, we do not need any assumptions to make the Choquet integrals well defined, since we will not deal with density functions. The following holds

$$
\begin{aligned}
\overline{H}(P) := \sup_{P \in \mathscr{P}} H(P) &= \sup_{P \in \mathscr{P}} \left( -\sum_{\omega \in \Omega} P(\{\omega\}) \log \left[ P(\{\omega\}) \right] \right) \\
&= \sup_{P \in \mathscr{P}} \sum_{\omega \in \Omega} P(\{\omega\})(-\log)\left[ P(\{\omega\}) \right] \\
&\leq \sum_{\omega \in \Omega} \sup_{P \in \mathscr{P}} \left\{ P(\{\omega\})(-\log)\left[ P(\{\omega\}) \right] \right\} & (17) \\
&\leq \sum_{\omega \in \Omega} \overline{P}(\{\omega\}) \sup_{P \in \mathscr{P}} (-\log)\left[ P(\{\omega\}) \right] & (18) \\
&= -\sum_{\omega \in \Omega} \overline{P}(\{\omega\}) \inf_{P \in \mathscr{P}} \log\left[ P(\{\omega\}) \right] & (19) \\
&= -\sum_{\omega \in \Omega} \overline{P}(\{\omega\}) \log \left[ \inf_{P \in \mathscr{P}} P(\{\omega\}) \right] & (20) \\
&= -\sum_{\omega \in \Omega} \overline{P}(\{\omega\}) \log \left[ \underline{P}(\{\omega\}) \right] = H(\overline{P}).
\end{aligned}
$$

The inequality in equation 17 comes from the well known fact that the sum of the suprema is at least equal to the supremum of the sum. The inequality in equation 18 comes from the fact that for differentiable functions, the product of the suprema is at least equal to the supremum of the product. The equality in equation 19 is true because for a generic function $f$, we have that $\sup -f = -\inf f$. Finally, the equality in equation 20 is true because the logarithm is a strictly increasing function. By Marinacci & Montrucchio (2004, Theorem 38), if $\overline{P}$ is concave, then inequality equation 17 holds with an equality, and so the bound is tighter.

The proof for $\underline{H}(P) \geq H(\underline{P})$ is similar; we use the facts that

- the sum of the infima is at most equal to the infimum of the sum;

- for differentiable functions, the product of the infima is at most equal to the infimum of the product;

- for a generic function $f$, $\inf -f = -\sup f$;

- by Marinacci & Montrucchio (2004, Theorem 38), if $\underline{P}$ is convex, then $\inf_{P \in \mathscr{P}} \sum_{\omega \in \Omega} P(\{\omega\})(-\log)\left[ P(\{\omega\}) \right] = \sum_{\omega \in \Omega} \inf_{P \in \mathscr{P}} \left\{ P(\{\omega\})(-\log)\left[ P(\{\omega\}) \right] \right\}$.

$\square$

*Proof of Lemma 16.* Fix any $p \in [1, \infty]$ and pick any $\mathbf{P} \in \mathscr{P}$. Because $\Phi_p^+(\mathbf{P}, n) \subset \Phi_p^+(\mathscr{P}, n)$, then $\inf_{\alpha \in \Phi_p^+(\mathscr{P}, n)} W_p(\alpha, P')$ can only be either equal or smaller than $\inf_{\alpha \in \Phi_p^+(\mathbf{P}, n)} W_p(\alpha, P')$. Now, if $\inf_{\alpha \in \Phi_p^+(\mathscr{P}, n)} W_p(\alpha, P') = \inf_{\alpha \in \Phi_p^+(\mathbf{P}, n)} W_p(\alpha, P')$, then $W_p^+(\mathscr{P}, P') = W_p^+(\mathbf{P}, P')$. If instead $\inf_{\alpha \in \Phi_p^+(\mathscr{P}, n)} W_p(\alpha, P') < \inf_{\alpha \in \Phi_p^+(\mathbf{P}, n)} W_p(\alpha, P')$, then $W_p^+(\mathscr{P}, P') < W_p^+(\mathbf{P}, P')$. This concludes the first part of the proof. Fix then any convex functional $f$ on $\mathbb{R}$ such that $f(0) = 1$; the proof is similar for $f$-divergences. $\square$

*Proof of Lemma 17.* The proof is very similar to that of Lemma 16. $\square$

# N  All Results: In-distribution and Out-of-distribution Evaluation

## N.1  CIFAR-10 Results

| | | CBDL | | | | | | Ensemble | | | | | | BNN | | | | | |
|---|---|---|---|---|---|---|---|---|---|---|---|---|---|---|---|---|---|---|---|
| | | Epistemic | | | Aleatoric | | | Epistemic | | | Aleatoric | | | Epistemic | | | Aleatoric | | |
| | | Low | Med | High | Low | Med | High | Low | Med | High | Low | Med | High | Low | Med | High | Low | Med | High |
| CIFAR-10C | gaussian | 0.145 | 0.169 | **0.176** | 0.066 | 0.099 | **0.102** | 0.012 | **0.016** | **0.016** | **0.065** | 0.056 | 0.055 | 0.268 | 0.359 | 0.38 | 0.292 | 0.388 | 0.412 |
| | shot noise | 0.129 | 0.158 | **0.174** | 0.053 | 0.084 | **0.104** | 0.01 | 0.014 | **0.016** | **0.069** | 0.061 | 0.055 | 0.224 | 0.315 | 0.378 | 0.249 | 0.342 | 0.409 |
| | speckle noise | 0.128 | 0.158 | **0.174** | 0.053 | 0.081 | **0.102** | 0.01 | 0.014 | **0.016** | **0.069** | 0.061 | 0.055 | 0.224 | 0.31 | 0.375 | 0.247 | 0.336 | 0.407 |
| | impulse | 0.134 | 0.161 | **0.174** | 0.052 | 0.085 | **0.118** | 0.011 | 0.015 | **0.017** | **0.069** | 0.059 | 0.051 | 0.231 | 0.338 | 0.423 | 0.257 | 0.364 | 0.454 |
| | defocus blur | 0.105 | 0.129 | **0.176** | 0.032 | 0.048 | **0.095** | 0.008 | 0.01 | **0.016** | **0.076** | 0.071 | 0.057 | 0.154 | 0.203 | 0.325 | 0.185 | 0.25 | 0.434 |
| | gaussian blur | 0.105 | 0.154 | **0.186** | 0.032 | 0.069 | **0.11** | 0.008 | 0.013 | **0.017** | **0.076** | 0.064 | 0.053 | 0.154 | 0.258 | 0.348 | 0.185 | 0.329 | 0.497 |
| | motion blur | 0.137 | 0.166 | **0.175** | 0.055 | 0.085 | **0.1** | 0.011 | 0.015 | **0.016** | **0.068** | 0.06 | 0.056 | 0.227 | 0.322 | 0.36 | 0.268 | 0.385 | 0.446 |
| | zoom blur | 0.148 | 0.161 | **0.176** | 0.063 | 0.077 | **0.1** | 0.012 | 0.014 | **0.016** | **0.066** | 0.062 | 0.056 | 0.253 | 0.287 | 0.338 | 0.318 | 0.365 | 0.457 |
| | snow | 0.131 | 0.157 | **0.163** | 0.05 | 0.077 | **0.086** | 0.01 | 0.014 | **0.015** | **0.07** | 0.062 | 0.059 | 0.221 | 0.31 | 0.337 | 0.242 | 0.326 | 0.355 |
| | fog | 0.104 | 0.122 | **0.154** | 0.033 | 0.045 | **0.09** | 0.008 | 0.009 | **0.013** | **0.076** | 0.072 | 0.06 | 0.155 | 0.199 | 0.321 | 0.187 | 0.246 | 0.397 |
| | brightness | 0.103 | 0.108 | **0.124** | 0.031 | 0.034 | **0.043** | 0.008 | 0.008 | **0.01** | **0.076** | 0.075 | 0.072 | 0.153 | 0.164 | 0.195 | 0.179 | 0.188 | 0.225 |
| | contrast | 0.107 | 0.145 | **0.165** | 0.035 | 0.066 | **0.148** | 0.008 | 0.012 | **0.017** | **0.075** | 0.065 | 0.046 | 0.162 | 0.265 | 0.419 | 0.197 | 0.342 | 0.69 |
| | elastic | 0.134 | 0.144 | **0.168** | 0.053 | 0.059 | **0.089** | 0.011 | 0.012 | **0.015** | **0.069** | 0.067 | 0.058 | 0.219 | 0.238 | 0.33 | 0.273 | 0.297 | 0.38 |
| | pixelate | 0.116 | 0.135 | **0.162** | 0.042 | 0.065 | **0.096** | 0.009 | 0.011 | **0.015** | **0.073** | 0.066 | 0.058 | 0.191 | 0.259 | 0.342 | 0.215 | 0.275 | 0.38 |
| | jpeg | 0.13 | 0.147 | **0.156** | 0.049 | 0.064 | **0.075** | 0.01 | 0.012 | **0.013** | **0.07** | 0.066 | 0.063 | 0.217 | 0.262 | 0.296 | 0.246 | 0.293 | 0.331 |
| | spatter | 0.117 | 0.145 | **0.147** | 0.041 | **0.065** | 0.062 | 0.009 | **0.012** | **0.012** | **0.073** | 0.065 | 0.066 | 0.191 | 0.279 | 0.266 | 0.216 | 0.299 | 0.291 |
| | saturate | 0.112 | 0.113 | **0.131** | 0.041 | 0.039 | **0.046** | 0.008 | 0.009 | **0.011** | 0.073 | **0.074** | 0.07 | 0.183 | 0.183 | 0.219 | 0.212 | 0.212 | 0.257 |
| | frost | 0.124 | 0.153 | **0.166** | 0.047 | 0.075 | **0.099** | 0.01 | 0.013 | **0.015** | **0.071** | 0.063 | 0.056 | 0.208 | 0.298 | 0.361 | 0.23 | 0.326 | 0.403 |

| | | Epistemic | Aleatoric | Epistemic | Aleatoric | Epistemic | Aleatoric |
|---|---|---|---|---|---|---|---|
| Dataset | In Dist-Clean | 0.102 | 0.031 | 0.008 | 0.076 | 0.151 | 0.179 |
| | MNIST | 0.184 (1.801) | 0.142 (4.626) | 0.021 (2.723) | 0.044 (0.572) | 0.445 (2.95) | 0.772 (4.312) |
| | Fashion MNIST | 0.183 (1.791) | 0.147 (4.808) | 0.022 (2.822) | 0.042 (0.557) | 0.431 (2.861) | 0.811 (4.53) |
| | SVHN | 0.183 (1.789) | 0.141 (4.606) | 0.019 (2.421) | 0.046 (0.608) | 0.394 (2.612) | 0.647 (3.616) |

Table 8: CIFAR-10 Results. The 4 BNNs trained have the following accuracies : $90, 89, 90, 89$ in percentage terms and rounded to the nearest whole number. For different categories of corruptions, increasing severity leads to higher levels of aleatoric uncertainty for CBDL. When exposed to completely unseen datasets, this reaches its peak. In contrast, the Ensemble has a reverse trend. For the single BNN, the network with the highest accuracy was selected.

## N.2  MNIST Results

| | | CBDL | | | | | | Ensemble | | | | | | BNN | | | | | |
|---|---|---|---|---|---|---|---|---|---|---|---|---|---|---|---|---|---|---|---|
| | | Epistemic | | | Aleatoric | | | Epistemic | | | Aleatoric | | | Epistemic | | | Aleatoric | | |
| | | Low | Med | High | Low | Med | High | Low | Med | High | Low | Med | High | Low | Med | High | Low | Med | High |
| MNIST-C | brightness | 0.21 | **0.246** | 0.195 | 0.128 | 0.087 | **0.158** | 0.023 | **0.026** | 0.024 | 0.045 | **0.05** | 0.045 | 0.323 | 0.599 | 0.767 | 0.11 | 0.229 | 0.301 |
| | canny edges | **0.162** | 0.158 | **0.162** | 0.174 | **0.176** | 0.174 | **0.016** | 0.015 | **0.016** | **0.042** | **0.042** | **0.042** | 0.641 | 0.658 | 0.651 | 0.296 | 0.289 | 0.296 |
| | dotted line | **0.18** | 0.173 | 0.176 | 0.097 | **0.102** | **0.102** | **0.014** | 0.013 | **0.014** | **0.055** | **0.055** | **0.055** | 0.375 | 0.377 | 0.376 | 0.153 | 0.152 | 0.15 |
| | fog | 0.185 | 0.187 | **0.19** | 0.163 | **0.167** | 0.166 | **0.023** | **0.023** | **0.023** | **0.043** | 0.042 | **0.043** | 0.749 | 0.827 | 0.826 | 0.326 | 0.359 | 0.361 |
| | glass blur | **0.161** | 0.145 | 0.155 | 0.181 | **0.206** | 0.2 | 0.016 | 0.016 | **0.02** | **0.038** | 0.033 | 0.033 | 0.634 | 0.706 | 0.751 | 0.323 | 0.389 | 0.386 |
| | impulse noise | **0.186** | 0.172 | 0.181 | 0.083 | 0.132 | **0.158** | 0.013 | 0.015 | **0.021** | **0.057** | 0.049 | 0.042 | 0.297 | 0.469 | 0.766 | 0.122 | 0.219 | 0.379 |
| | motion blur | **0.184** | 0.161 | 0.157 | 0.123 | 0.185 | **0.195** | 0.016 | 0.018 | **0.019** | **0.049** | 0.035 | 0.029 | 0.404 | 0.765 | 0.918 | 0.188 | 0.449 | 0.542 |
| | rotate | **0.189** | 0.167 | 0.142 | 0.072 | 0.134 | **0.196** | 0.013 | 0.014 | **0.015** | **0.059** | 0.048 | 0.033 | v0.247 | 0.451 | 0.749 | 0.096 | 0.211 | 0.379 |
| | scale | **0.196** | 0.169 | 0.121 | 0.08 | 0.152 | **0.232** | 0.014 | **0.015** | 0.013 | **0.057** | 0.044 | 0.025 | 0.269 | 0.478 | 0.859 | 0.103 | 0.217 | 0.541 |
| | shear | **0.188** | 0.177 | 0.153 | 0.065 | 0.102 | **0.184** | 0.012 | 0.014 | **0.017** | **0.061** | 0.055 | 0.036 | 0.219 | 0.346 | 0.736 | 0.082 | 0.147 | 0.366 |
| | shot noise | **0.188** | 0.179 | 0.18 | 0.061 | 0.08 | **0.113** | 0.012 | 0.012 | **0.014** | **0.062** | 0.059 | 0.052 | 0.213 | 0.247 | 0.376 | 0.078 | 0.091 | 0.162 |
| | spatter | **0.186** | 0.174 | 0.176 | 0.074 | **0.132** | 0.127 | 0.013 | **0.016** | **0.016** | **0.059** | 0.047 | 0.049 | 0.247 | 0.455 | 0.421 | 0.098 | 0.226 | 0.195 |
| | stripe | 0.18 | 0.182 | **0.184** | **0.165** | 0.163 | 0.161 | **0.021** | 0.02 | **0.021** | 0.04 | **0.04** | **0.04** | 0.745 | 0.76 | 0.76 | 0.225 | 0.22 | 0.229 |
| | translate | 0.191 | **0.192** | **0.192** | 0.061 | 0.086 | **0.128** | 0.013 | 0.015 | **0.019** | **0.061** | 0.056 | 0.046 | 0.214 | 0.292 | 0.474 | 0.078 | 0.112 | 0.204 |
| | zigzag | **0.18** | 0.176 | 0.179 | 0.119 | **0.123** | 0.122 | **0.016** | **0.016** | **0.016** | **0.051** | 0.05 | 0.05 | 0.476 | 0.477 | 0.476 | 0.201 | 0.202 | 0.201 |

| | | Epistemic | Aleatoric | Epistemic | Aleatoric | Epistemic | Aleatoric |
|---|---|---|---|---|---|---|---|
| Dataset | In Dist-Clean | 0.188 | 0.057 | 0.012 | 0.063 | 0.198 | 0.071 |
| | CIFAR | 0.185 (0.986) | 0.167 (2.94) | 0.023 (1.948) | 0.042 (0.67) | 0.821 (4.155) | 0.372 (5.124) |
| | Fashion MNIST | 0.162 (0.866) | 0.187 (3.287) | 0.019 (1.618) | 0.035 (0.564) | 0.919 (4.654) | 0.461 (6.496) |
| | SVHN | 0.190 (1.015) | 0.168 (2.963) | 0.022 (1.847) | 0.043 (0.69) | 0.923 (4.674) | 0.297 (4.183) |

Table 9: The 4 BNNs trained have the following accuracy: $99, 99, 99, 98$ in percentage terms and rounded to the nearest whole number. These are the probabilities of the most likely label to be the correct one according to the 4 different BNNs. For the single BNN, the network with the highest accuracy was selected. For different categories of corruptions, increasing severity leads to higher levels of aleatoric uncertainty for IBNNs. When exposed to completely unseen datasets, this gets close to the highest aleatoric uncertainty. The same is not true for EBNN. The epistemic uncertainty for IBNNs shows a less consistent trend in this case.

## N.3 SVHN Results

| | | CBDL | | | | | | Ensemble | | | | | | BNN | | | | | |
|---|---|---|---|---|---|---|---|---|---|---|---|---|---|---|---|---|---|---|---|
| | | Epistemic | | | Aleatoric | | | Epistemic | | | Aleatoric | | | Epistemic | | | Aleatoric | | |
| | | Low | Med | High | Low | Med | High | Low | Med | High | Low | Med | High | Low | Med | High | Low | Med | High |
| SVHN-C | brightness | 0.062 | 0.074 | **0.102** | 0.011 | 0.019 | **0.053** | 0.005 | 0.006 | **0.008** | **0.083** | 0.08 | 0.07 | 0.113 | 0.161 | 0.293 | 0.136 | 0.198 | 0.405 |
| | contrast | 0.077 | 0.099 | **0.143** | 0.022 | 0.048 | **0.174** | 0.006 | 0.008 | **0.018** | **0.08** | 0.072 | 0.04 | 0.17 | 0.261 | 0.37 | 0.247 | 0.487 | 1.212 |
| | defocus blur | 0.109 | **0.154** | 0.138 | 0.044 | 0.147 | **0.218** | 0.009 | 0.017 | **0.02** | **0.072** | 0.045 | 0.029 | 0.272 | 0.486 | 0.431 | 0.353 | 0.877 | 1.334 |
| | elastic | 0.182 | 0.177 | **0.192** | **0.14** | 0.122 | 0.102 | **0.023** | 0.021 | 0.022 | 0.04 | 0.046 | **0.05** | 0.562 | 0.495 | 0.489 | 0.84 | 0.779 | 0.639 |
| | fog | 0.157 | 0.171 | **0.181** | 0.095 | 0.124 | **0.135** | 0.016 | 0.02 | **0.023** | **0.056** | 0.047 | 0.042 | 0.448 | 0.534 | 0.599 | 0.602 | 0.709 | 0.704 |
| | frost | 0.089 | 0.118 | **0.131** | 0.026 | 0.052 | **0.063** | 0.007 | 0.01 | **0.012** | **0.078** | 0.069 | 0.066 | 0.2 | 0.319 | 0.372 | 0.238 | 0.381 | 0.442 |
| | gaussian blur | 0.068 | 0.131 | **0.14** | 0.017 | 0.087 | **0.208** | 0.005 | 0.012 | **0.019** | **0.081** | 0.06 | 0.032 | 0.143 | 0.38 | 0.546 | 0.176 | 0.603 | 1.151 |
| | gaussian noise | 0.116 | 0.163 | **0.182** | 0.031 | 0.061 | **0.103** | 0.01 | 0.018 | **0.025** | **0.074** | 0.061 | 0.044 | 0.251 | 0.429 | 0.655 | 0.251 | 0.41 | 0.586 |
| | impulse noise | 0.13 | 0.169 | **0.185** | 0.035 | 0.06 | **0.102** | 0.012 | 0.018 | **0.026** | **0.072** | 0.061 | 0.045 | 0.3 | 0.449 | 0.653 | 0.296 | 0.43 | 0.589 |
| | jpeg | 0.07 | 0.083 | **0.129** | 0.014 | 0.018 | **0.041** | 0.005 | 0.006 | **0.012** | **0.082** | 0.08 | 0.071 | 0.124 | 0.153 | 0.287 | 0.134 | 0.166 | 0.306 |
| | motion blur | 0.1 | 0.157 | **0.167** | 0.027 | 0.078 | **0.149** | 0.008 | 0.015 | **0.02** | **0.077** | 0.06 | 0.041 | 0.233 | 0.438 | 0.605 | 0.272 | 0.568 | 0.889 |
| | pixelate | 0.061 | 0.123 | **0.18** | 0.011 | 0.036 | **0.095** | 0.004 | 0.011 | **0.02** | **0.083** | 0.073 | 0.052 | 0.102 | 0.222 | 0.516 | 0.114 | 0.254 | 0.591 |
| | saturate | 0.063 | 0.065 | **0.141** | 0.011 | 0.012 | **0.075** | 0.005 | 0.005 | **0.014** | **0.083** | 0.083 | 0.062 | 0.105 | 0.108 | 0.376 | 0.116 | 0.12 | 0.415 |
| | shot noise | 0.119 | 0.17 | **0.183** | 0.031 | 0.067 | **0.106** | 0.011 | 0.019 | **0.026** | **0.074** | 0.059 | 0.044 | 0.255 | 0.462 | 0.659 | 0.256 | 0.441 | 0.584 |
| | snow | 0.131 | **0.16** | **0.16** | 0.043 | 0.076 | **0.096** | 0.012 | **0.017** | **0.017** | **0.07** | 0.059 | 0.055 | 0.3 | 0.435 | 0.492 | 0.31 | 0.443 | 0.519 |
| | spatter | 0.088 | 0.127 | **0.163** | 0.021 | 0.037 | **0.06** | 0.007 | 0.012 | **0.017** | **0.079** | 0.072 | 0.063 | 0.169 | 0.277 | 0.404 | 0.191 | 0.301 | 0.403 |
| | speckle noise | 0.1 | 0.143 | **0.181** | 0.023 | 0.046 | **0.085** | 0.008 | 0.015 | **0.023** | **0.077** | 0.067 | 0.052 | 0.201 | 0.345 | 0.655 | 0.207 | 0.343 | 0.516 |
| | zoom blur | 0.061 | 0.061 | **0.064** | 0.011 | 0.012 | **0.013** | 0.004 | 0.004 | **0.005** | **0.083** | **0.083** | **0.083** | 0.105 | 0.108 | 0.121 | 0.117 | 0.123 | 0.136 |

| | | Epistemic | Aleatoric | Epistemic | Aleatoric | Epistemic | Aleatoric |
|---|---|---|---|---|---|---|---|
| Dataset | In Dist-Clean | 0.061 | 0.01 | 0.004 | 0.083 | 0.097 | 0.107 |
| | CIFAR | 0.181 (2.992) | 0.121 (11.778) | 0.026 (6.029) | 0.040 (0.479) | 0.655 (6.765) | 0.716(6.69) |
| | MNIST | 0.198 (3.272) | 0.063 (6.154) | 0.024 (5.495) | 0.056 (0.675) | 0.329 (3.393) | 0.336 (3.139) |
| | Fashion MNIST | 0.199 (3.288) | 0.113 (10.984) | 0.026 (6.002) | 0.041 (0.491) | 0.103 (6.375) | 0.125 (6.178) |

Table 10: The 4 BNNs trained have the following accuracy : $95, 95, 96, 95$ in percentage terms and rounded to the nearest whole number. These are the probabilities of the most likely label to be the correct one according to the 4 different BNNs. For the single BNN, the network with the highest accuracy was selected. For different categories of corruptions, increasing severity leads to higher levels of aleatoric uncertainty for IBNNs. When exposed to completely unseen datasets, this reaches its peak. The same is not true for EBNN. For the epistemic uncertainty as well, there is a clear trend with increasing corruption severity.

### N.4 Fashion MNIST

| | | CBDL | | | | | | Baseline | | | | | | BNN | | | | | |
|---|---|---|---|---|---|---|---|---|---|---|---|---|---|---|---|---|---|---|---|
| | | Epistemic | | | Aleatoric | | | Epistemic | | | Aleatoric | | | Epistemic | | | Aleatoric | | |
| | | Low | Med | High | Low | Med | High | Low | Med | High | Low | Med | High | Low | Med | High | Low | Med | High |
| Fashion MNIST-C | brightness | 0.199 | **0.212** | 0.205 | **0.108** | 0.106 | 0.094 | 0.021 | **0.024** | 0.022 | **0.051** | 0.048 | **0.051** | 0.568 | 0.519 | 0.475 | 0.261 | 0.294 | 0.262 |
| | canny edges | **0.18** | 0.175 | 0.177 | 0.157 | **0.164** | 0.16 | **0.021** | 0.02 | 0.02 | **0.043** | 0.042 | **0.043** | 0.729 | 0.734 | 0.753 | 0.216 | 0.215 | 0.217 |
| | dotted line | 0.167 | **0.168** | **0.168** | **0.051** | **0.051** | **0.051** | 0.012 | **0.013** | **0.013** | **0.067** | 0.066 | 0.066 | 0.336 | 0.349 | 0.36 | 0.127 | 0.131 | 0.129 |
| | fog | **0.19** | 0.185 | 0.187 | 0.128 | **0.132** | 0.129 | **0.021** | 0.02 | **0.021** | **0.046** | 0.045 | **0.046** | 0.489 | 0.463 | 0.448 | 0.264 | 0.25 | 0.237 |
| | glass blur | 0.189 | 0.189 | **0.193** | 0.114 | 0.127 | **0.133** | 0.017 | 0.019 | **0.022** | **0.052** | 0.048 | 0.045 | 0.517 | 0.641 | 0.777 | 0.209 | 0.274 | 0.354 |
| | impulse noise | 0.169 | 0.196 | **0.202** | 0.052 | 0.09 | **0.111** | 0.013 | 0.019 | **0.02** | **0.066** | 0.055 | 0.051 | 0.304 | 0.498 | 0.675 | 0.118 | 0.203 | 0.276 |
| | motion blur | **0.182** | 0.171 | 0.163 | 0.106 | 0.157 | **0.174** | 0.017 | **0.02** | 0.019 | **0.053** | 0.041 | 0.038 | 0.505 | 0.616 | 0.641 | 0.216 | 0.296 | 0.285 |
| | rotate | **0.185** | 0.173 | 0.162 | 0.074 | 0.164 | **0.184** | 0.016 | **0.02** | **0.02** | **0.06** | 0.037 | 0.035 | 0.392 | 0.754 | 0.797 | 0.162 | 0.338 | 0.341 |
| | scale | 0.154 | **0.172** | 0.127 | 0.054 | 0.123 | **0.223** | 0.01 | **0.015** | 0.014 | **0.067** | 0.05 | 0.028 | 0.302 | 0.657 | 0.917 | 0.11 | 0.209 | 0.398 |
| | shear | 0.172 | 0.177 | **0.182** | 0.063 | 0.13 | **0.144** | 0.014 | 0.019 | **0.02** | **0.063** | 0.045 | 0.039 | 0.333 | 0.609 | 0.719 | 0.134 | 0.257 | 0.279 |
| | shot noise | 0.16 | 0.178 | **0.197** | 0.046 | 0.059 | **0.088** | 0.01 | 0.013 | **0.017** | **0.068** | 0.064 | 0.056 | 0.268 | 0.344 | 0.49 | 0.097 | 0.121 | 0.17 |
| | spatter | 0.15 | 0.189 | **0.191** | 0.044 | **0.082** | 0.074 | 0.01 | **0.018** | 0.017 | **0.07** | 0.057 | 0.059 | 0.279 | 0.479 | 0.459 | 0.112 | 0.21 | 0.19 |
| | stripe | **0.21** | 0.208 | 0.208 | 0.125 | **0.126** | 0.125 | **0.025** | **0.025** | **0.025** | **0.041** | **0.041** | **0.041** | 0.686 | 0.698 | 0.697 | 0.349 | 0.352 | 0.341 |
| | translate | 0.153 | **0.189** | 0.17 | 0.046 | 0.094 | **0.158** | 0.01 | 0.018 | **0.019** | **0.069** | 0.054 | 0.041 | 0.27 | 0.499 | 0.721 | 0.102 | 0.21 | 0.317 |
| | zigzag | 0.187 | **0.19** | 0.189 | 0.065 | 0.065 | **0.066** | **0.016** | **0.016** | **0.016** | **0.061** | **0.061** | **0.061** | 0.39 | 0.404 | 0.411 | 0.151 | 0.154 | 0.152 |

| | | Epistemic | Aleatoric | Epistemic | Aleatoric | Epistemic | Aleatoric |
|---|---|---|---|---|---|---|---|
| Dataset | In Dist-Clean | 0.121 | 0.029 | 0.007 | 0.075 | 0.197 | 0.351 |
| | CIFAR | 0.205 (1.705) | 0.104 (3.561) | 0.022 (3.178) | 0.050 (0.667) | 0.408 (2.073) | 0.221 (0.629) |
| | MNIST | 0.165 (1.367) | 0.182 (6.192) | 0.021 (3.069) | 0.033 (0.443) | 0.909 (4.618) | 0.328 (0.932) |
| | SVHN | 0.218 (1.81) | 0.109 (3.742) | 0.026 (3.827) | 0.046 (0.618) | 0.598 (3.036) | 0.305 (0.867) |

Table 11: The 4 BNNs trained have the following accuracies : 93, 92, 92, 92 in percentage terms and rounded to the nearest whole number. These are the probabilities of the most likely label to be the correct one according to the 4 different BNNs. For the single BNN, the network with the highest accuracy was selected. For different categories of corruptions, increasing severity leads to higher levels of aleatoric uncertainty for IBNNs. Which is high when exposed to completely unseen datasets as well. The same is not true for EBNN.

# O   Evaluation Summary by Noise

## O.1   CIFAR-10 Results

| | | BNN | Ensemble | CBDL |
|---|---|---|---|---|
| | gaussian noise | $0.613 \pm 0.116$ | $0.703 \pm 0.116$ | $\mathbf{0.715 \pm 0.168}$ |
| | shot noise | $0.669 \pm 0.077$ | $\mathbf{0.794 \pm 0.077}$ | $0.793 \pm 0.144$ |
| | impulse noise | $0.66 \pm 0.132$ | $0.776 \pm 0.132$ | $\mathbf{0.791 \pm 0.191}$ |
| | speckle noise | $0.683 \pm 0.057$ | $0.829 \pm 0.057$ | $\mathbf{0.839 \pm 0.115}$ |
| | gaussian blur | $0.741 \pm 0.073$ | $\mathbf{0.836 \pm 0.073}$ | $0.825 \pm 0.191$ |
| | defocus blur | $0.756 \pm 0.044$ | $\mathbf{0.919 \pm 0.044}$ | $0.912 \pm 0.105$ |
| | motion blur | $0.813 \pm 0.036$ | $\mathbf{0.902 \pm 0.036}$ | $0.889 \pm 0.067$ |
| | zoom blur | $0.78 \pm 0.029$ | $\mathbf{0.899 \pm 0.029}$ | $0.888 \pm 0.066$ |
| CIFAR-10 | fog | $0.753 \pm 0.034$ | $\mathbf{0.946 \pm 0.034}$ | $0.94 \pm 0.07$ |
| | frost | $0.745 \pm 0.025$ | $\mathbf{0.884 \pm 0.025}$ | $0.884 \pm 0.062$ |
| | snow | $0.763 \pm 0.039$ | $0.927 \pm 0.039$ | $\mathbf{0.929 \pm 0.044}$ |
| | spatter | $0.773 \pm 0.036$ | $0.955 \pm 0.036$ | $\mathbf{0.956 \pm 0.019}$ |
| | contrast | $0.665 \pm 0.155$ | $\mathbf{0.785 \pm 0.155}$ | $0.782 \pm 0.234$ |
| | brightness | $0.737 \pm 0.025$ | $\mathbf{0.991 \pm 0.025}$ | $0.99 \pm 0.004$ |
| | saturate | $0.751 \pm 0.019$ | $\mathbf{0.984 \pm 0.019}$ | $0.983 \pm 0.009$ |
| | jpeg | $0.775 \pm 0.011$ | $\mathbf{0.94 \pm 0.011}$ | $0.937 \pm 0.023$ |
| | pixelate | $0.685 \pm 0.097$ | $0.848 \pm 0.097$ | $\mathbf{0.855 \pm 0.132}$ |
| | elastic | $0.804 \pm 0.021$ | $\mathbf{0.952 \pm 0.021}$ | $0.95 \pm 0.024$ |

Table 12: Results for CIFAR-10C, organized by noise. For different categories of corruptions, increasing severity leads to higher levels of aleatoric uncertainty for CBDL. When exposed to completely unseen datasets, this reaches its peak. In contrast, the baseline has a reverse trend.

## O.2   MNIST Results

| | | BNN | Ensemble | CBDL |
|---|---|---|---|---|
| | shot noise | $0.597 \pm 0.07$ | $0.999 \pm 0.07$ | $\mathbf{0.999 \pm 0.001}$ |
| | impulse noise | $0.56 \pm 0.296$ | $\mathbf{0.716 \pm 0.296}$ | $0.7 \pm 0.396$ |
| | glass blur | $0.365 \pm 0.217$ | $0.406 \pm 0.217$ | $\mathbf{0.41 \pm 0.271}$ |
| | motion blur | $0.498 \pm 0.317$ | $0.65 \pm 0.317$ | $\mathbf{0.654 \pm 0.394}$ |
| | fog | $0.343 \pm 0.269$ | $0.364 \pm 0.269$ | $\mathbf{0.579 \pm 0.282}$ |
| | spatter | $0.714 \pm 0.089$ | $0.991 \pm 0.089$ | $\mathbf{0.991 \pm 0.007}$ |
| | brightness | $0.63 \pm 0.323$ | $0.803 \pm 0.323$ | $\mathbf{0.904 \pm 0.113}$ |
| | shear | $0.663 \pm 0.121$ | $\mathbf{0.916 \pm 0.121}$ | $0.915 \pm 0.168$ |
| MNIST | rotate | $0.64 \pm 0.066$ | $0.990 \pm 0.066$ | $\mathbf{0.999 \pm 0.0}$ |
| | scale | $0.618 \pm 0.32$ | $\mathbf{0.755 \pm 0.32}$ | $0.749 \pm 0.382$ |
| | translate | $0.694 \pm 0.142$ | $0.992 \pm 0.142$ | $\mathbf{0.987 \pm 0.023}$ |
| | dotted line | $0.728 \pm 0.019$ | $\mathbf{0.995 \pm 0.019}$ | $0.994 \pm 0.001$ |
| | zigzag | $0.777 \pm 0.023$ | $\mathbf{0.931 \pm 0.023}$ | $0.927 \pm 0.014$ |
| | stripe | $\mathbf{0.288 \pm 0.279}$ | $0.06 \pm 0.279$ | $0.106 \pm 0.08$ |
| | canny edges | $0.84 \pm 0.043$ | $0.849 \pm 0.043$ | $\mathbf{0.911 \pm 0.036}$ |

Table 13: Results for MNIST-C, organized by noise. For different categories of corruptions, increasing severity leads to higher levels of aleatoric uncertainty for CBDL. When exposed to completely unseen datasets, this reaches its peak. In contrast, the baseline has a reverse trend.

### O.3  Fashion MNIST Results

| | | BNN | Ensemble | CBDL |
|---|---|---|---|---|
| Fashion MNIST | shot noise | $0.538 \pm 0.134$ | $\mathbf{0.636 \pm 0.134}$ | $0.624 \pm 0.161$ |
| | impulse noise | $0.411 \pm 0.251$ | $0.511 \pm 0.251$ | $\mathbf{0.519 \pm 0.344}$ |
| | glass blur | $\mathbf{0.407 \pm 0.164}$ | $0.374 \pm 0.164$ | $0.385 \pm 0.227$ |
| | motion blur | $0.66 \pm 0.061$ | $0.928 \pm 0.061$ | $\mathbf{0.933 \pm 0.034}$ |
| | fog | $0.045 \pm 0.031$ | $0.122 \pm 0.031$ | $\mathbf{0.312 \pm 0.094}$ |
| | spatter | $0.665 \pm 0.043$ | $\mathbf{0.937 \pm 0.043}$ | $0.936 \pm 0.019$ |
| | brightness | $0.095 \pm 0.117$ | $0.273 \pm 0.117$ | $\mathbf{0.371 \pm 0.201}$ |
| | shear | $0.593 \pm 0.143$ | $0.691 \pm 0.143$ | $\mathbf{0.719 \pm 0.165}$ |
| | rotate | $0.45 \pm 0.277$ | $\mathbf{0.488 \pm 0.277}$ | $0.451 \pm 0.347$ |
| | scale | $\mathbf{0.437 \pm 0.259}$ | $0.38 \pm 0.259$ | $0.396 \pm 0.363$ |
| | translate | $0.586 \pm 0.147$ | $0.716 \pm 0.147$ | $\mathbf{0.752 \pm 0.178}$ |
| | dotted line | $0.661 \pm 0.024$ | $0.945 \pm 0.024$ | $\mathbf{0.947 \pm 0.01}$ |
| | zigzag | $0.669 \pm 0.019$ | $0.915 \pm 0.019$ | $\mathbf{0.919 \pm 0.015}$ |
| | stripe | $\mathbf{0.141 \pm 0.046}$ | $0.016 \pm 0.046$ | $0.071 \pm 0.049$ |
| | canny edges | $\mathbf{0.143 \pm 0.086}$ | $0.035 \pm 0.086$ | $0.062 \pm 0.023$ |

Table 14: Results for Fashion MNIST-C, organized by noise. For different categories of corruptions, increasing severity leads to higher levels of aleatoric uncertainty for CBDL. When exposed to completely unseen datasets, this reaches its peak. In contrast, the baseline has a reverse trend.

### O.4  SVHN Results

| | | BNN | Ensemble | CBDL |
|---|---|---|---|---|
| SVHN | Gaussian Noise | $0.475 \pm 0.304$ | $0.544 \pm 0.304$ | $\mathbf{0.549 \pm 0.339}$ |
| | Shot Noise | $0.447 \pm 0.298$ | $0.502 \pm 0.298$ | $\mathbf{0.504 \pm 0.332}$ |
| | Impulse Noise | $0.478 \pm 0.304$ | $0.522 \pm 0.304$ | $\mathbf{0.532 \pm 0.315}$ |
| | Speckle Noise | $0.586 \pm 0.211$ | $0.69 \pm 0.211$ | $\mathbf{0.695 \pm 0.28}$ |
| | Gaussian Blur | $\mathbf{0.634 \pm 0.325}$ | $0.627 \pm 0.325$ | $0.62 \pm 0.39$ |
| | Defocus Blur | $\mathbf{0.374 \pm 0.361}$ | $0.363 \pm 0.361$ | $0.351 \pm 0.405$ |
| | Motion Blur | $0.765 \pm 0.094$ | $\mathbf{0.803 \pm 0.094}$ | $0.795 \pm 0.133$ |
| | Zoom Blur | $0.675 \pm 0.02$ | $0.998 \pm 0.02$ | $\mathbf{0.998 \pm 0.002}$ |
| | Fog | $\mathbf{0.432 \pm 0.264}$ | $0.363 \pm 0.264$ | $0.34 \pm 0.246$ |
| | Frost | $0.79 \pm 0.032$ | $\mathbf{0.952 \pm 0.032}$ | $0.945 \pm 0.035$ |
| | Snow | $0.738 \pm 0.034$ | $\mathbf{0.779 \pm 0.034}$ | $0.775 \pm 0.08$ |
| | Spatter | $0.714 \pm 0.093$ | $\mathbf{0.807 \pm 0.093}$ | $0.798 \pm 0.188$ |
| | Contrast | $\mathbf{0.852 \pm 0.115}$ | $0.781 \pm 0.115$ | $0.803 \pm 0.284$ |
| | Brightness | $0.715 \pm 0.067$ | $\mathbf{0.992 \pm 0.067}$ | $0.989 \pm 0.016$ |
| | Saturate | $0.707 \pm 0.078$ | $\mathbf{0.93 \pm 0.078}$ | $0.916 \pm 0.14$ |
| | JPEG | $0.744 \pm 0.051$ | $\mathbf{0.951 \pm 0.051}$ | $0.949 \pm 0.073$ |
| | Pixelate | $0.577 \pm 0.275$ | $0.721 \pm 0.275$ | $\mathbf{0.729 \pm 0.35}$ |
| | Elastic | $\mathbf{0.274 \pm 0.23}$ | $0.213 \pm 0.23$ | $0.227 \pm 0.208$ |

Table 15: Results for SVHN-C, organized by noise. For different categories of corruptions, increasing severity leads to higher levels of aleatoric uncertainty for CBDL. When exposed to completely unseen datasets, this reaches its peak. In contrast, the baseline has a reverse trend.

# P    Accuracy vs Rejection Rate – All Results

## P.1    CIFAR10-C

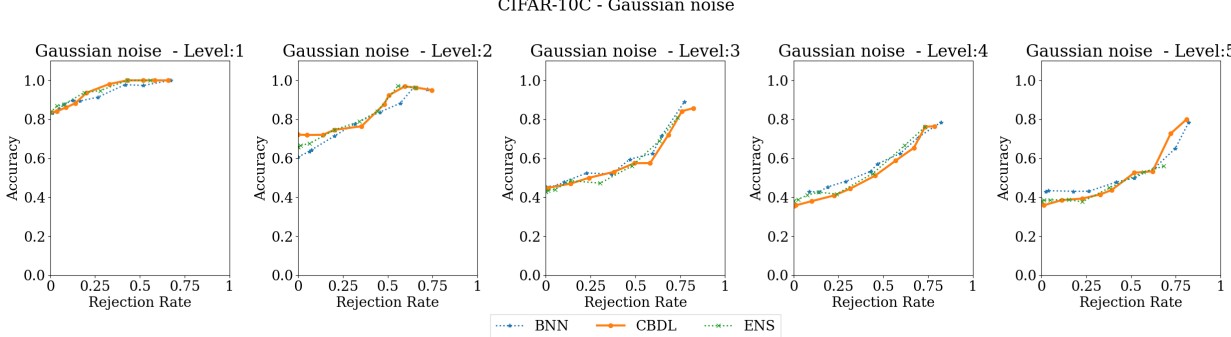

Figure 12: Accuracy vs Rejection Rate - CIFAR10C gaussian noise.

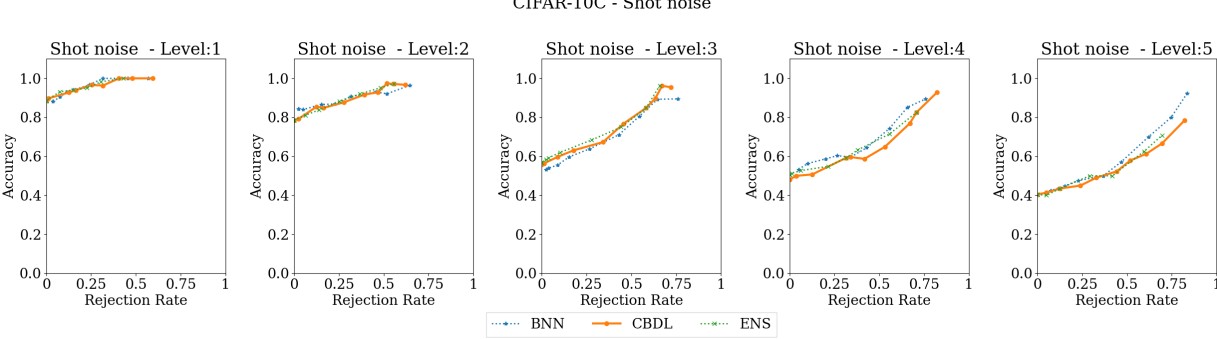

Figure 13: Accuracy vs Rejection Rate - CIFAR10C shot noise.

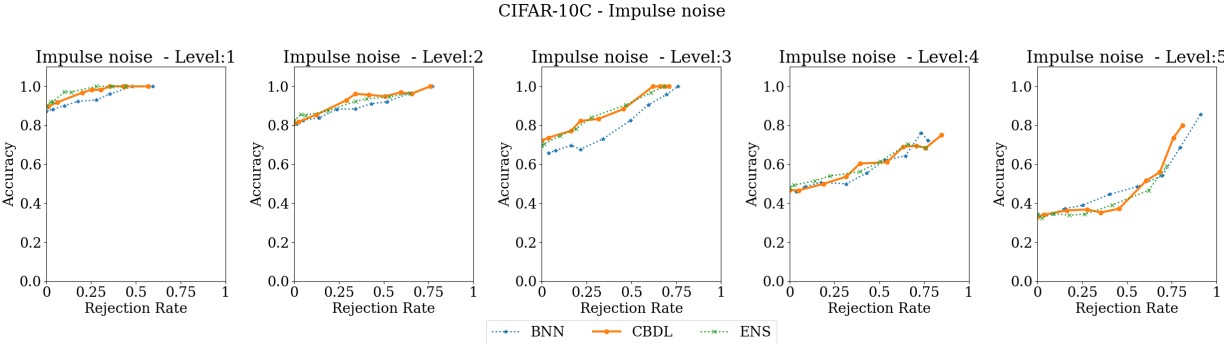

Figure 14: Accuracy vs Rejection Rate - CIFAR10C impulse noise.

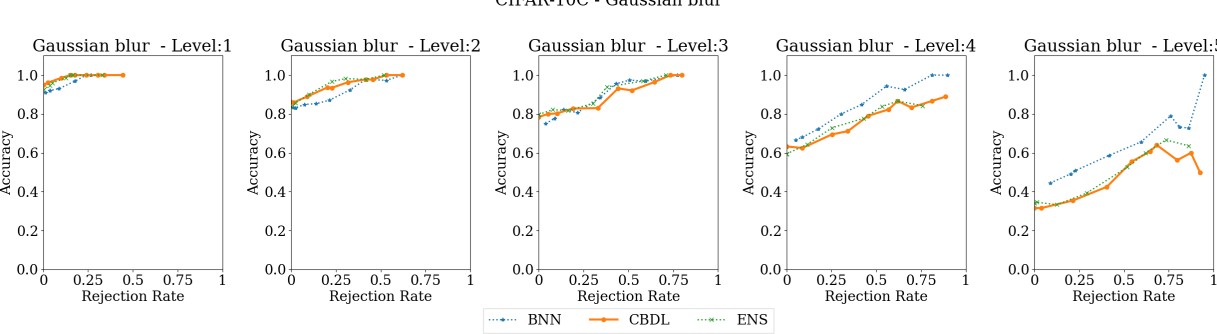

Figure 15: Accuracy vs Rejection Rate - CIFAR10C speckle noise.

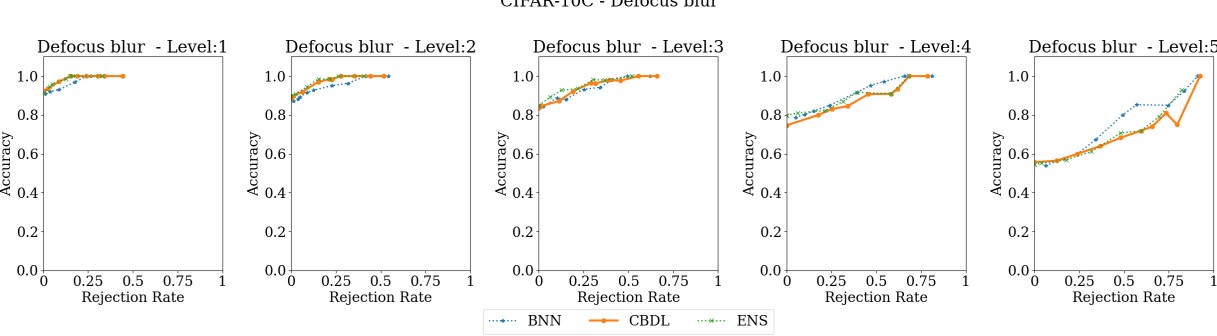

Figure 16: Accuracy vs Rejection Rate - CIFAR10C gaussian blur.

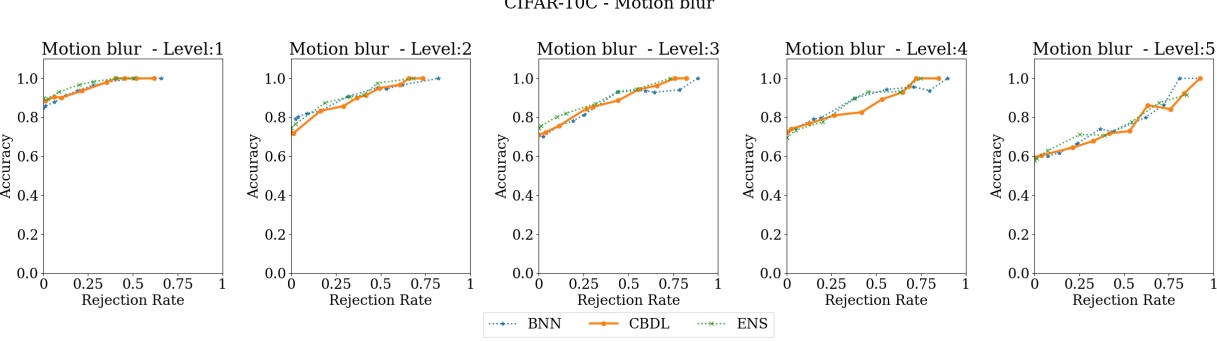

Figure 17: Accuracy vs Rejection Rate - CIFAR10C defocus blur.

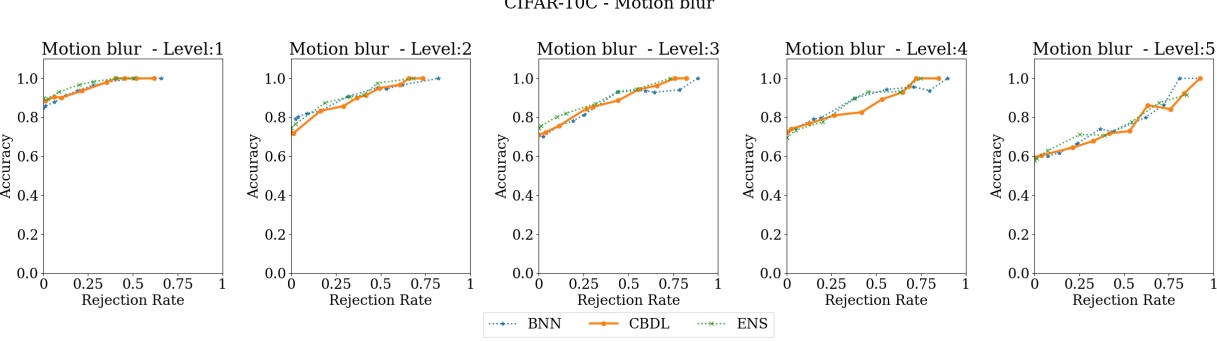

Figure 18: Accuracy vs Rejection Rate - CIFAR10C motion blur.

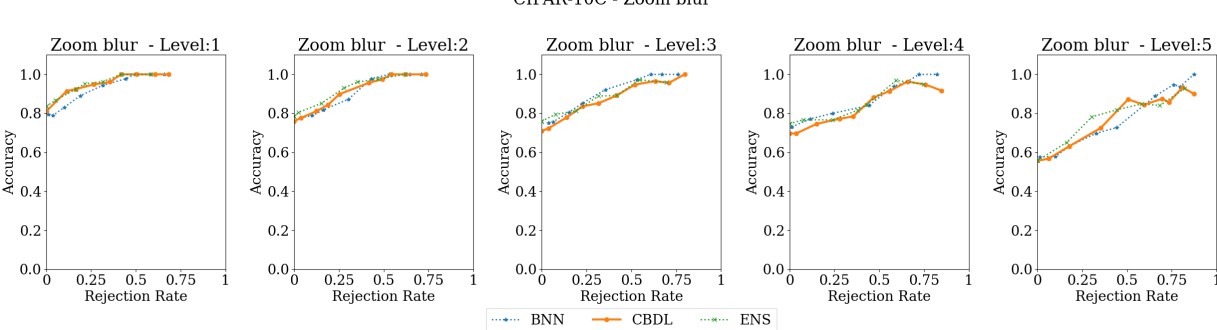

Figure 19: Accuracy vs Rejection Rate - CIFAR10C zoom blur.

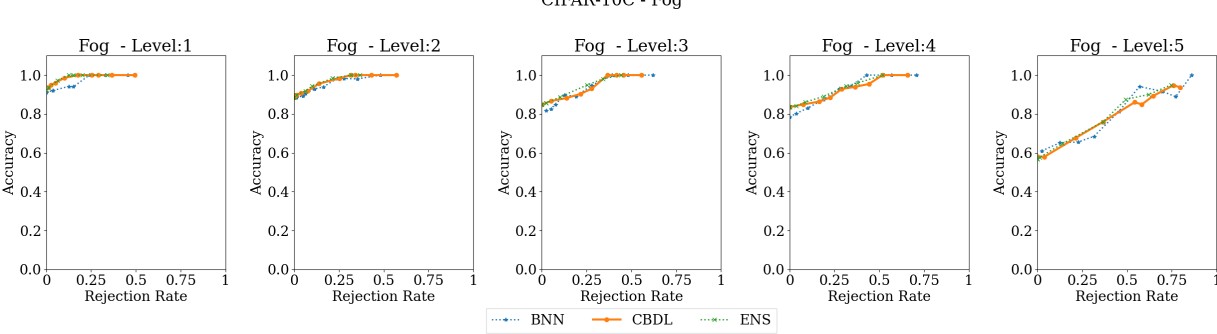

Figure 20: Accuracy vs Rejection Rate - CIFAR10C fog.

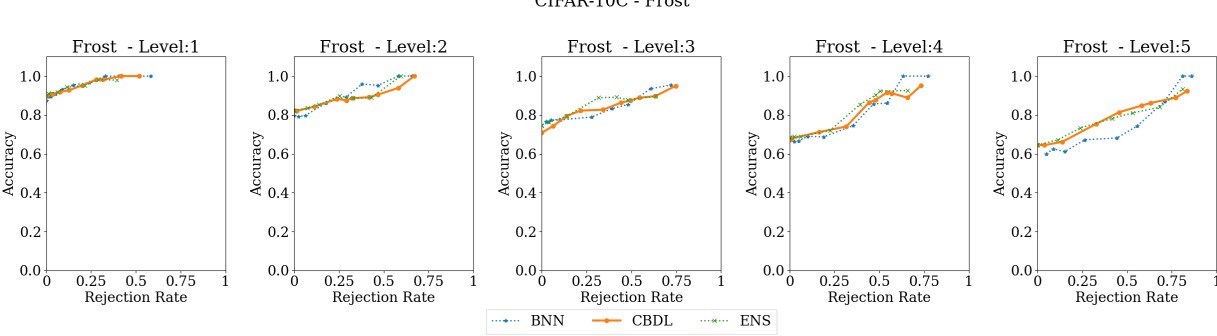

Figure 21: Accuracy vs Rejection Rate - CIFAR10C frost.

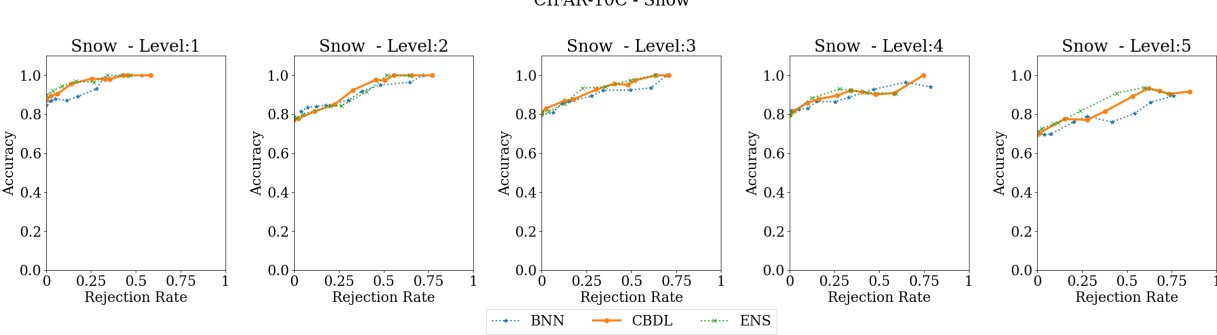

Figure 22: Accuracy vs Rejection Rate - CIFAR10C snow.

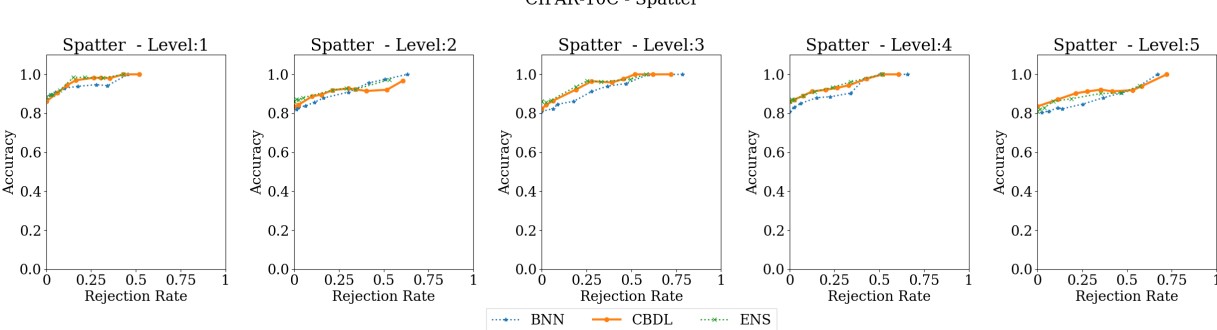

Figure 23: Accuracy vs Rejection Rate - CIFAR10C spatter.

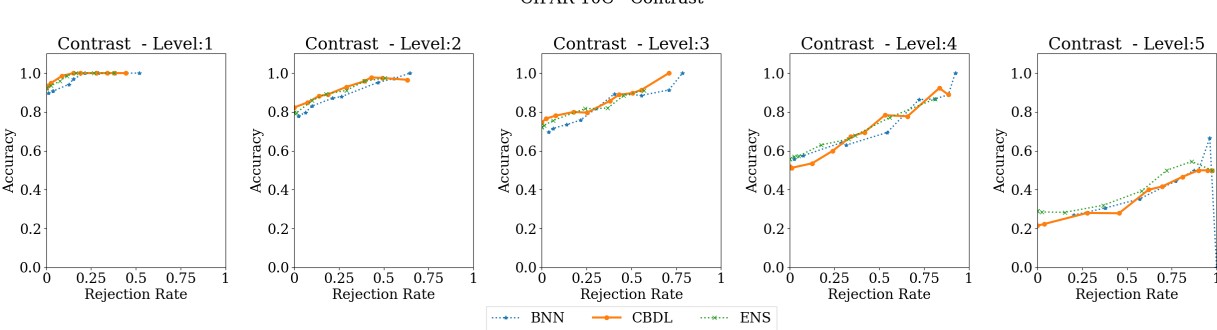

Figure 24: Accuracy vs Rejection Rate - CIFAR10C contrast.

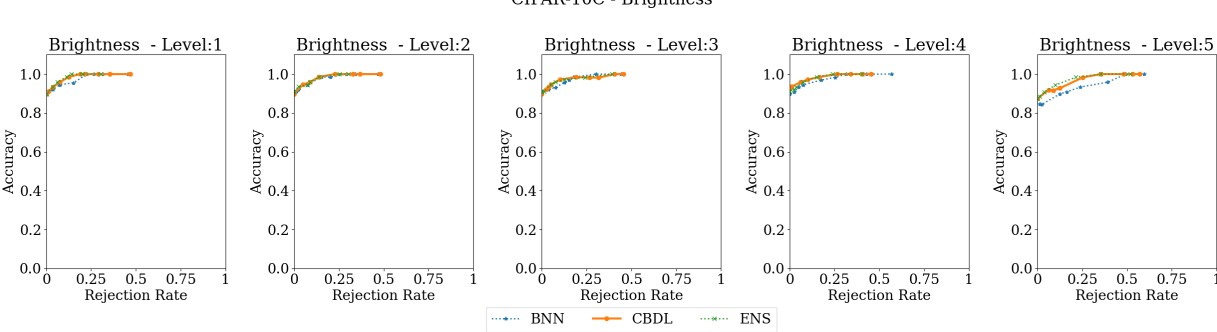

Figure 25: Accuracy vs Rejection Rate - CIFAR10C brightness.

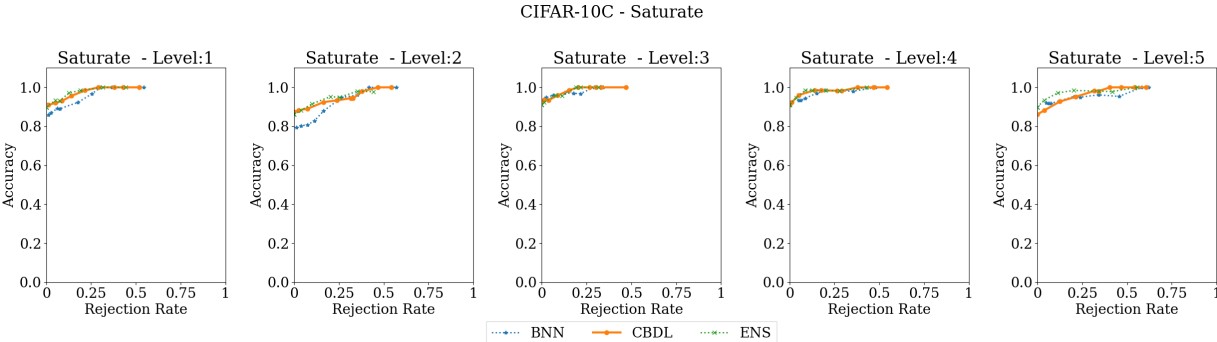

Figure 26: Accuracy vs Rejection Rate - CIFAR10C saturate.

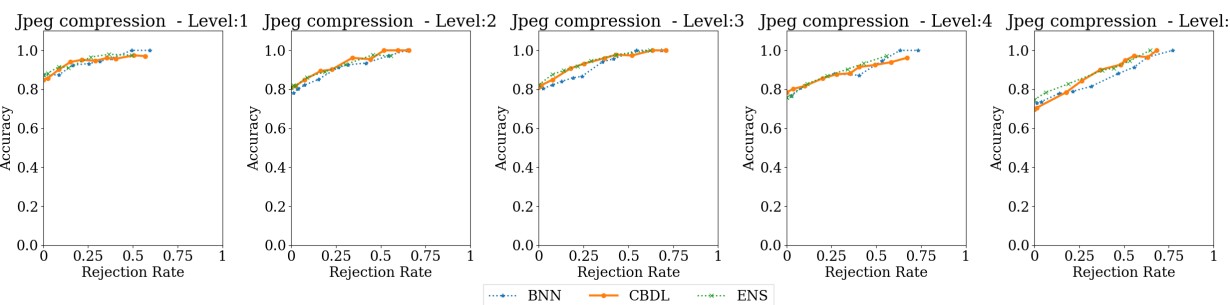

Figure 27: Accuracy vs Rejection Rate - CIFAR10C jpeg compression.

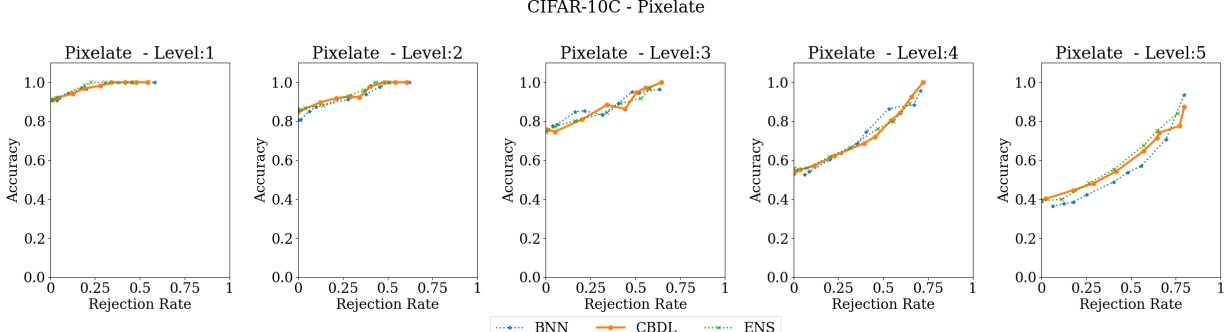

Figure 28: Accuracy vs Rejection Rate - CIFAR10C pixelate.

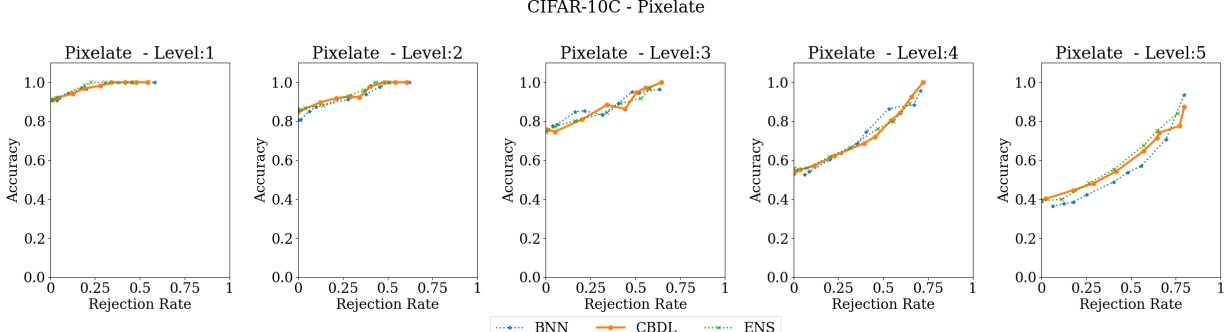

Figure 29: Accuracy vs Rejection Rate - CIFAR10C elastic transform.

## P.2 MNIST-C

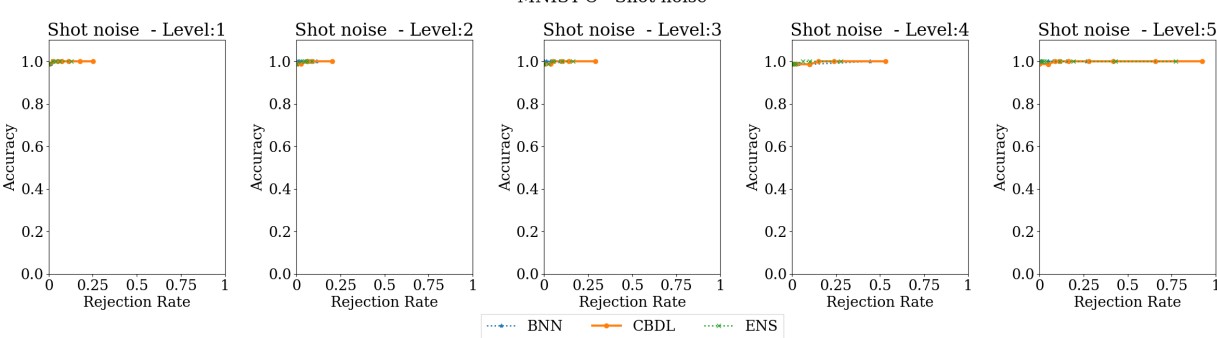

Figure 30: Accuracy vs Rejection Rate - MNISTC shot noise.

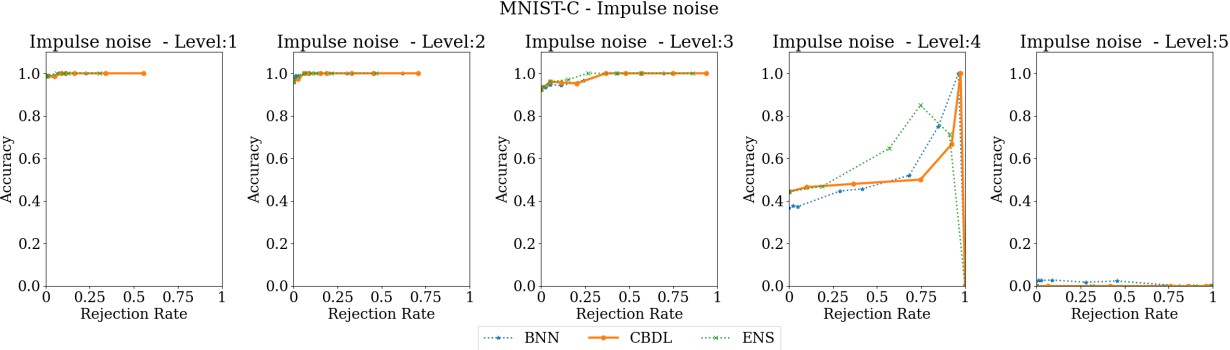

Figure 31: Accuracy vs Rejection Rate - MNISTC impulse noise.

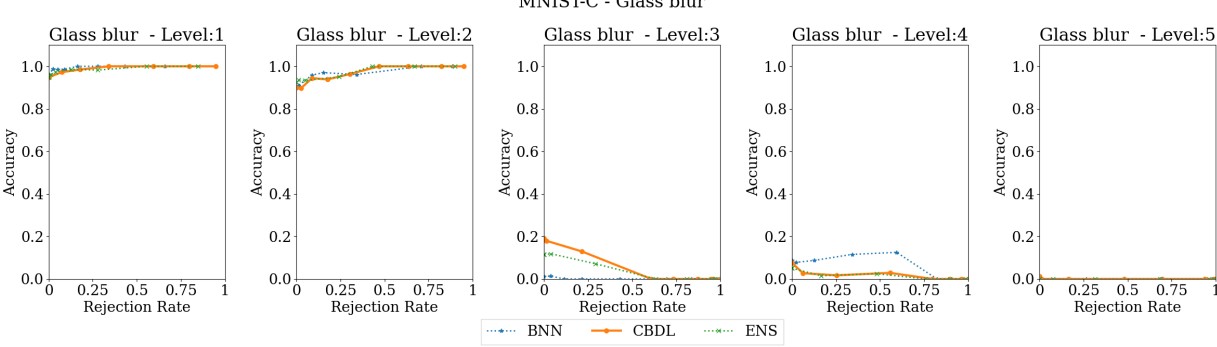

Figure 32: Accuracy vs Rejection Rate - MNISTC glass blur.

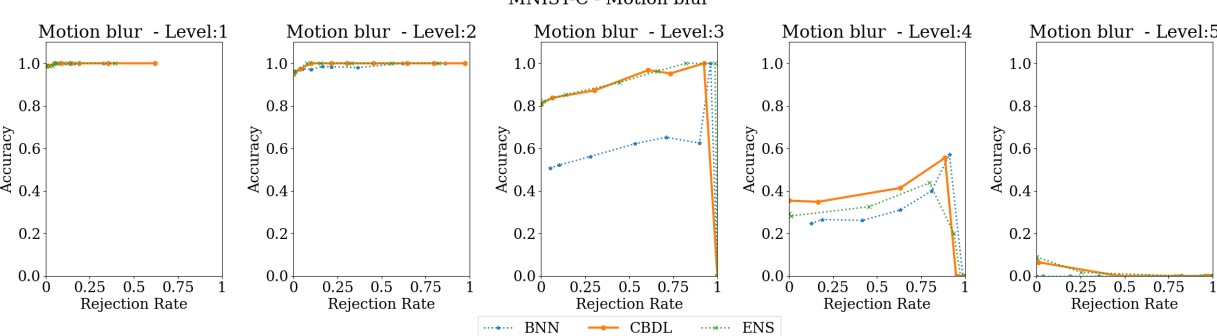

Figure 33: Accuracy vs Rejection Rate - MNISTC motion blur.

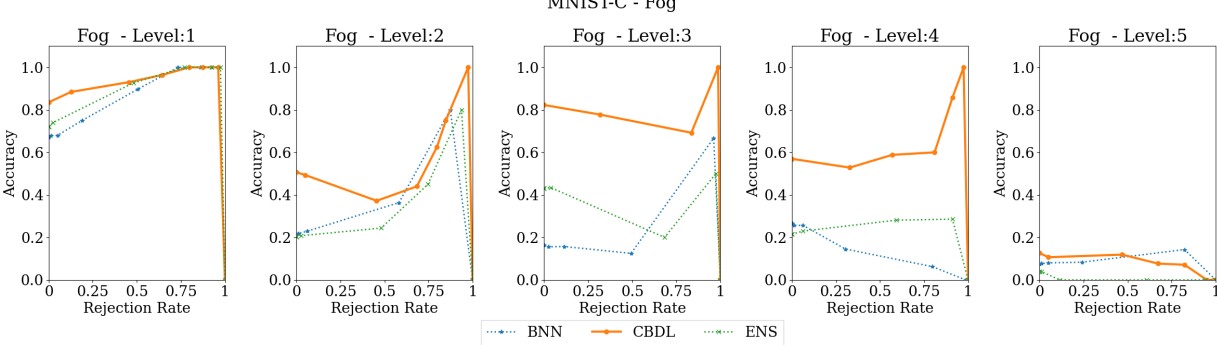

Figure 34: Accuracy vs Rejection Rate - MNISTC fog.

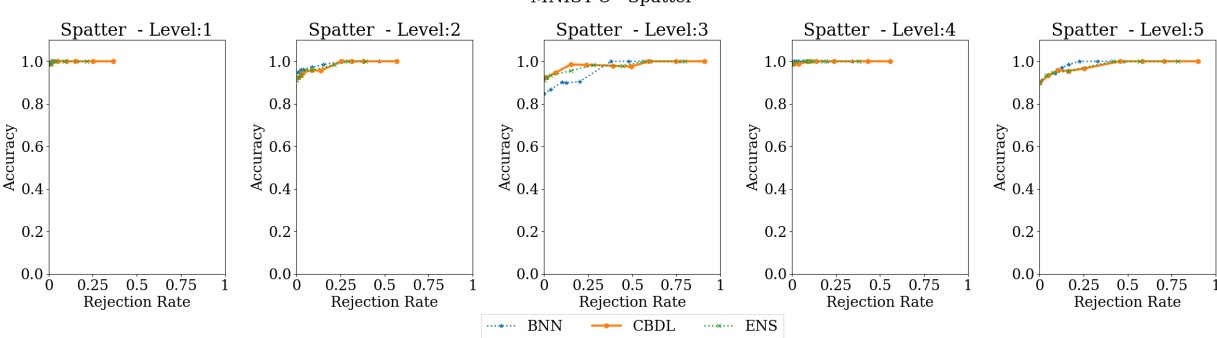

Figure 35: Accuracy vs Rejection Rate - MNISTC spatter.

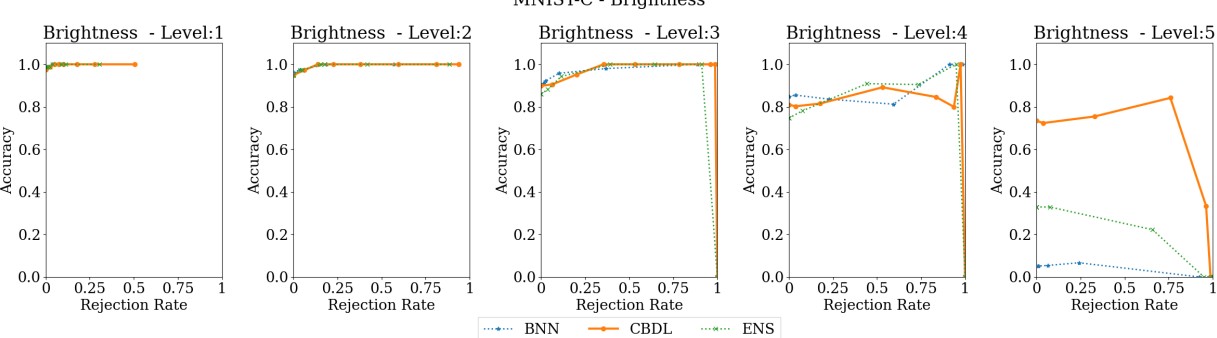

Figure 36: Accuracy vs Rejection Rate - MNISTC brightness.

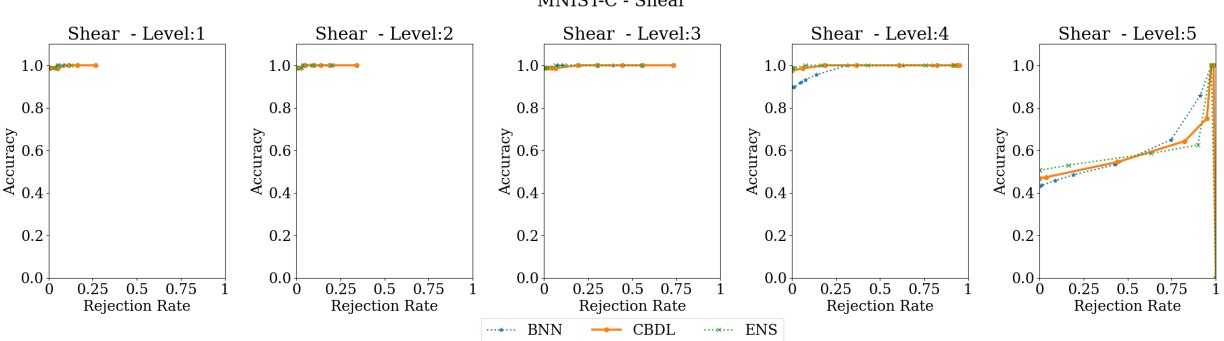

Figure 37: Accuracy vs Rejection Rate - MNISTC shear.

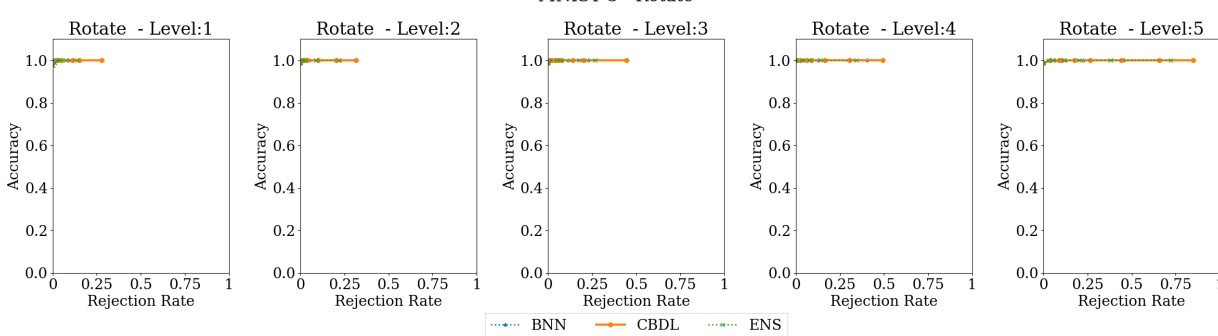

Figure 38: Accuracy vs Rejection Rate - MNISTC rotate.

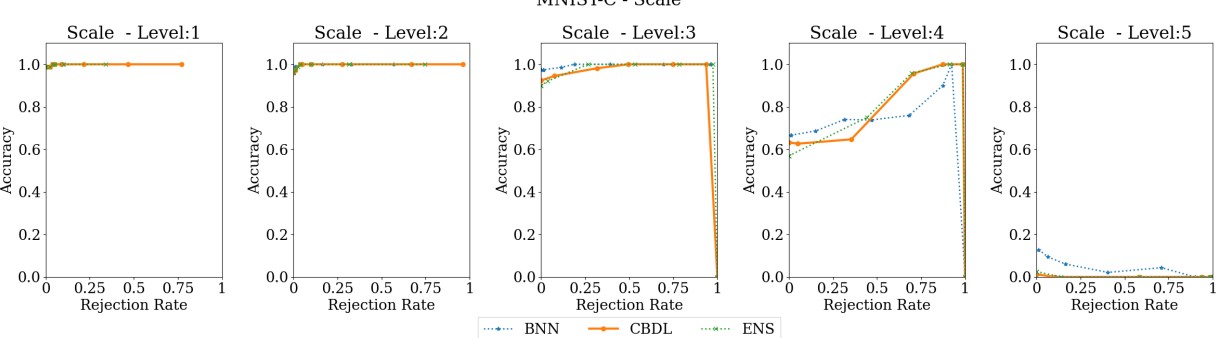

Figure 39: Accuracy vs Rejection Rate - MNISTC scale.

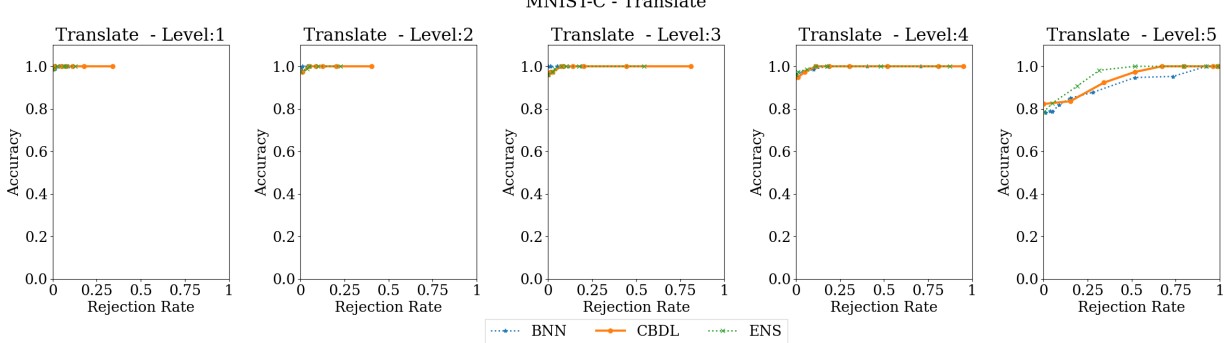

Figure 40: Accuracy vs Rejection Rate - MNISTC translate.

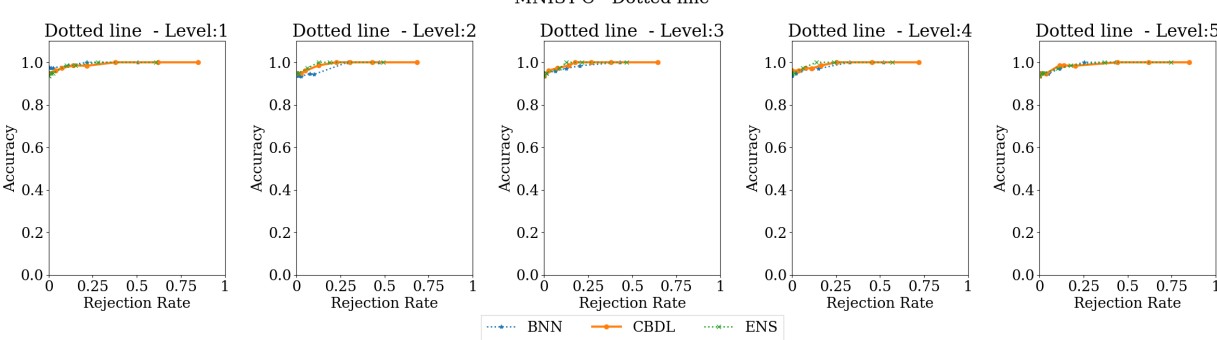

Figure 41: Accuracy vs Rejection Rate - MNISTC dotted line.

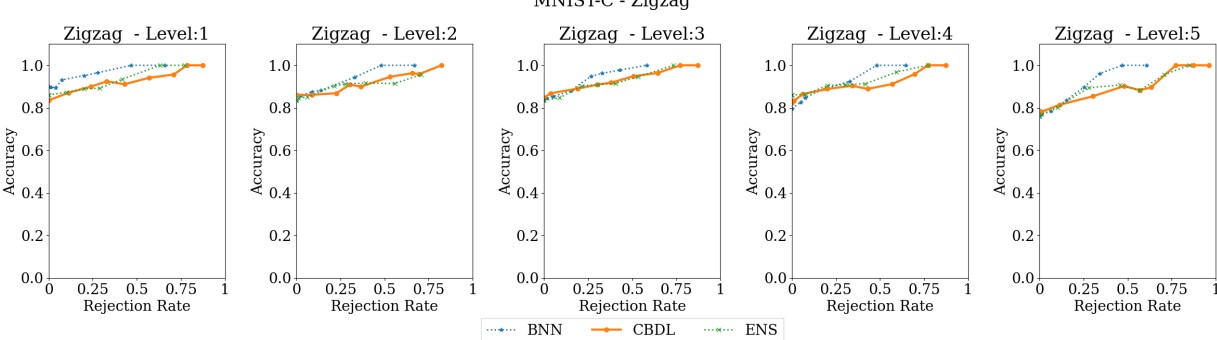

Figure 42: Accuracy vs Rejection Rate - MNISTC zigzag.

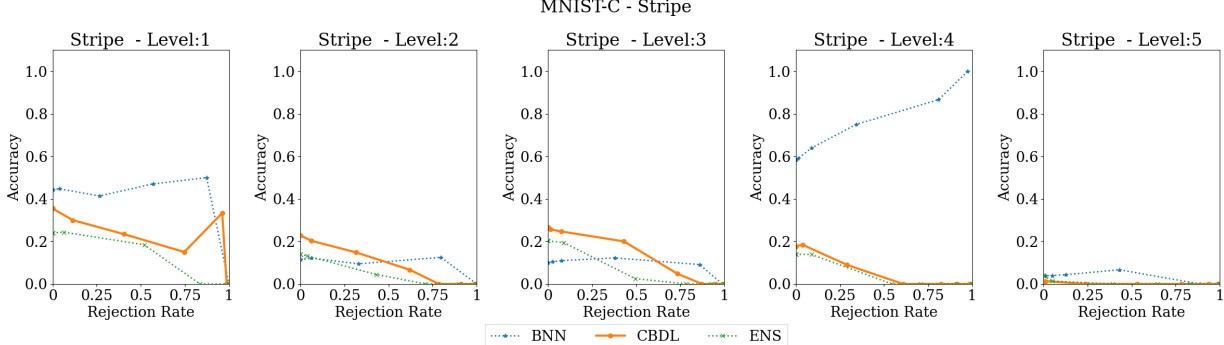

Figure 43: Accuracy vs Rejection Rate - MNISTC stripe.

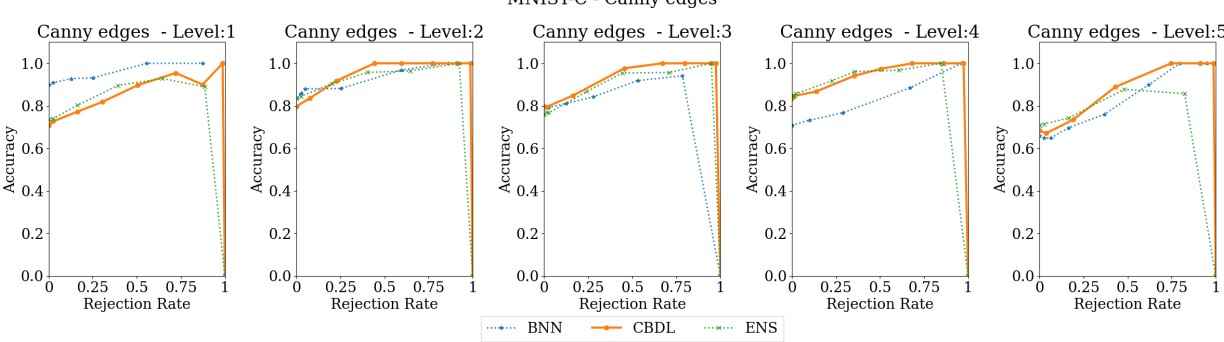

Figure 44: Accuracy vs Rejection Rate - MNISTC canny edges.

## P.3 Fashion MNIST-C

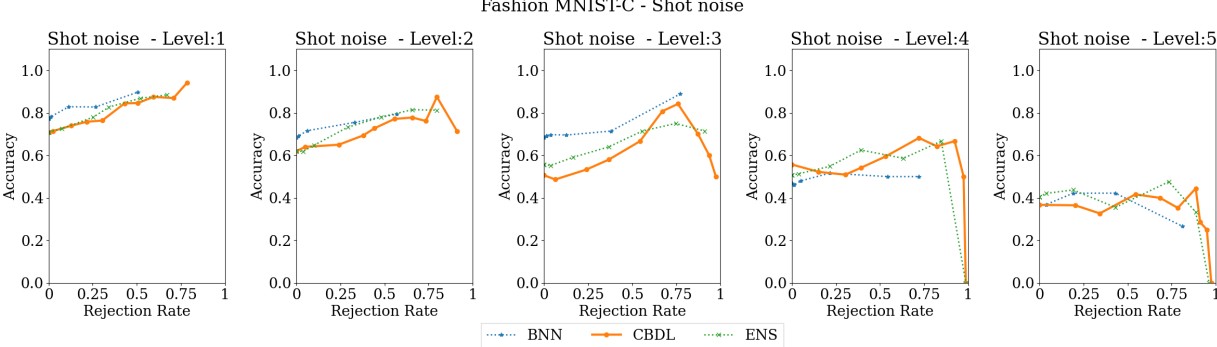

Figure 45: Accuracy vs Rejection Rate - FashionMNISTC shot noise.

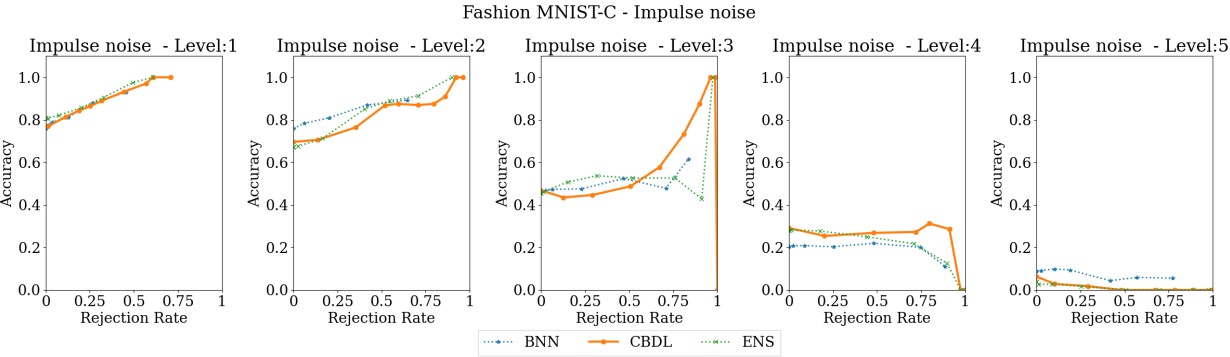

Figure 46: Accuracy vs Rejection Rate - FashionMNISTC impulse noise.

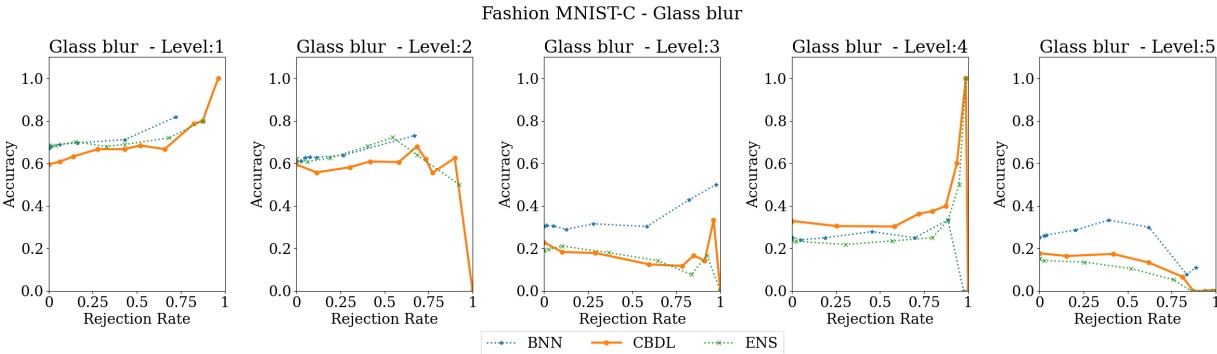

Figure 47: Accuracy vs Rejection Rate - FashionMNISTC glass blur.

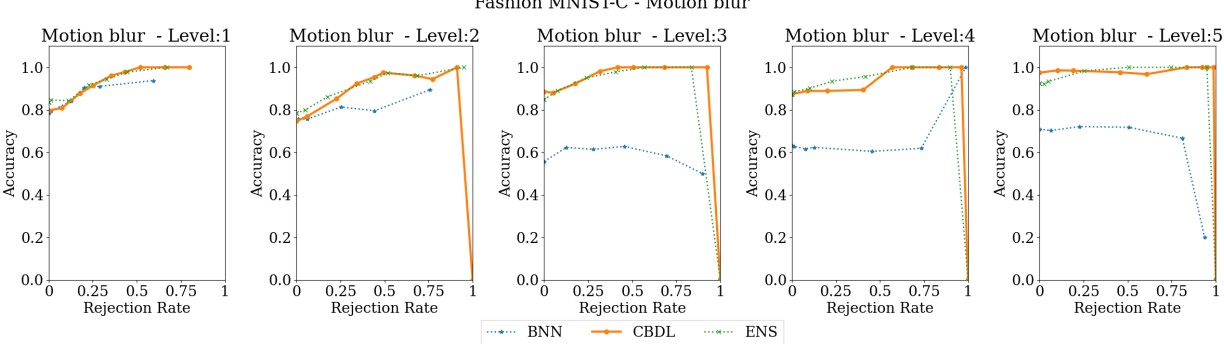

Figure 48: Accuracy vs Rejection Rate - FashionMNISTC motion blur.

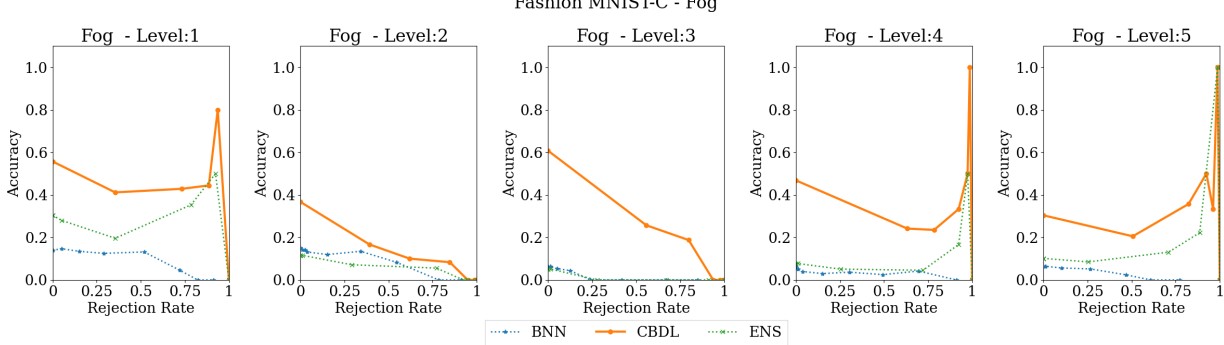

Figure 49: Accuracy vs Rejection Rate - FashionMNISTC fog.

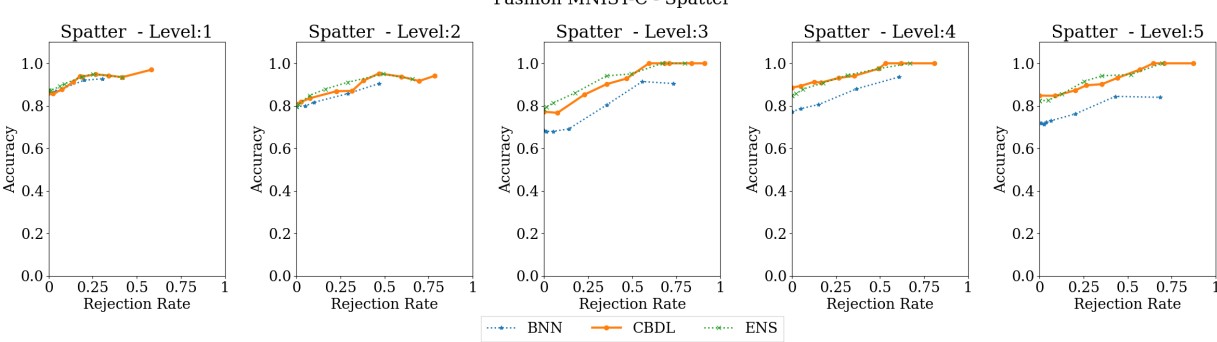

Figure 50: Accuracy vs Rejection Rate - FashionMNISTC spatter.

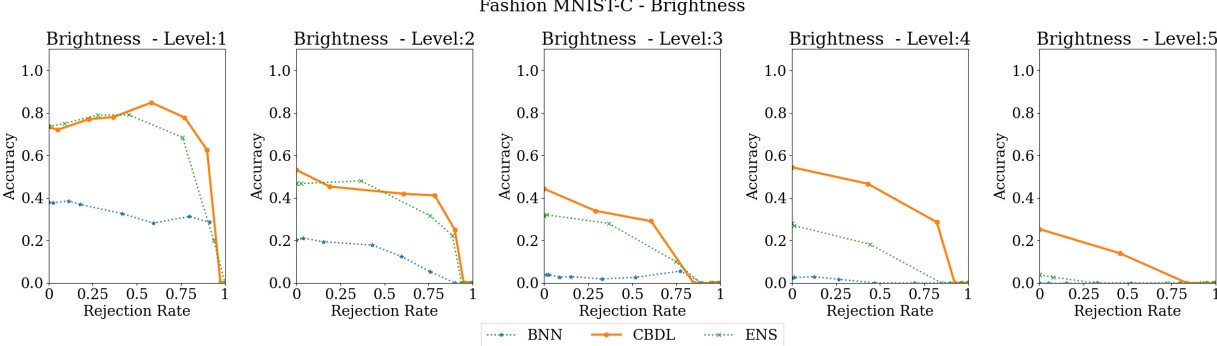

Figure 51: Accuracy vs Rejection Rate - FashionMNISTC brightness.

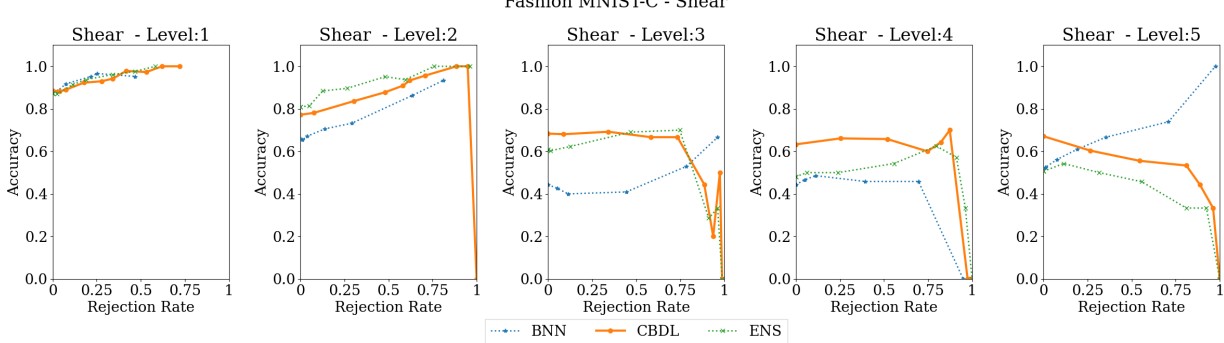

Figure 52: Accuracy vs Rejection Rate - FashionMNISTC shear.

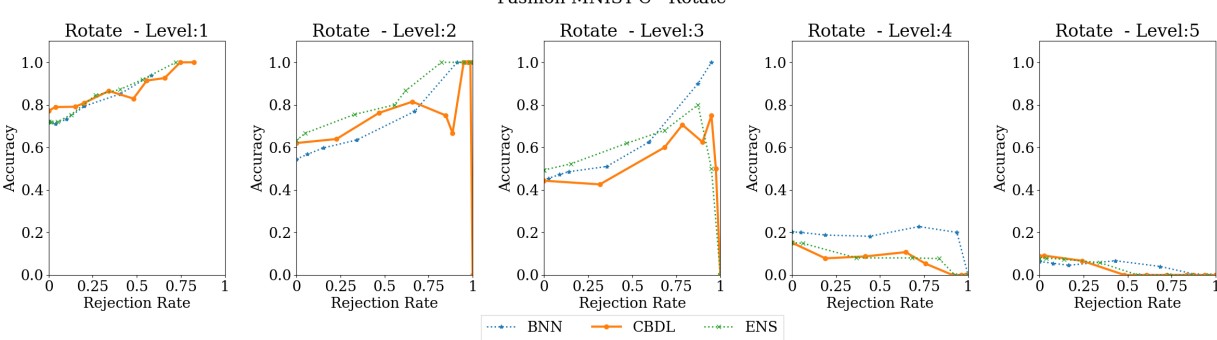

Figure 53: Accuracy vs Rejection Rate - FashionMNISTC rotate.

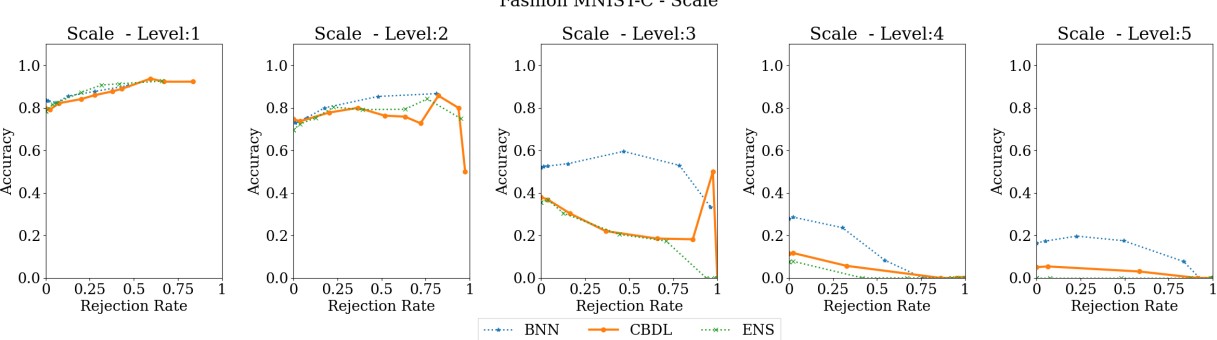

Figure 54: Accuracy vs Rejection Rate - FashionMNISTC scale.

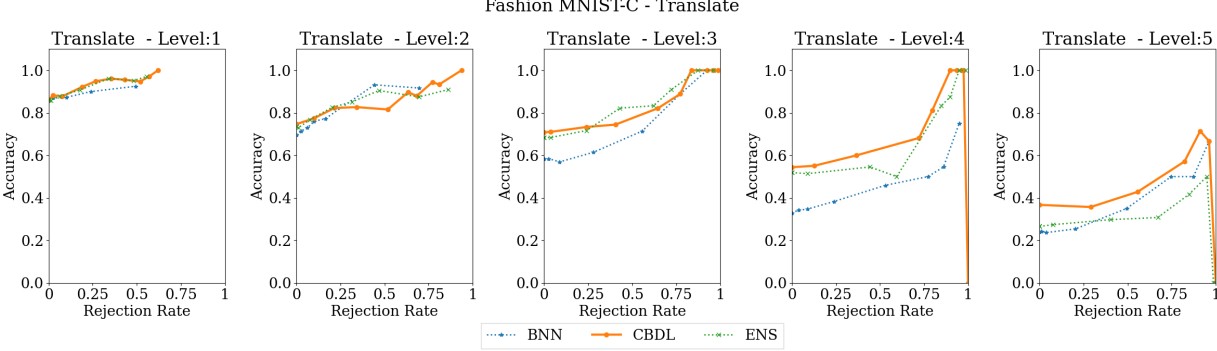

Figure 55: Accuracy vs Rejection Rate - FashionMNISTC translate.

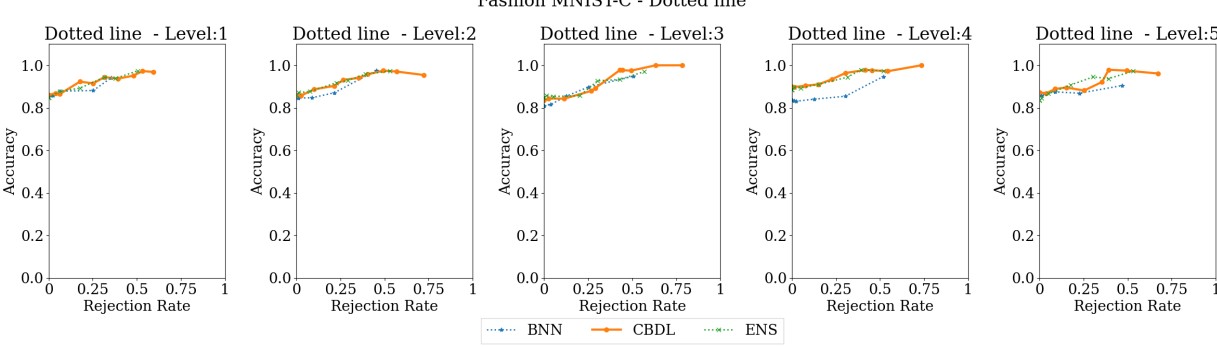

Figure 56: Accuracy vs Rejection Rate - FashionMNISTC dotted line.

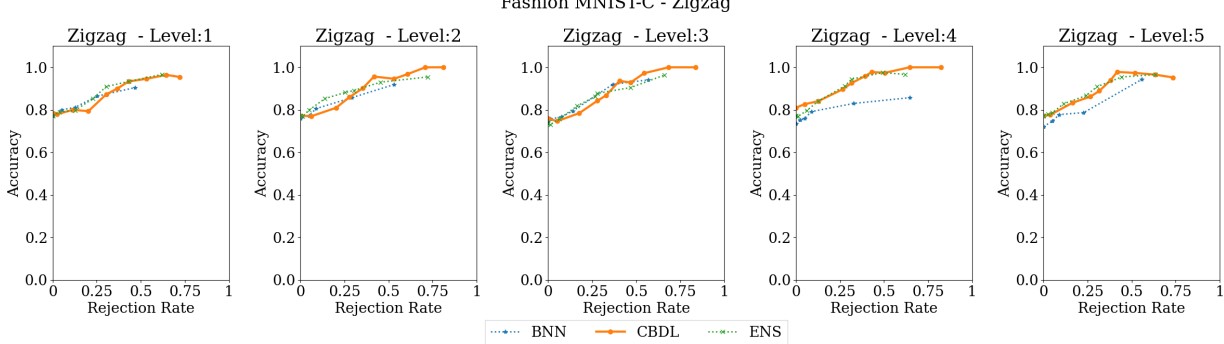

Figure 57: Accuracy vs Rejection Rate - FashionMNISTC zigzag.

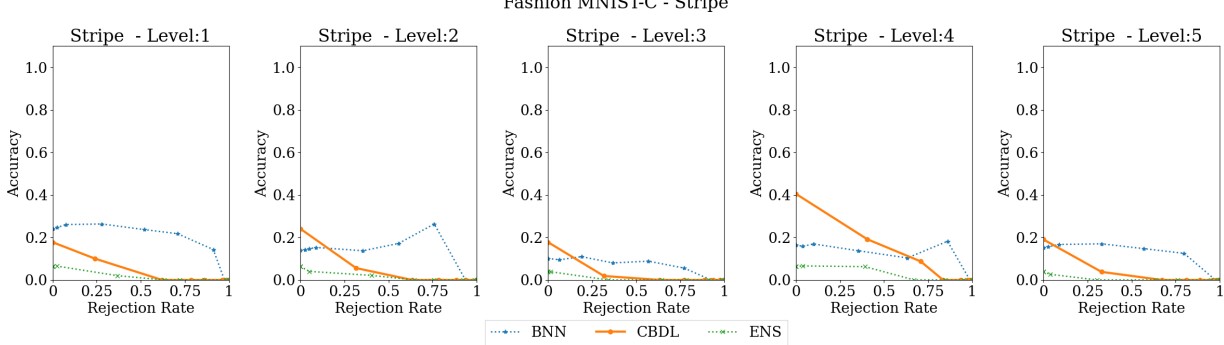

Figure 58: Accuracy vs Rejection Rate - FashionMNISTC stripe.

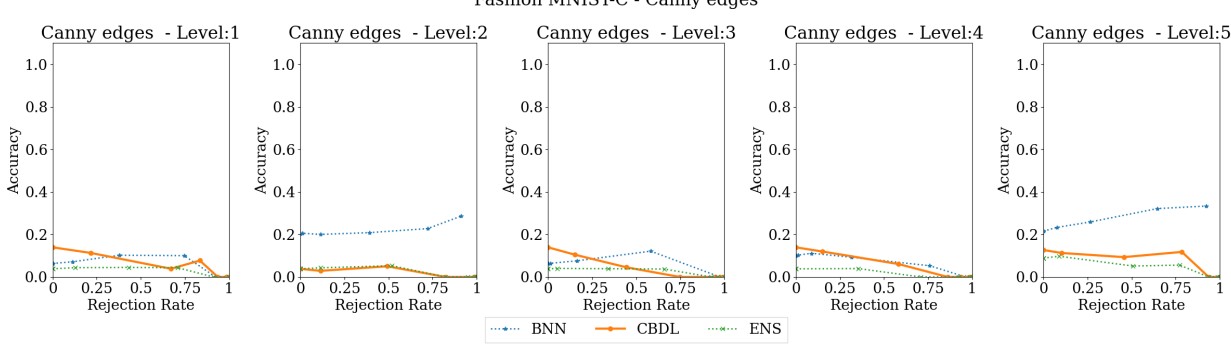

Figure 59: Accuracy vs Rejection Rate - FashionMNISTC canny edges.

## P.4 SVHN-C

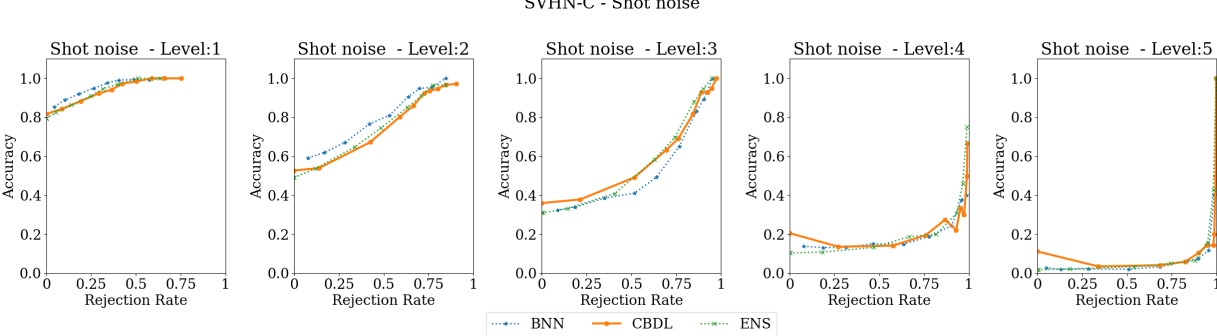

Figure 60: Accuracy vs Rejection Rate - SVHNC Gaussian Noise.

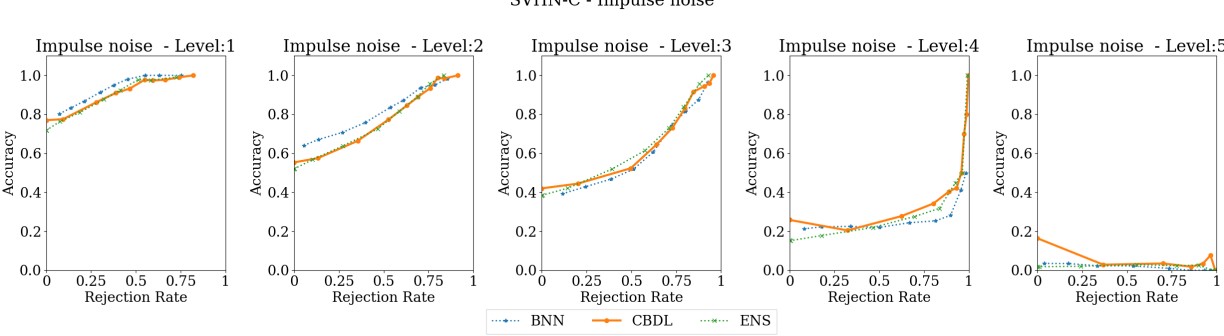

Figure 61: Accuracy vs Rejection Rate - SVHNC Shot Noise.

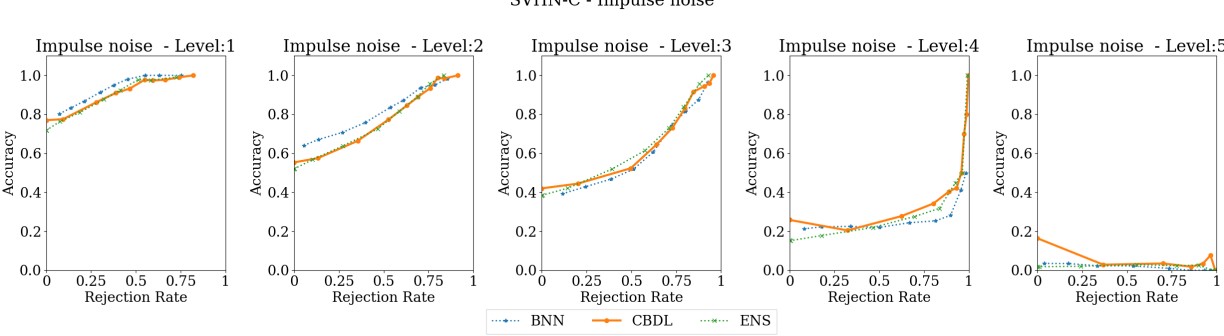

Figure 62: Accuracy vs Rejection Rate - SVHNC Impulse Noise.

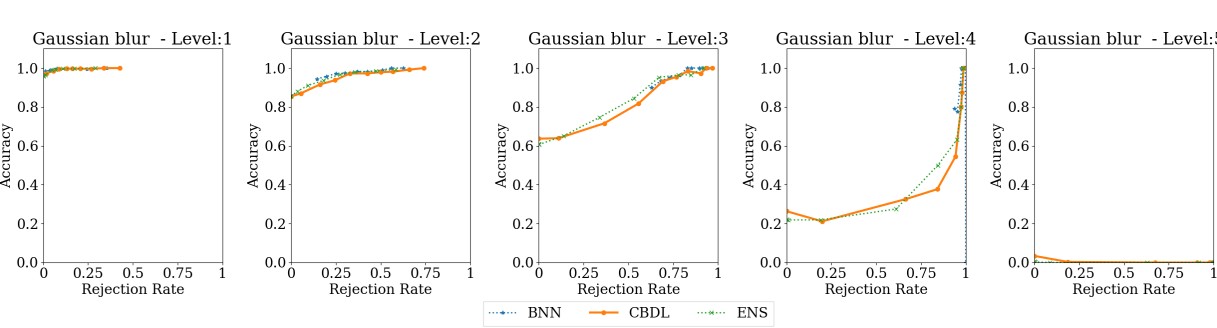

Figure 63: Accuracy vs Rejection Rate - SVHNC Speckle Noise.

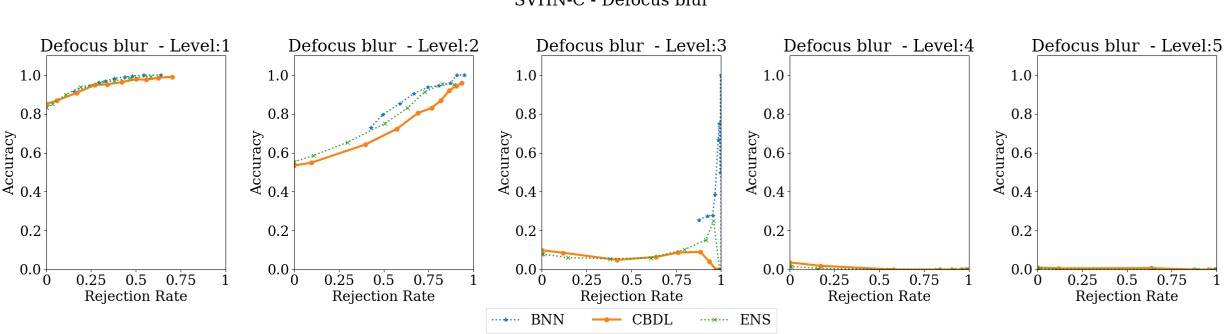

Figure 64: Accuracy vs Rejection Rate - SVHNC Gaussian Blur.

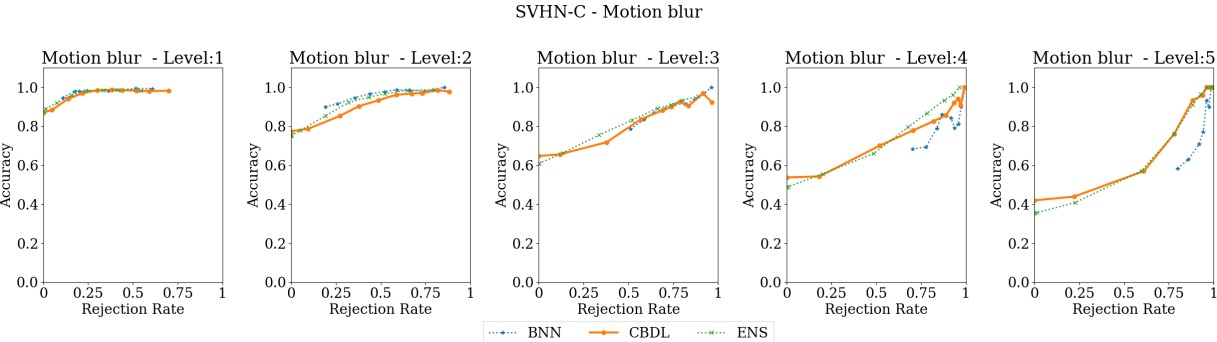

Figure 65: Accuracy vs Rejection Rate - SVHNC Defocus Blur.

Figure 66: Accuracy vs Rejection Rate - SVHNC Motion Blur.

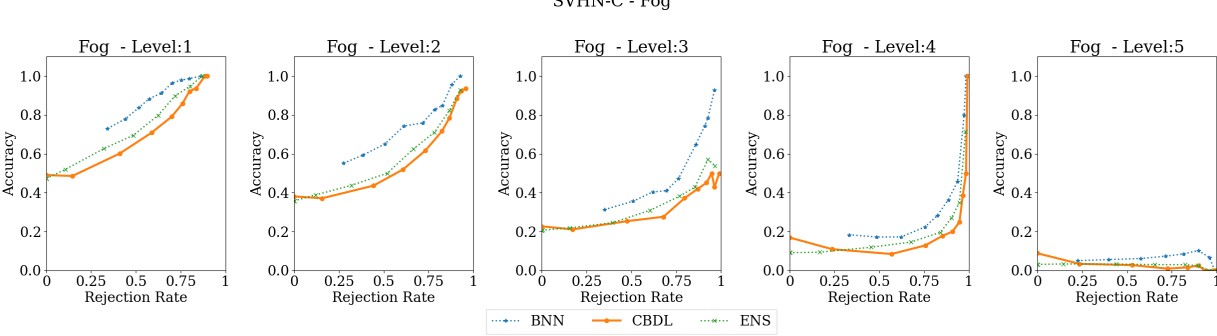

Figure 67: Accuracy vs Rejection Rate - SVHNC Zoom Blur.

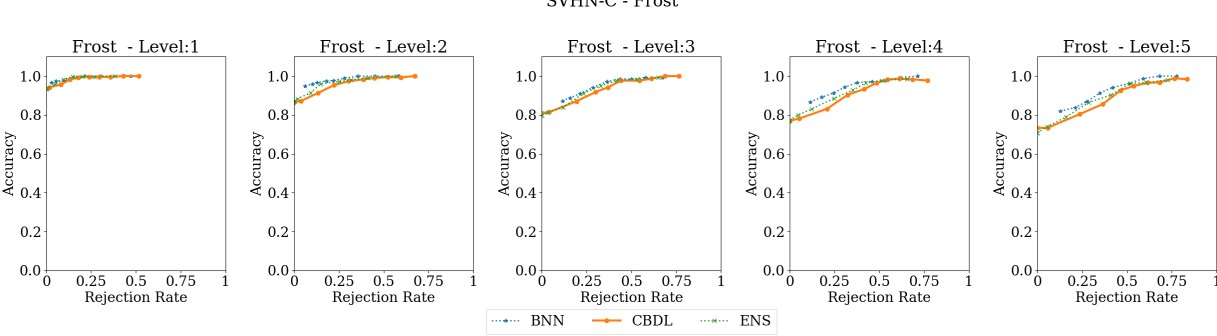

Figure 68: Accuracy vs Rejection Rate - SVHNC Fog.

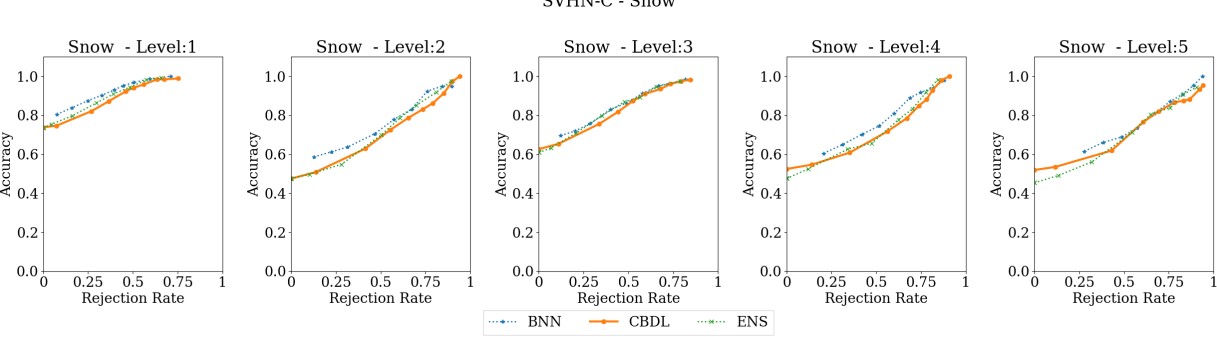

Figure 69: Accuracy vs Rejection Rate - SVHNC Frost.

Figure 70: Accuracy vs Rejection Rate - SVHNC Snow.

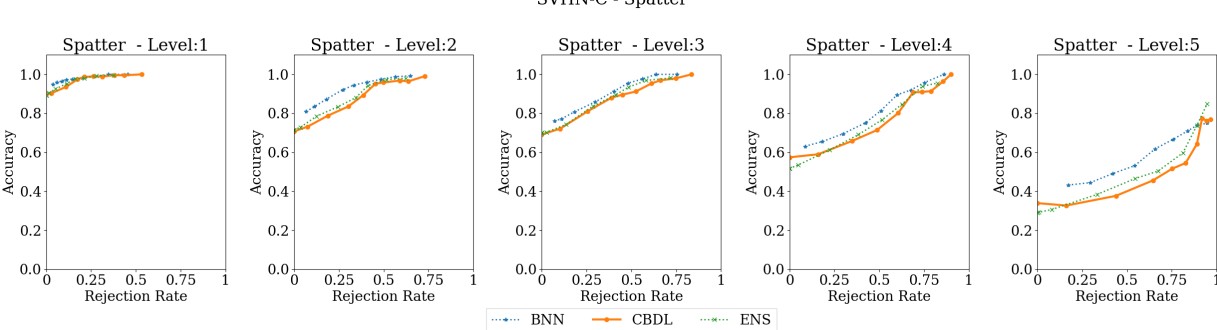

Figure 71: Accuracy vs Rejection Rate - SVHNC Spatter.

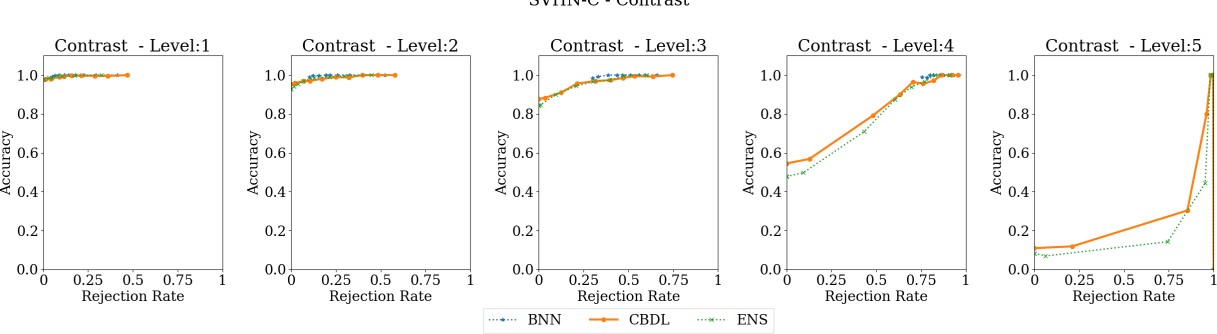

Figure 72: Accuracy vs Rejection Rate - SVHNC Contrast.

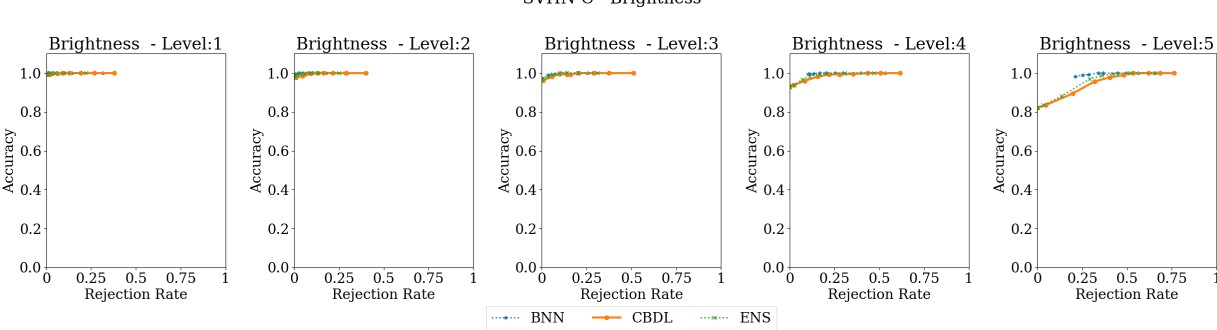

Figure 73: Accuracy vs Rejection Rate - SVHNC Brightness.

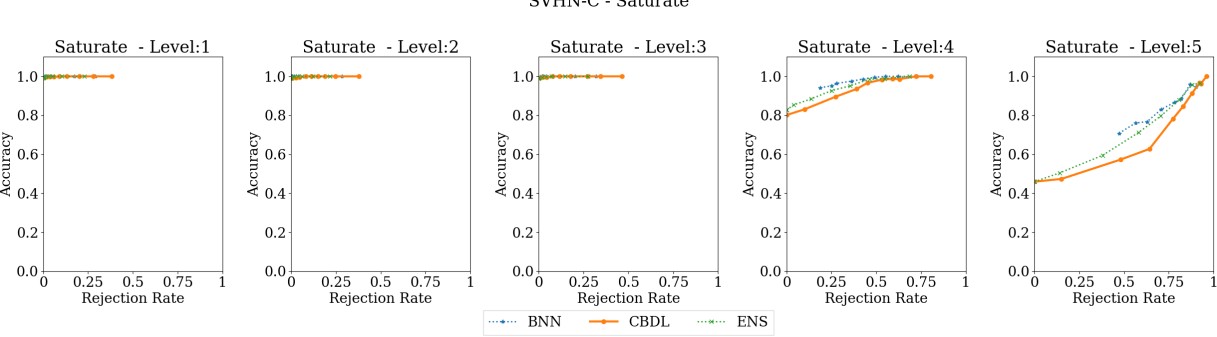

Figure 74: Accuracy vs Rejection Rate - SVHNC Saturate.

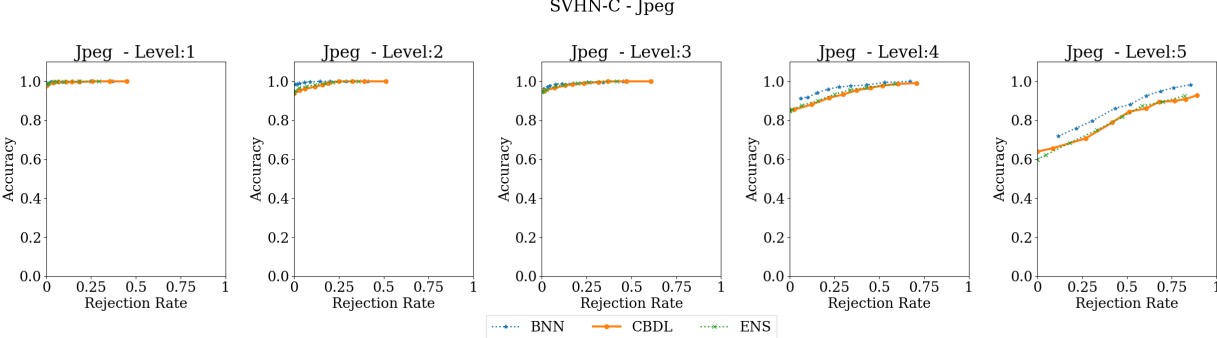

Figure 75: Accuracy vs Rejection Rate - SVHNC JPEG.

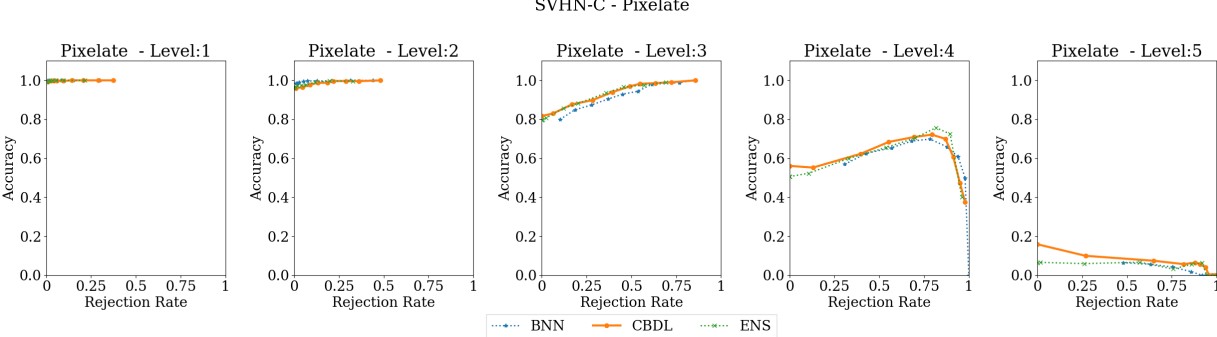

Figure 76: Accuracy vs Rejection Rate - SVHNC Pixelate.

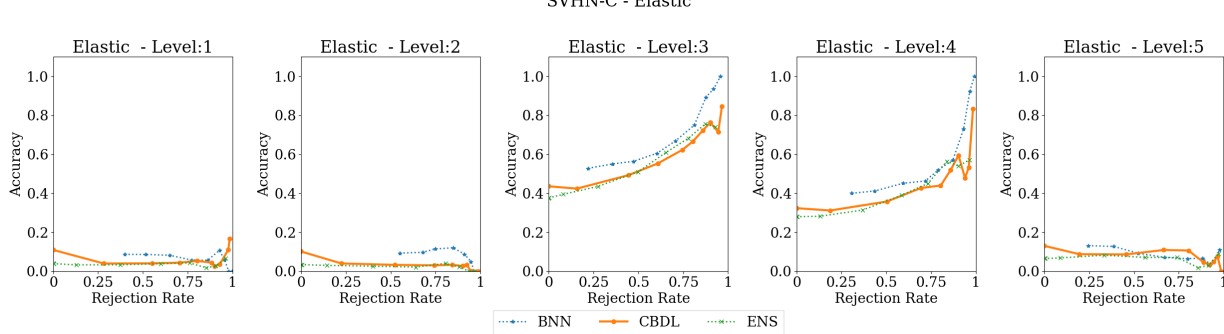

Figure 77: Accuracy vs Rejection Rate - SVHNC Elastic.

## Q   Code and Dataset Availability

Code and datasets for implementation and replication of results will be available at `https://github.com/PRECISE/credal-bayesian-deep-learning.git`.

