# OpenReview forum: "Credal Bayesian Deep Learning"
_TMLR — Accepted by TMLR_

### Review · Reviewer_CP1D · 2024-06-20

**Summary Of Contributions:**

The paper focuses on quantifying uncertainty in deep learning. The proposed approach is based on credal sets, i.e., on sets of probability distributions, Bayesian learning, and on disaggregation of bounds on total uncertainty (TU) into aleatoric uncertainty (AU) and epistemic uncertainty (EU) component bounds. The authors compare their approach to (ensembles of) Bayesian neural networks ((e)BNNs), providing theoretical and empirical evidence that their credal Bayesian deep learning (CBDL) approach leads to improved uncertainty estimates under various scenarios. All in all, I think the authors mostly follow up on the claims made by presenting a combination of theoretical arguments and empirical evidence.

**Audience:**

Yes

**Broader Impact Concerns:**

I have no broader impact concerns for this paper.

**Claims And Evidence:**

Yes

**Requested Changes:**

Questions and suggestions roughly in decreasing order of importance:

1) Sec1, p1: "In this paper, we present a procedure that allows us to give a machine such a desirable quality." Please tone down the claim: there are existing methods which do this (even if the proposed method would improve on all of them, which is not clear), and the proposed method also does not work perfectly (see e.g. results in Table 3).

2) Sec1, p2: "CBDL also gives a way of quantifying and disentangling different types of uncertainties within $P_{pred}$". As far as I can see, the approach to disentangling AU and EU using TU comes from Abellàn et al. 2006? If so, please cite them also in the Intro, now this reads like you are proposing a novel approach to the disaggregation problem.

3) Sec1, p3: Comparing CBDL with non-Bayesian techniques: I do not fully understand why comparing to existing methods beyond Bayesian ones is out of scope. Can you explain?

4) W.r.t. to the "bad choice" discussion on p11: How sensitive the resulting uncertainty estimates are on the choice of the set of priors and likelihoods for the propose method? How much better such robustness is compared, e.g., to finite eBNNs? It would be good to have some discussion and empirical experiments to demonstrate this.

5) Sec4.2.2, p20: "We observe that for lower values of α the gains of CBDL are more pronounced." Looking at Table 7, the performance improvements with lowest alpha level (0.9) are negative, which I interpret as saying that CBDL is worse than eBNNs. Do I misread this (also: is there spelling mistake in Table 7: IBNN should be CBDL?)?

6) E.g. Sec1, p2: P_pred "enjoys desirable probabilistic guarantees", also Sec1, p3. Do all the stated theoretical properties carry over when the posterior/predictive distribution is only approximated, e.g., via VI, instead of calculated exactly? Please state this clearly in the paper.

7) Sec1, p3: "CBDL [...] is able to quantify both EU and AU, [...] in a principled manner, unlike ensemble of BNNs." As far as I know, disentangling TU into AU and EU like done in this paper is also not without potential problems (see e.g. Sec.3.3 in Hüllermeier et al. 2022). Why is this more principled than if one does something similar with eBNNS? You also state (e.g. in Abstract) that, heuristically, CBDL "allows to train [...] infinite ensemble of BNNs". If so, again, why is this more principled than if one uses finite ensemble of BNNs instead?

8) Sec3.2, p12 "This does not mean that we suffer from under-confidence". I am not sure if I understand the claim, so just to clarify: e.g. enlarging the set of priors/likelihoods around a true oracle prior/likelihood does not come with any cost in terms of increased uncertainty? How would having a larger set of priors/likelihoods compare to having single correctly specified prior/likelihood?

9) Sec1, p2: "This because [sic] selecting a unique prior and a unique likelihood implicitly assumes perfect knowledge around the true prior and the true data generating process. In turn, a unique distribution is only able to retrieve AU". I would have guessed that single prior and likelihood would result in basically estimating TU (i.e., pretending that EU is zero does not actually make the uncertainty "go away"). Can you elaborate?

10) Abstract:  please clarify the claim: "Although Bayesian Neural Networks (BNNs) allow for uncertainty in the predictions to be assessed, different sources of uncertainty are indistinguishable". Do you mean specifically using a single BNN here?

11) Footnote 11, p9: you might as well state these for convenience at least in some appendix.

12) Sec4.1, p14: is there some motivation for the scale of severity, or is this just an empirical labeling?

13) Including a table with the notation and abbreviations might help in reading the paper: it is rather easy to get lost in soup of letters at some point.

**Strengths And Weaknesses:**

### Strengths

* The proposed approach has some theoretical justification and empirically leads to improved uncertainty estimates, especially for the AU component, compared to the eBNN baseline under various settings.

* The problem of uncertainty quantification is important and timely.

* The empirical problems seem interesting and relevant for the topic.


### Weaknesses

* The paper is generally a bit hard to read, with plenty of side comments and detours. Improving, clarifying and focusing the writing could make the paper a lot easier to read and understand.

* There is next to no discussion and no empirical comparison against other existing approaches beyond (e)BNNs.

* While the proposed approach does seem to work better than the baseline, many of the results are still not very satisfying (e.g., AU estimates are not consistently increasing when increasing the corruption level).

* Claiming that the proposed approach is theoretically well-justified while, e.g., eBNNs are not seems like a strong claim, since as far as I know, it is not generally clear how disentangling TU into AU and EU should actually be done in a principled manner (see e.g. Sec.3.3 in Hüllermeier et al. 2022 for a discussion).

* One of the main limitations of the proposed approach seems to be the high computational complexity, yet this aspect is not discussed much, especially in connection to the empirical results.

#### References:

Hüllermeier et al. 2022: Quantification of Credal Uncertainty in Machine Learning: A Critical Analysis and Empirical Comparison.

---

> ### Author Response · Authors · 2024-07-13
> **Thank you for your comments**
>
> We thank the reviewer for their insightful comments. Let us address them in depth.
>
> *The paper is generally a bit hard to read, with plenty of side comments and detours. Improving, clarifying and focusing the writing could make the paper a lot easier to read and understand.*
>
> We thank the reviewer for this suggestion. In the updated version, we will make our writing more terse.
>
> *While the proposed approach does seem to work better than the baseline, many of the results are still not very satisfying (e.g., AU estimates are not consistently increasing when increasing the corruption level).*
>
> Our AU measure increases when the corruption level is increased, while the one used by eBNNs does not. This proves empirically that aleatoric uncertainty measures based on credal sets improve on those based on ensembles of BNNs. The reviewer is right, though, in stating that for some types of corruption the maximum AU is reached not at the maximum level of corruption. Nevertheless, when this happens, the AU level at the peak of corruption is very close in value to the maximum AU level. This could have two possible explanations. It could be due to empirical reasons linked to the specific dataset and AU measure we used, or it may be that for some types of datasets uncertainty reaches a “critical plateau” after which credal-set based measures cannot register an uncertainty increase any more. While these are profound questions that deserve a separate treatment, we will add a remark addressing them in the new version of the manuscript. We also point out how the downstream tasks capability of CBDL are apparent in the f1/10 and pancreas experiments. We discuss more about the latter in the “Requested Changes'' part of our answers.
>
> *One of the main limitations of the proposed approach seems to be the high computational complexity, yet this aspect is not discussed much, especially in connection to the empirical results.*
>
> We thank the reviewer for the opportunity to clarify on this issue. As we point out in the paper, the computational cost we pay is needed for a better quantification of different types of uncertainty. We realize, though, that this is a shortcoming, and we are currently working on overcoming it. In particular, an evidential approach seems to be a promising route, as it allows us to derive a credal set without resorting to approximating multiple posteriors. If the reviewer is interested in the actual CPU or GPU time, we can add it as a footnote or in the appendices.
>
> *Sec1, p1: "In this paper, we present a procedure that allows us to give a machine such a desirable quality." Please tone down the claim: there are existing methods which do this (even if the proposed method would improve on all of them, which is not clear), and the proposed method also does not work perfectly (see e.g. results in Table 3).*
>
> We agree with the reviewer, and we will rephrase the sentence.
>
> *Sec1, p2: "CBDL also gives a way of quantifying and disentangling different types of uncertainties within P_{pred}". As far as I can see, the approach to disentangling AU and EU using TU comes from Abellàn et al. 2006? If so, please cite them also in the Intro, now this reads like you are proposing a novel approach to the disaggregation problem.*
>
> We agree with the reviewer, and we will make our reliance on Abellan’s measures clearer. We notice in passing how in the paper we mention that, once the predictive credal set is derived, the user can choose different uncertainty measures, as long as the total uncertainty measure they choose is upper bounded.

---

> > ### Author Response · Authors · 2024-07-13
> > **Thank you for your comments (cont'd)**
> >
> > *Sec1, p3: Comparing CBDL with non-Bayesian techniques: I do not fully understand why comparing to existing methods beyond Bayesian ones is out of scope. Can you explain?*
> >
> > We thank the reviewer for the opportunity to clarify this point. We only compare against Bayesian techniques because a comparison against non-Bayesian ones is either difficult to justify or unfair. Indeed, the initial knowledge component, captured by the prior distribution (or the set of priors, in our case), has no direct comparisons with other approaches. Take conformal prediction for instance. It is a model-free method, which means that no prior knowledge around the experiment at hand is required. The only choice the user makes is which non-conformity score to use. How would we compare this approach with the choice of different priors, which conveys the idea that the user knows that a true distribution over the space of neural network parameters exists, but it is not fully known? In addition, despite the ubiquitous claim that conformal prediction (CP) is an uncertainty quantification tool, it is actually not. CP is an uncertainty ***representation*** tool. Indeed, CP represents uncertainty via the conformal prediction region. It ***does not*** quantify it: there is no real value attached to any kind of uncertainty (aleatoric or epistemic). Some claim that the diameter of the conformal prediction region quantifies the uncertainty, but even in that case, it is unable to distinguish between aleatoric and epistemic. Indeed, the diameter is a positive function of both: it increases as both increase, and hence it cannot be used to distinguish the two. We will add this argument in a remark in the introduction to further clarify why we only focus on Bayesian baselines.
> >
> > *W.r.t. to the "bad choice" discussion on p11: How sensitive the resulting uncertainty estimates are on the choice of the set of priors and likelihoods for the proposed method? How much better such robustness is compared, e.g., to finite eBNNs? It would be good to have some discussion and empirical experiments to demonstrate this.*
> >
> > We thank the reviewer for the deep question. Credal sets are able to “self-regulate”. That is, in the case of prior-likelihood conflict (which happens e.g. if the prior set is ill-specified), the posterior credal set, and in turn the predictive credal set too, will be wider. This will be reflected in the uncertainty measures which will register an excess of epistemic uncertainty. In that case, we could retrain the model using an augmented dataset (e.g. via semantic preserving transformation), to obtain a narrower posterior set, since (if the prior is a proper subset of the whole space of distributions over the parameters of the neural network) “data swamps the prior”. That is, as the number of training data increases, CBDL will “care more” about the likelihood credal set in deriving the posterior credal set. Finite eBNNs do not enjoy this robustness because they average over the induced predictive distributions, and hence the prior-likelihood conflict information is washed out by the average. We will add a discussion about this matter, but we will not add any more experiments. The reason is that for them to be meaningful, we would need to know the true prior and the true likelihood, something that never holds in practice.
> >
> > *Sec4.2.2, p20: "We observe that for lower values of α the gains of CBDL are more pronounced." Looking at Table 7, the performance improvements with lowest alpha level (0.9) are negative, which I interpret as saying that CBDL is worse than eBNNs. Do I misread this (also: is there a spelling mistake in Table 7: IBNN should be CBDL?)?*
> >
> > We thank the reviewer for catching the typo: it is indeed CBDL and not IBNN. The reviewer is right: CBDL’s downstream performance is worse than eBNNs’ ones when 1 - alpha = 0.9. This is due to the fact that alpha = 0.1 is a relatively large value, and very seldomly used in applications where high levels of safety need to be ensured. When alpha is 0.05 or 0.01, CBDL outperforms eBNNs.
> >
> > *E.g. Sec1, p2: P_pred "enjoys desirable probabilistic guarantees", also Sec1, p3. Do all the stated theoretical properties carry over when the posterior/predictive distribution is only approximated, e.g., via VI, instead of calculated exactly? Please state this clearly in the paper.*
> >
> > We thank the reviewer for pointing this out: yes, they all carry over because they only hinge upon the fact that P_{pred} is a credal set. We will make this clearer in the new version.

---

> > > ### Author Response · Authors · 2024-07-13
> > > **Thank you for your comments (cont'd)**
> > >
> > > *Sec1, p3: "CBDL [...] is able to quantify both EU and AU, [...] in a principled manner, unlike ensemble of BNNs." As far as I know, disentangling TU into AU and EU like done in this paper is also not without potential problems (see e.g. Sec.3.3 in Hüllermeier et al. 2022). Why is this more principled than if one does something similar with eBNNS? You also state (e.g. in Abstract) that, heuristically, CBDL "allows to train [...] infinite ensemble of BNNs". If so, again, why is this more principled than if one uses a finite ensemble of BNNs instead?*
> > >
> > > We thank the reviewer for giving us the opportunity to improve this sentence. In the new version, we will be more explicit in acknowledging that upper and lower entropy suffer from some shortcoming. In addition, as we pointed out before, the reason we derive a credal set is that we can use any of the available uncertainty measures for credal sets to quantify and disentangle epistemic and aleatoric uncertainties. For example, in classification problems, we can use the generalized Hartley measure – which overcomes the shortcomings of upper and lower entropy – to quantify epistemic uncertainty. We only need that the measure we choose for the total uncertainty is upper bounded.
> > > Our statement concerning the eBNNs, although maybe a little too harsh, stems from the fact that the measures for epistemic and aleatoric uncertainties used in ensemble models are chosen for their convenience, rather than a profound theoretical reason. We will make this clearer in the updated version.
> > > Finally, infinite ensembles allow to take into account all possible combinations of the plausible priors and likelihoods elicited by the researcher. eBNNs, instead, only consider a “uniform” combination, where all weights are equal. One can see eBNNs as selecting one particular element from  a credal set. For this reason, because CBDL allows to keep distributions separate and explicitly account for their difference, we state that CBDL is a more principled approach.
> > >
> > > *Sec3.2, p12 "This does not mean that we suffer from under-confidence". I am not sure if I understand the claim, so just to clarify: e.g. enlarging the set of priors/likelihoods around a true oracle prior/likelihood does not come with any cost in terms of increased uncertainty? How would having a larger set of priors/likelihoods compare to having single correctly specified prior/likelihood?*
> > >
> > > We apologize for the confusion. Here, we mean the following. If the oracle prior and the oracle likelihood belong to the respective credal sets, the oracle posterior will belong to the posterior credal set as well. The posterior credal set, though, will typically not be a singleton given by the oracle posterior. This is because no finite amount of training data is able to reduce the epistemic uncertainty embedded in the credal sets to zero, and hence to recover a singleton posterior credal set. This is a well-known property of credal sets studied first by Walley in 1991. This does not mean that we are under-confident, though: indeed, the relative epistemic uncertainty (given by the difference between prior and posterior epistemic uncertainty, divided by the prior epistemic uncertainty) drops significantly. We will make this point clearer in the new version.
> > >
> > > *Sec1, p2: "This because [sic] selecting a unique prior and a unique likelihood implicitly assumes perfect knowledge around the true prior and the true data generating process. In turn, a unique distribution is only able to retrieve AU". I would have guessed that single prior and likelihood would result in basically estimating TU (i.e., pretending that EU is zero does not actually make the uncertainty "go away"). Can you elaborate?*
> > >
> > > The reviewer is correct: when a single prior and likelihood are selected, we only estimate TU which coincides with AU, since the EU is zero in that case. We will make this clearer in the new version.
> > >
> > > *Abstract: please clarify the claim: "Although Bayesian Neural Networks (BNNs) allow for uncertainty in the predictions to be assessed, different sources of uncertainty are indistinguishable". Do you mean specifically using a single BNN here?*
> > >
> > > Indeed we do: a single BNN is unable to quantify EU because the agent implicitly assumes perfect knowledge of the prior and of the likelihood.
> > >
> > > *Footnote 11, p9: you might as well state these for convenience at least in some appendix.*
> > >
> > > We thank the reviewer for the suggestion that we will implement in the new version.
> > >
> > > *Sec4.1, p14: is there some motivation for the scale of severity, or is this just an empirical labeling?*
> > >
> > > It is an empirical labeling to better visualize and convey our results.
> > >
> > > *Including a table with the notation and abbreviations might help in reading the paper: it is rather easy to get lost in a soup of letters at some point.*
> > >
> > > We thank the reviewer for the suggestion that we will implement in the new version.

---

### Review · Reviewer_tTAz · 2024-07-07

**Summary Of Contributions:**

Adapted from my previous review (Paper1573 by Reviewer cXJk):

The paper introduces Credal Bayesian Deep Learning (CBDL), which allows for considering multiple priors and likelihoods and offer a different way of quantifying aleatoric and epistemic uncertainty. In empirical studies, they compare CBDL to ensembles of BNNs (EBNNs) for both regression and classification (in the appendix).

The paper suggests using multiple likelihoods and priors and simply performing Bayesian inference on all combinations to obtain a set of BNNs.

To quantify and disentangle aleatoric and epistemic uncertainty for an input the paper suggests determining the highest $\overline{H}(P)$ and lowest entropy $\underline{H}(P)$ and then using the decomposition:

$$
\underbrace{\bar{H}(P)}\_{\text {total uncertainty }}=\underbrace{\underline{H}(P)}\_{\text {aleatoric uncertainty }}+\underbrace{[\bar{H}(P)-\underline{H}(P)]}\_{\text {epistemic uncertainty}}.
$$

To make predictions, the paper defines regions of imprecise high density across the BNNs.

---

Comparison to the previous submission: https://draftable.com/compare/sPCFPgnAKMzD

**Audience:**

Yes

**Claims And Evidence:**

No

**Requested Changes:**

From the last review:

1. to add comparisons to single BNNs as a baseline using the mean, stddev output parameterization as the ensembling for EBNNs might not be necessary,
2. compare to BMA as the cited paper is specific to HMC-based inference, which you do not seem to use anyway.

In particular, EBNNs are flawed compared to a single (merged) BNN, as detailed above, so please compare them to that. Writing

> From the previous sections, it is clear that CBDL improves on the uncertainty quantification capabilities of single BNNs

as the start of the "Experiments" section is not sufficient as evidence and, hence, one of the listed contributions is not evidenced.

### Typos etc

p. 2: "This because"
p. 4: the last paragraph cites Jospin et al (2022) 5 times
p. 5: the "(Berger, 1984), (Walley, 1991, Section 5.9)" citation should be merged.

Finally:

Compress the PDF before uploading please. I was able to compress the 4.3MB PDF down to 900KB.

**Strengths And Weaknesses:**

The paper addresses an important point that often comes up with Bayesian neural networks and Bayesian approaches in general: the question of selecting a prior and likelihood to perform Bayesian inference on. Indeed, an often-heard claim against BNNs is that the choice of prior and likelihood can appear arbitrary, and while Bayesian inference is principled and rational, the Bayesian viewpoint does not help much with the former. It provides a good overview and many references for underlying research on uncertainty quantification and Bayesian theory.

Thus, the research question is of great importance and the paper will find interest within the community.

At the same time, there are a few weaknesses:

1. The paper mentions Kendall & Gal and other works that explicitly disentangle epistemic and aleatoric uncertainty with a single BNN but then claims that BNNs cannot disentangle aleatoric and epistemic uncertainty. Thus, related work that is mentioned in this context is not compared or acknowledged properly.
2. On the face of it, the approach trains an ensemble of BNNs with different architectures and/or priors (see also Wenzel et al., 2020). As such, it would seem necessary to compare to approaches that use meta-priors as this could be viewed as Bayesian inference over meta-priors where we choose a uniform (uninformative) prior over the prior distributions and likelihoods.
3. The experiments only compare an ensemble of BNNs with the proposed method. While the paper mentions that they do not apply BMA (Bayesian model averaging) due to a paper that found that it does not well for BNNs approximated with certain settings for HMC, a comparison to regular BNNs (not ensembled) and different approximation methods would have been helpful to compare the quality of the proposed method.

Thus, I don’t see all the statements as sufficiently evidenced yet.

## Details

> We note in passing that EU cannot be captured using a single BNN (Hüllermeier & Waegeman, 2021).

This is simply wrong. Epistemic Uncertainty as defined in the same paragraph:

> EU, instead, refers to the lack of knowledge about the data-generating process; as such, it is reducible

can be measured using the concept of Expected Information Gain (EIG), introduced by Lindley (1956) and revisited in much research since then. This concept is exactly equivalent to the mutual information used to quantify EU in BNNs (see Kendal & Gal, etc.). The paper references these but does not actually compare them. Further, Hüllermeier & Waegeman (2021) is likely not the right citation.

> since AU is irreducible, there is an increasing need for ML techniques that are able to detect an excess of AU and query for human help.

This seems wrong as well. AU is irreducible, so assuming it is captured correctly by the model, querying a human for help will not help as AU is irreducible. Querying humans for help only helps for high EU. See the success of EU for active learning in BNNs and Bayesian Optimal Experiment Design.

### Ensemble of BNNs

From my previous review:

> **The problem with the comparison between EBNNs and CBDL is that for EBNNs, there is not necessarily a clean disentanglement of uncertainties, as each BNN might also capture some of the epistemic uncertainty within itself. That is, a sufficiently powerful model class could learn all possible uncertainty, so the resulting ensemble would express no epistemic uncertainty.**

Further, I am not aware of *anyone* using EBNNs. The citation (Egele et al., 2021) does not substantiate this: it does not propose EBNNs from my reading.

# IHDR as Union of HDRs

The paper states that the construction of IHDRs (Def 4) can be operationalized via a union of HDRs (Def 5). I thank the authors for clarifying my earlier misunderstanding about this. However, I fail to see how the second property is always fulfilled: does the union of HDRs really always provide a minimal cover of high density?

---

> ### Author Response · Authors · 2024-07-13
> **Thank you for your comments**
>
> We thank the reviewer for their insightful comments. Let us address them in depth.
>
> *Epistemic Uncertainty can be measured using the concept of Expected Information Gain (EIG), introduced by Lindley (1956) and revisited in much research since then.*
>
> We thank the reviewer for their comment. As far as we know, the EIG is defined as the expectation with respect to the likelihood distribution of the KL divergence between the posterior and the prior. This only quantifies how much information we gain around the parameter of interest (in this case, the parameters of the neural network) after training. While extremely interesting, this does not inform us on the uncertainty of the agent around the data generating process. Let us be clearer. Suppose we are in a one-dimensional setting, and that we are in a conjugate case where the likelihood is a Normal with known variance, and the prior on the mean of such Normal is itself a Normal with known parameters. Then, the posterior is again a Normal, and (if the prior is “well-specified”) will likely be more concentrated around the true parameter. That is, it will have “skinnier tails” and a higher peak around the true mean. EIG will quantify this “transition” from the prior to the posterior Normal, but it does not inform us of the uncertainty that the agent has around whether a Normal prior was a correct choice in the first place. What if the prior was a Student-t or a Cauchy instead? Then, the value of EIG would change. In this work, and in much current literature on uncertainty quantification – see e.g. recent works by Hüllermeier, Destercke, Denoeux, Cuzzolin, Benavoli, and others – EU captures model specification uncertainty. Think of another example. Suppose an agent wants to convey the idea of maximal uncertainty. Typically, this is done via the Uniform distribution. But the latter is unable to distinguish between the exact knowledge that all elements of the state space are equally likely, or the concept of maximal vagueness that brings the agent to assign the same probability to all possible elements. This is known as the Laplace paradox, and was first studied by Kyburg and Levi in the 1960s, and then further expanded by Walley in 1991. In the introductory part to his famous book, Walley states that “imprecise probabilities (which credal sets are a part of) are a powerful way of conducting analyses in the presence of both indeterminacy (what we call epistemic uncertainty) and uncertainty (what we call aleatoric uncertainty)”. We need imprecise probabilities to deal with indeterminacy/EU. We also thank the reviewer for pointing out that Hüllermeier & Waegeman (2021) is likely not the right citation: we will add more compelling ones in the updated version, such as https://openreview.net/pdf?id=MhLnSoWp3p, together with a remark making clearer why a single distribution is not well-suited to capture EU.
> Finally, recently https://arxiv.org/abs/2407.01985, it has been shown empirically that single Bayesian neural networks have severe difficulties in quantifying epistemic uncertainty. (One of) the reason(s) is the one we put forth in our paper: there is not a theoretically sound reason why they should be able to do so. We will add this argument, and the relevant reference, to the updated version of our manuscript.
>
> *AU is irreducible, so assuming it is captured correctly by the model, querying a human for help will not help as AU is irreducible. Querying humans for help only helps for high EU. See the success of EU for active learning in BNNs and Bayesian Optimal Experiment Design.*
>
> We thank the reviewer for pointing this out. It is certainly true that querying humans can help with EU as well, and we will include this in the new version of the paper. Instead of querying a human, in the new version we will point out how, in the case of excess of AU the model should flag this, so that the user knows to “proceed with caution”.

---

> > ### Comment · Reviewer_tTAz · 2024-07-21
> > **Re Epistemic Uncertainty**
> >
> > I disagree with many parts in this reply:
> >
> > > "EU, instead, refers to the lack of knowledge about the data generating process; as such, it is reducible. It can be lessened on the basis of additional data."
> >
> > This matches the usual definition of epistemic uncertainty. The expected information gain is a good proxy for this given plenty of prior literature.
> >
> > > But it does not inform us of the uncertainty that the agent has around whether a Normal prior was a correct choice in the first place. What if the prior was a Student-t or a Cauchy instead? Then, the value of EIG would change.
> >
> > The simple way to deal with this is to treat the model distribution as a discrete random variable, add a prior for that, and include this new parameter in the expected information gain.
> >
> > > Suppose an agent wants to convey the idea of maximal uncertainty. Typically, this is done via the Uniform distribution. But the latter is unable to distinguish between the exact knowledge that all elements of the state space are equally likely, or the concept of maximal vagueness that brings the agent to assign the same probability to all possible elements.
> >
> > Bayesian NNs and deep ensembles can perfectly capture this concept/example. For example, for 10 classes, let's have at least 10 ensemble members. Each ensemble member can output a different class with one-hot probability. The BMA is uniform then, and the epistemic uncertainty, captured via the mutual information, is maximal.
> >
> > See also the perspective that credal sets are just hyperpriors with uniform distribution --- from the last review:
> >
> > ---
> >
> > ### Meta-priors
> >
> > We can introduce a random variable $K$ that denotes the prior $\times$ likelihood combination. Then we can set up the following probabilistic model:
> > $$
> > p(y|x, \theta, k) = p(y|x, \theta, k) \, p(\theta|k) \, p(k),
> > $$
> > if we choose a uniform prior we recover the approach introduced by the paper:
> > $$
> > \begin{aligned}
> > p(y|x,\mathcal{D}) &= \mathbb{E}\_{p(k)} \, \mathbb{E}\_{p(\theta | \mathcal{D}, k)} p(y|x, \theta, k) \\
> > &\approx \mathbb{E}\_{p(k)} \, \mathbb{E}\_{q_k(\theta)} p(y|x, \theta, k),
> > \end{aligned}
> > $$
> > where we approximate $p(\theta | \mathcal{D}, k)$ using $q\_k(\theta)$ via variational approximation:
> > $$
> > q\_k(\theta) := \arg\min\_q \,\, D\_{KL}(q(\theta) || p(\theta | \mathcal{D}, k)).
> > $$
> > In this context, it would be interest to compare to Empirical Bayes and use a more informative prior for $p(k)$.
> >
> > ---
> >
> > I don't see why this paper's approach should not be viewed as introducing an additional hyperparameter with a uniform prior and compared to BNNs on that basis.

---

> > > ### Author Response · Authors · 2024-07-23
> > > **Thank you for your comments**
> > >
> > > *The simple way to deal with this is to treat the model distribution as a discrete random variable, add a prior for that, and include this new parameter in the expected information gain. [...] Bayesian NNs and deep ensembles can perfectly capture this concept/example. For example, for 10 classes, let's have at least 10 ensemble members. Each ensemble member can output a different class with one-hot probability. The BMA is uniform then, and the epistemic uncertainty, captured via the mutual information, is maximal.*
> > >
> > > We thank the reviewer for giving us the opportunity to further expand on this point. We first point out that there seems to be a confusion between first and second order probabilities. Laplace’s paradox refers to the former, while the reviewer argues in favor of the latter. Let us be clearer about this. Second order distributions are distributions over distributions. This is exactly what the reviewer refers to here: in particular, call $\Delta_\mathcal{Y}$ the space of probabilities on the space of classes $\mathcal{Y} = \lbrace{ 1,\ldots, 10 \rbrace}$. Then, the distribution the reviewer refers to is a discrete uniform over $\lbrace{ \delta_{\lbrace{ 1 \rbrace}}, \ldots, \delta_{\lbrace{ 10 \rbrace}} \rbrace} \subset \Delta_\mathcal{Y}$, where $\delta_{\lbrace{ 1 \rbrace}}$ refers to the Dirac distribution (or one-hot probability) over class $\lbrace{ 1 \rbrace}$, and so on. This second order distribution, that one can call second order discrete uniform, can then be used to derive a predictive distribution. While in this framework EIG might be a somehow good measure for epistemic uncertainty, as we pointed out in our previous answers, second-order approaches suffer from shortcomings. We will not compare our model, whose main feature is being able to quantify and disentangle different types of uncertainties in a theoretically sound manner, to a way of proceeding (second order approaches) whose negative sides have been extensively studied, both theoretically and empirically. We will, though, add a paragraph in which we explain in detail what we presented so far in this answer session.
> > >
> > > *See also the perspective that credal sets are just hyperpriors with uniform distribution. [...] I don't see why this paper's approach should not be viewed as introducing an additional hyperparameter with a uniform prior.*
> > >
> > > We thank the reviewer for giving us the opportunity to clarify this once again. Credal sets are not hyperpriors with uniform distribution. The hyperprior process suggested by the reviewer can be used to select one distribution from the predictive credal set. In particular, it selects the center of gravity of the predictive credal set (https://bellman.ciencias.uniovi.es/~emiranda/centroids-anor.pdf, Section 3.2)  While very interesting in its own right, this is not what our method does. In fact, it would actually defy the purpose of working with credal sets. During training, our method produces a set of posterior distributions, which, during inference, is used to derive a set of predictive distributions. They are kept separate; we do not average over them. The output at inference time, then, is either the predictive IHDR, or the prediction having the highest lower probability. The theoretical downsides of a hyperprior with uniform distribution were addressed in the previous answers.

---

> ### Author Response · Authors · 2024-07-13
> **Thank you (cont'd)**
>
> *The problem with the comparison between EBNNs and CBDL is that for EBNNs, there is not necessarily a clean disentanglement of uncertainties, as each BNN might also capture some of the epistemic uncertainty within itself. That is, a sufficiently powerful model class could learn all possible uncertainty, so the resulting ensemble would express no epistemic uncertainty. Further, I am not aware of anyone using EBNNs.*
>
> We agree with the reviewer: using ensembles of Bayesian Neural Networks (EBNNs) to disentangle and quantify different types of uncertainties is suboptimal. That is exactly what our empirical section aims at showing. We point out once again, though, that a single distribution – for example the predictive distribution associated with a single BNN – cannot gauge EU. On the other hand, a collection of distributions can; EU will capture the idea of “disagreement” between said distributions. Our empirical section shows that distilling the information that is present in such a collection in the first and second moments of a distribution – which is what EBNNs do – is less effective in quantifying and disentangling uncertainties than keeping the distributions separate in the form of a credal set. We focused on ensembling BNNs in this way because just averaging over the elements of the collection would return one element of the credal set, and working with the credal set as a whole is overall a better choice, as explained in Section 3.2. Indeed, it allows to hedge against distribution drift and misspecification. We will add a remark in the new version of the paper to make this clear.
> Regarding the choice of a sufficiently powerful model, it is very seldom the case that the EU goes to zero when a finite amount of data is collected. When that happens, it means that the uncertainty around the correct prior and/or likelihood to choose was very low to begin with: there is no intrinsic error in that. Typically, though, EU goes to zero only asymptotically, as a finite amount of data is almost never enough to overcome initial uncertainty. This was first pointed out by Walley in his 1991 book, but it was also recently re-discovered in Imprecise Machine Learning, see e.g. Figure 3 of https://arxiv.org/pdf/2209.03302, which we will cite in the new version of the manuscript.
> The reviewer might also mean that, while a single BNN cannot properly quantify EU, when part of an ensemble it might still capture some EU, and this would interfere with the quantification by the EBNN. This is a profound point: how do single components of an ensemble contribute to the quantification of the “global EU” faced by the agent? We currently do not have an answer to this point, and we will work on this issue in future research. As of now, though, we see no reason to claim that, when the uncertainty captured by an ensemble is distilled into only one distribution, there could be “uncertainty spills”, like when one pours water in a glass too hastily, and some finishes on the table instead of in the glass. We will certainly add this remark in the new version of the manuscript.
> Finally, we point out how ensembles of BNNs are not seldomly used, as a rapid “ensemble of bayesian neural networks” search in Google Scholar shows. As a more concrete example, many highly cited recent papers using EBNNs-like approaches can be found e.g. in the Related Works section of https://proceedings.neurips.cc/paper_files/paper/2023/file/07fbde96bee50f4e09303fd4f877c2f3-Paper-Conference.pdf
>
> *The paper states that the construction of IHDRs (Def 4) can be operationalized via a union of HDRs (Def 5). I thank the authors for clarifying my earlier misunderstanding about this. However, I fail to see how the second property is always fulfilled: does the union of HDRs really always provide a minimal cover of high density?*
>
> We thank the reviewer for the profound question. Yes, the union of HDRs always provides a minimal cover of high density, with respect to all the distributions in the credal set. In a more applied setting, this means that the probability of the union of the $\alpha$-HDRs containing the true output for the new input $\tilde{x}$ is at least $1-\alpha$, where $\alpha$ is chosen by the user, for all probability in the predictive credal set.

---

> > ### Author Response · Authors · 2024-07-13
> > **Thank you for your comments (cont'd)**
> >
> > *Add comparisons to single BNNs as a baseline using the mean, stddev output parameterization as the ensembling for EBNNs might not be necessary*
> >
> > As we argued before, using a single BNN the user is only able to capture aleatoric uncertainty in a principled manner. A single distribution cannot capture the uncertainty around whether the chosen distribution is the correct one. Hence, in the experimental environment of our paper, we would see an increase in aleatoric uncertainty as the corruption of the dataset increases (as we observe for CBDL and EBNN), and there could instead be no comparison regarding epistemic uncertainty. In turn, contrasting a single BNN with CBDL would not gain us any insight on which method to prefer, as CBDL is also able to capture epistemic uncertainty in a theoretically sound manner. We will make this clearer at the beginning of the Experiments section. If we were to use a single BNN to quantify EU, we would end up facing the same shortcomings highlighted in https://arxiv.org/abs/2407.01985. If instead the reviewer refers to the BNN obtained simply by averaging over the extreme elements of the credal set, the reason why it would be unfair to compare it with CBDL is that the latter hedges against distribution shift and misspecification, while the former does not, as we point out in Section 3.2. We will make this clearer too in the new version of the paper.
> >
> > *Compare to BMA as the cited paper is specific to HMC-based inference, which you do not seem to use anyway.*
> >
> > While we understand why the reviewer is concerned about BMA as a possible baseline for CBDL, let us inspect the reasons why we do not consider it. We begin with the motivations we have in the current version of the manuscript. Meaningful applications of BMA require to implement full batch Hamiltonian Monte Carlo in order to get to the true posterior (Izmailov et al., 2021b). This is computationally prohibitive in many cases, and it also suffers from shortcomings, as shown by Izmailov et al. (2021a).
> > Furthermore, BMA can be seen as an instance of a model featuring second-order distributions, i.e. distributions over distributions. In particular, the distribution $Q$ in equation (1) is a second order distribution. These types of models have been recently shown to suffer from major pitfalls when used to quantify EU due to their sensitivity to regularization parameters, and to underestimate AU (https://arxiv.org/pdf/2306.11113, https://arxiv.org/pdf/2402.09056, https://arxiv.org/abs/2203.06102). We will add this further motivation in the updated version of our manuscript.
> >
> > *Typos etc.*
> >
> > We thank the reviewer for spotting them, and we will mend them in the new version.

---

> > > ### Comment · Reviewer_tTAz · 2024-07-21
> > > **Please run the requested emotional**
> > >
> > > I will reply to the comments in full later. But to be clear, please run the experiment with single BNNs I've asked for. I will strongly reject otherwise. Thank you for your consideration.
> > >
> > > Best wishes

---

> > > > ### Author Response · Authors · 2024-07-23
> > > > **Thank you for your request**
> > > >
> > > > We thank the reviewer for their brutal honesty. We will include an experiment with a single BNN using the mean, standard deviation output parameterization. The posterior associated with such a single BNN induces a (single) posterior predictive distribution. The behavior of the AU measure associated with the latter – that is, its variance –  will mimic the behavior of the AU measure based on lower entropy of the CBDL. As pointed out in our previous answers, though, such a unique distribution cannot be used to capture EU. The theoretical motivation for this is the same as in our previous answer; a visual representation of this fact can be found in https://openreview.net/pdf?id=MhLnSoWp3p, Figure 1, left, and a rigorous analysis of why single BNNs suffer from shortcomings when used to gauge EU can be found in https://arxiv.org/abs/2407.01985.
> > > >
> > > > Regarding single BNNs, we should add an argument to our previous comment. There is a (rather un-Bayesian) way of interpreting the output of a BNN which will give rise to a second order distribution over the parameters of the neural network. Let us be more precise (we will add the following argument in the final version of our manuscript). Let $\Theta$ be the parameter space, $\mathcal{Y}$ be the space of outputs, and $\mathcal{X}$ be the space of inputs.
> > > >
> > > > Call $P \in \Delta_\Theta$ the prior distribution having pdf $p$, so that $\theta \sim p(\cdot)$, and $L \in \Delta_\mathcal{Y}$ be the likelihood having pdf $\ell$, so that $y \sim \ell(\cdot \mid x,\theta)$. Call $D=\left\lbrace{(x_i,y_i)}\right\rbrace_{i=1}^n$ the training set, assumed to issue i.i.d. from a distribution on $\Delta_{\mathcal{X}\times \mathcal{Y}}$. (As usual, we assume $\left\lbrace{x_i}\right\rbrace_{i=1}^n$ independent from $\theta$.) We can write the posterior pdf as $p(\theta \mid D)$, so that, after training, we know that $\theta \sim p(\cdot \mid D)$.
> > > >
> > > > Now, notice that different $\theta$'s give rise to different probabilistic predictions on $\mathcal{Y}$. To see this, pick $\theta_1,\theta_2\in\Theta$, $\theta_1 \neq \theta_2$, and notice that, in general, $\ell(y \mid x, \theta_1) \neq \ell(y \mid x, \theta_2)$. As a consequence, we can derive a **second order distribution** $\mathbb{L} \in \Delta(\Delta_\mathcal{Y})$ having pdf $\mathbf{l}$ as
> > > >
> > > > $$\mathbf{l} \left(\ell(y \mid x,\theta)\right) = \int_\Theta \mathbf{1} \left[
> > > > \ell(y \mid x, \tilde{\theta}) = \ell(y \mid x, \theta) \right] P(\text{d}\tilde{\theta} \mid D).$$
> > > >
> > > > So we can write $\ell(y \mid x,\theta) \sim \mathbf{l}(\cdot)$. As we can see, this expresses the ambiguity around the correct choice of the likelihood, and hence around the true data generating process.
> > > >
> > > > At this point, given a new input $\tilde x \in \mathcal{X}$, we can derive a predictive pdf $\ell_\text{induced}(\cdot \mid \tilde{x}, D)$ as
> > > >
> > > > $$\ell_\text{induced}(y \mid \tilde{x},D) = \int_{\Delta_\mathcal{Y}} \int_\Theta \ell(y \mid \tilde{x},\theta) \underbrace{\frac{p(\theta) \prod_{i=1}^n \ell (y_i \mid x_i,\theta)}{\int_\Theta p(\theta) \prod_{i=1}^n \ell (y_i \mid x_i,\theta) \text{ d}\theta}}_{\text{posterior computed according to likelihood } \ell} \text{d}\theta \text{ } \mathbb{L}(\text{d}\ell).$$
> > > >
> > > > Being a second-order approach, though, this has the shortcomings of second-order procedures that were highlighted in https://arxiv.org/pdf/2306.11113, https://arxiv.org/pdf/2402.09056, https://arxiv.org/abs/2203.06102. Namely, it suffers from major pitfalls when used to quantify EU due to its sensitivity to regularization parameters, and it tends to underestimate AU.

---

> > ### Comment · Reviewer_tTAz · 2024-07-24
> > **EBNNs in the Literature**
> >
> > Could the authors clarify the reference to the ENN paper in their reply:
> >
> > > Finally, we point out how ensembles of BNNs are not seldomly used, as a rapid “ensemble of bayesian neural networks” search in Google Scholar shows. As a more concrete example, many highly cited recent papers using EBNNs-like approaches can be found e.g. in the Related Works section of https://proceedings.neurips.cc/paper_files/paper/2023/file/07fbde96bee50f4e09303fd4f877c2f3-Paper-Conference.pdf
> >
> > The related work section therein mentions ensemble-based BNN approaches that train multiple particles:
> >
> > > Ensemble-based BNNs train multiple particles independently. This incurs computational
> > cost that scales with the number of particles. A thriving literature has emerged that seeks
> > the benefits of large ensembles at lower computational cost
> >
> > However, these particles are not BNNs themselves but regular DNNs that are then ensembled, so they are not EBNNs in the sense of this submission.
> >
> > Again, I am unaware of many papers that use ensembles of BNNs. I would much appreciate a clarification on this.

---

> > > ### Author Response · Authors · 2024-07-24
> > > **An Example**
> > >
> > > For example, this paper uses an approach similar to our EBNN one, where the ensemble distribution is a Normal whose mean is the average of the means, and the covariance matrix is an average of the covariance matrices (whereas we use a slightly more general value, i.e. the average of the covariances plus the variance of the averages)
> > >
> > > https://iopscience.iop.org/article/10.3847/1538-3881/ab2390/pdf

---

> ### Comment · Reviewer_tTAz · 2024-07-23
> **Please follow-up with experiment results**
>
> Thank you! Please let me know when the results with a single BNN and using the conventional approach of estimating epistemic uncertainty using the EIG/MI between predictions and parameters have been added. I'll consider the rest of your arguments then.
>
> Best wishes
>
> PS: The second part of your answer seems to contradict your first reply:
>
> > While in this framework EIG might be a somehow good measure for epistemic uncertainty, as we pointed out in our previous answers, second-order approaches suffer from shortcomings. We will not compare our model, whose main feature is being able to quantify and disentangle different types of uncertainties in a theoretically sound manner, to a way of proceeding (second order approaches) whose negative sides have been extensively studied, both theoretically and empirically. We will, though, add a paragraph in which we explain in detail what we presented so far in this answer session.
>
> I've asked for you to compare to a single BNN and compute AU and EU according to the well-established literature. This means either using a deep ensemble approximation and computing the EU as disagreement between the predictions (using the mutual information), or training an approximate BNN using VI or similar and computing the EU via the mutual information for parameter samples. Alternatively, one can also use the law of total variance decomposition (like in the Deep Ensemble paper that uses an ensemble of models with mean, stddev outputs) to decompose the total variance into aleatoric and epistemic variance.

---

> ### Author Response · Authors · 2024-07-24
> **Reply**
>
> We thank the reviewer for giving us the opportunity of being clearer about this. We will not compare against second order distribution approaches, given the shortcomings pointed out in our replies. In turn, we will not use EIG (again, a distribution over the models induces a second-order distribution).
>
> Instead, in the single BNN case, we will measure AU using the variance of the posterior distribution on the parameters of a single neural network, and EU using the approach in Section 2.1 in https://papers.nips.cc/paper_files/paper/2017/file/2650d6089a6d640c5e85b2b88265dc2b-Paper.pdf
>
> There, the authors consider $\theta_1$ , ... , $\theta_T$ samples from the (VI approximated) posterior on the parameters of the BNN. Then, given training set $D$ and a new input $\tilde{x}$, they compute
> $p_c=P(\tilde{y} =c \mid \tilde{x},D) \approx \frac{1}{T} \sum_{t=1}^T \text{Softmax}(\Phi_{\theta_t}(\tilde{x}))$, for all $c \in \lbrace{ 1,\ldots,C }\rbrace$.
>
> After that, they let $EU=H(p)=-\sum_{c=1}^C p_c \log p_c$.
>
> Would these measure satisfy the reviewer?

---

> > ### Comment · Reviewer_tTAz · 2024-07-24
> > **Wrong EU in Reply**
> >
> > To be clear: I've asked for the EU via the EIG or alternatively via the equivalent variance of means (in the regression case). So no: what you suggest is simply the entropy (total uncertainty) of the BMA prediction, which is **not** the epistemic uncertainty.
> >
> > In your submission, you write
> >
> > > **Results.** Following Egele et al. (2021), for EBNN we posit that $1 / R \sum\_{r=1}^R \sigma\_r^2$ captures the aleatoric uncertainty associated with $P\_{\text {ens }}$, and $1 /(R-1) \sum\_{r=1}^R\left(\mu\_r-\mu\_{\text {ens }}\right)^2$ captures the epistemic uncertainty associated with $P\_{\text {ens }} ;$ we use these values as baselines. ${ }^{16}$.
> >
> > First, Egele et al. (2021) do not use "EBNNs", so your citation is wrong. They use an ensemble of NNs that predict mean and stddev each and then do the well-established approach of estimating AU and EU using the ensemble outputs.
> >
> > Secondly, what Egele describes is precisely the baseline I have been requesting! Train an ensemble and then compute the AU and EU over these models, but the models themselves are not BNNs, and you do not average MC samples or similar before computing the metrics. The latter (which you do in your baseline) causes the EU to collapse, as described previously.
> >
> > So please correct the citation and perform the experiment. Please let me know if you can do that.
> >
> > Thanks

---

> > > ### Author Response · Authors · 2024-07-24
> > > **Thank you for the clarification**
> > >
> > > We thank the reviewer for their clarification. We will substitute Egele's reference with https://iopscience.iop.org/article/10.3847/1538-3881/ab2390/pdf
> > >
> > > Regarding the experiment request, if we understand correctly, the reviewer means the following. We train $R$ different neural networks, and obtain $R$ posteriors on the parameters of the neural networks. At that point, as a measure of EU, we use EIG computed as
> > > $\frac{1}{R} \sum_{r=1}^R KL(\text{posterior}_r || \text{prior}_r).$
> > > As a measure of AU, instead, we use the average of the posterior variances. Is this what the reviewer refers to?
> > >
> > > We also take this opportunity to apologize in case our previous answers came across as rude.

---

> > > > ### Comment · Reviewer_tTAz · 2024-07-24
> > > > **Re Experiment Request**
> > > >
> > > > Thank you!
> > > >
> > > > Yeah, except for each DNN, you will just have a MAP or MLE (depending on whether you use, e.g., weight decay or not). If you want, you can follow Egele exactly, which will mean training R DNNs using cross-entropy loss to get MLE estimates and then computing AU and EU accordingly, as already described in your **Results.** paragraph.
> > > >
> > > > Alternatively, if you want to be more Bayesian, you can use SGLD to train each network separately to obtain a single parameter sample each, or you can take a single of your trained MFVI BNNs (assuming they each output a mean and stddev) and compute the AU and EU by taking 10*20 MC samples and plugging them into your AU and EU estimators (without averaging).
> > > >
> > > > This is essentially what Egele et al were/are doing.

---

### Review · Reviewer_9eVm · 2024-07-09

**Summary Of Contributions:**

This paper proposes credal Bayesian deep learning. The main idea is to consider of a set of $K$ priors/ its convex hull set and another set of $S$ likelihood distribution/its convex hull set (e.g., different architectures), subsequently variational approach is applied to get $K \times S$ posteriors (each corresponds to one pair of prior and likelihood distribution), and finally $K \times S$ posteriors are used to make predictions and compute the Imprecise Highest Density Region over the predictions.

**Audience:**

Yes

**Broader Impact Concerns:**

There is no ethical concern of the work.

**Claims And Evidence:**

Yes

**Requested Changes:**

- Please give more motivation for leveraging the imprecise probabilities to Bayesian learning.

- What is the fundamental difference of considering a mixture of some priors and a convex hull over the set of priors and similar question for the likelihood distribution?

- Please make changes based on my comments in the weakness section.

**Strengths And Weaknesses:**

## Strengths
- The idea makes sense.
- The theory developed is quite solid.

## Weaknesses
- The story is not really motivative and convincing. The section to motivate why we need to leverage the imprecise probabilities in Bayesian learning is not well motivated and convinced to me. The authors should rewrite this section.
- It is still unclear the benefit of the imprecise probabilities, e.g., this enables the computation of  Imprecise Highest Density Region and Aleatoric and Epistemic Uncertainties? Moreover, the definition of $\bar{H}(P)$ as sup of $H(P)$ is not rigorous because the sup is over $P$. Instead, it should be $H(\Pi')$?
- Is the final Imprecise Highest Density Region over the label set is similar to the conformal prediction? If so, the authors should compare with the conformal prediction approaches for regression and classification to demonstrate the merit.
- The proposed approach seems to be time and memory consuming because we need to do variational approach $K \times S$ times and store $K \times S$ deep learning models?
- The authors should enrich the experiments to compare with the current SOTA approaches in Bayesian Neural Networks. As far as I know, there are recently many works in Bayesian Neural Networks.
- In Tables 1 and 2, the authors should report the expected calibration error.

---

> ### Author Response · Authors · 2024-07-13
> **Thank you for your comments**
>
> We thank the reviewer for their insightful comments. Let us address them in depth.
>
> *The story is not really motivative and convincing. The section to motivate why we need to leverage the imprecise probabilities in Bayesian learning is not well motivated and convincing to me. The authors should rewrite this section.*
>
> We thank the reviewer for this remark. In the updated version, we will make it clearer why imprecise probabilities are crucial in uncertainty quantification. To sum it up, imprecise probabilistic techniques (which credal sets are an example of) and ensemble of distributions (such as ensemble of Bayesian neural networks) are the only ways to gauge epistemic uncertainty in a theoretically sound fashion. A credal set approach works better than an ensemble one both at disentangling the different types of uncertainty, and in downstream tasks. For a longer answer, we refer the reviewer to the first answer we gave to Reviewer tTAz.
>
> *It is still unclear the benefit of the imprecise probabilities, e.g., this enables the computation of Imprecise Highest Density Region and Aleatoric and Epistemic Uncertainties? Moreover, the definition of $\overline{H}(P)$ as sup of $H(P)$ is not rigorous because the sup is over $P$.*
>
> We thank the reviewer for this remark, and we will strive to make it clearer in the updated version of our paper. As the reviewer correctly points out, the use of credal sets enables us to compute IHDRs and to quantify and disentangle total uncertainty into its aleatoric and epistemic components in a theoretically sound fashion. The experimental section also shows that our credal approach improves on an ensemble one.
> The definition of upper entropy is not ours. It was given first in Abellan et al (2006), and successively in Hüllermeier and Wageman (2021). It is defined as the maximum entropy that can be achieved by the elements of the credal set $\Pi^\prime$. This can be expressed mathematically as the sup out of all $P \in \Pi^\prime$ of the entropy $H(P)$. The reviewer might be confused because they think of the entropy of the upper probability. That concept needed separate treatment, and was confined to Appendix E. The entropy of the upper probability, though, plays no role in the main story of our paper in quantifying the total uncertainty.
>
> *Is the final Imprecise Highest Density Region over the label set similar to the conformal prediction? If so, the authors should compare with the conformal prediction approaches for regression and classification to demonstrate the merit.*
>
> We thank the reviewer for this comment. The IHDR is indeed similar in spirit to a conformal prediction region, but the way it is derived is entirely different. We pointed out in our third answer to the requested changes by Reviewer CP1D the reason why we do not compare to conformal prediction, but let us report our answer here for completeness.
> We only compare against Bayesian techniques because a comparison against non-Bayesian ones is either difficult to justify or unfair. Indeed, the initial knowledge component, captured by the prior distribution (or the set of priors, in our case), has no direct comparisons with other approaches. Take conformal prediction for instance. It is a model-free method, which means that no prior knowledge around the experiment at hand is required. The only choice the user makes is which non-conformity score to use. How would we compare this approach with the choice of different priors, which conveys the idea that the user knows that a true distribution over the space of neural network parameters exists, but it is not fully known? Even though the diameter of the conformal prediction region captures uncertainty, it is unable to distinguish between aleatoric and epistemic. Indeed, the diameter is a positive function of both: it increases as both increase, and hence it cannot be used to distinguish the two. We will add this argument in a remark in the introduction to further clarify why we only focus on Bayesian baselines.
> Finally, it is easy to understand why conformal prediction regions are usually larger than IHDRs. The reason is while the former need to guarantee that the correct output $\tilde{y}$ for the new input $\tilde{x}$ belongs to the region w.p. $\geq 1-\alpha$ **for all** distributions on the output space $\mathcal{Y}$, for IHDRs the probabilistic guarantee only needs to hold for all distributions in the predictive credal set $\mathcal{P}_\text{pred}$. This is because CBDL is a model-based approach, and so the modeling effort is repaid by only needing to worry about the distributions that the user deems plausible, and that hence form the prior and likelihood credal sets (and which in turn allow to derive the predictive credal set).

---

> ### Author Response · Authors · 2024-07-13
> **Thank you for you comments (cont'd)**
>
> *The proposed approach seems to be time and memory consuming because we need to do $K \times S$ variational approach times and store $K \times S$ deep learning models?*
>
> We thank the reviewer for pointing this out. While they are certainly right, our approach allows us (i) to forego any additional assumptions on the nature of the lower and upper probabilities that are oftentimes required by other imprecise-probabilities based techniques (as pointed out after discussing Step 3 of Algorithm 1), and (ii) to derive a predictive credal set, which in turn allows us to quantify and disentangle epistemic and aleatoric uncertainty in a theoretical sound fashion. In addition, as we mentioned when answering to Reviewer CP1D, an evidential approach seems to be a promising route to overcome this computational bottleneck, as it allows us to derive a credal set without resorting to approximating multiple posteriors. We will mention this in the updated version of our manuscript.
>
> *The authors should enrich the experiments to compare with the current SOTA approaches in Bayesian Neural Networks. As far as I know, there are recently many works in Bayesian Neural Networks.*
>
> We thank the reviewer for this suggestion, but we argue similarly to what we did for a question asked by Reviewer tTAz why comparing to a single BNN would be pointless. Using a single BNN, the user is only able to capture aleatoric uncertainty in a principled manner. A single distribution cannot capture the uncertainty around whether the chosen distribution is the correct one. Hence, in the experimental environment of our paper, we would see an increase in aleatoric uncertainty as the corruption of the dataset increases (as we observe for CBDL and EBNN), and there could instead be no comparison regarding epistemic uncertainty. In turn, contrasting a single BNN with CBDL would not gain us any insight on which method to prefer, as CBDL is also able to capture epistemic uncertainty in a theoretically sound manner. We will make this clearer at the beginning of the Experiments section. If we were to use a single BNN to quantify EU, we would end up facing the same shortcomings highlighted in https://arxiv.org/abs/2407.01985.
>
> *In Tables 1 and 2, the authors should report the expected calibration error.*
>
> We thank the reviewer for this suggestion, which we will implement in the new version of the manuscript.
>
> *What is the fundamental difference of considering a mixture of some priors and a convex hull over the set of priors, and similar question for the likelihood distribution?*
>
> We thank the reviewer for these profound questions. A simple way of answering them is the following. Consider three distributions. The credal set they induce is their convex hull, that is, the collection of all their possible (convex) mixtures. Hence, considering only one mixture corresponds to considering only one element of the credal set. We tried to make this clear in our manuscript, especially in Figure 2, where the solid black curve represents one mixture of the two Normals (the blue and brown dashed curves), and hence one element of the credal set that they induce. We will strive to make it clearer in the new version of the paper. Maybe, a mathematical representation will help. Consider again three distributions, $P_1$, $P_2$, $P_3$. Then, the credal set they induce is $\lbrace{Q: Q= \alpha P_1 + \beta P_2 + (1-\alpha-\beta) P_3 \rbrace}$, where $\alpha,\beta \in [0,1]$, and $\alpha + \beta \leq 1$. On the other hand, a mixture of these three distributions consists of selecting particular values $\alpha^\star$ and $\beta^\star$, and only consider the distribution $Q^\star = \alpha^\star P_1 + \beta^\star P_2 + (1-\alpha^\star-\beta^\star) P_3$.

---

### Author Response · Authors · 2024-08-05
**Extension**

Dear Editors in Chief, Action Editor, and Reviewers,

We thank you all for the fruitful discussion and comments on our Credal Bayesian Deep Learning paper.

We are writing to you to kindly ask you an extension of 3 days to submit your final decision, so to have time until August 9. We are in the process of finishing the experiment requested by Reviewer tTAz, and we want to report it before you decide for acceptance or rejection. The delay was due to sickness of one of the authors (who was hit by the recent Covid wave).

We hope you’ll accept our sincerest apologies,

Your,

Authors

---

> ### Author Response · Authors · 2024-09-02
> **Additional comments (Reviewer tTAz)**
>
> We thank the reviewer once again for their comments. We will address them now one by one.
>
> 1. We thank the reviewer for pointing this out. We made a mistake in the computation of the AUC, which is amended in the new version https://www.dropbox.com/scl/fi/rqne5xl80iwz4nt7mkfk3/CBDL_TMLR_Additional_Experiments_v2.pdf?rlkey=lgmqtmtdv92qnwhhzg1litddx&st=331gfmi7&dl=0 As the reviewer can see, we consistently improve on a single BNN, and we also improve on an ensemble of BNN except for two datasets, where our performance is essentially the same as the ensemble.
>
> 2. We thank the reviewer for their question. We will provide these results tomorrow evening (EST); given the ones we have already in the new version, though, we are positive that the OOD detection performance will be better. We will still confirm this with our empirical results and report the AUROC scores.
>
> 3. We thank the reviewer for pointing this out. In the new version, we pointed out more clearly in the caption to the tables that four refers to the number of models we trained for the CBDL and Baseline (Ensemble) method. For the single BNN, we used the model that demonstrated the highest accuracy.

---

> ### Author Response · Authors · 2024-09-02
> **Additional Comments (Reviewer CP1D)**
>
> We thank the reviewer for pointing this out. The requested summaries have been added to the new version of our extended experimental results https://www.dropbox.com/scl/fi/rqne5xl80iwz4nt7mkfk3/CBDL_TMLR_Additional_Experiments_v2.pdf?rlkey=lgmqtmtdv92qnwhhzg1litddx&st=331gfmi7&dl=0

---

> ### Author Response · Authors · 2024-09-04
> **Additional comments (Reviewer tTAz, cont'd)**
>
> We are now able to give results pertaining AUROCs for the OOD detection performance. Table 3 here https://shorturl.at/uiboB shows the AUROC results for OOD Detection performance. Each group of results shows which dataset the models were trained on, and which dataset the model was tested on. We report AUROC when using both the epistemic uncertainty and aleatoric uncertainty of each approach. In general, the baseline approach performed worst in all cases. When using aleatoric uncertainty, a single BNN performed relatively better than CBDL for OOD detection, though CBDL was still comparable and outperformed when trained on Fashion MNIST. When using epistemic uncertainty, CBDL clearly outperformed the other approaches on most datasets. This is in line with other works where epistemic uncertainty was considered important for OOD detection (Kendall and Gal, 2017).
>
> We conjecture that this may be due to the fact that a single BNN is not able to gauge EU properly. Hence, the single BNN flags an instance as OOD when it comes from the “tail” of the distribution; this is well-captured by aleatoric uncertainty. On the contrary, CBDL is able to gauge EU properly, and hence it is able to capture when OOD happens by looking at the “disagreement” of the elements of the predictive credal set, captured by the difference between upper and lower entropy. Further verification of the results will be made before the final version of the paper, but we do not expect major differences in the observations or conclusions.

---

> > ### Comment · Reviewer_CP1D · 2024-09-05
> > **Further verification still coming?**
> >
> > Thanks for the updates again, to me these seem good so far. Are you still planning to add something related to the experiments reported in Table 3 (as a detail, please at least add some description of the results to Table 3 similar to the previous tables)?

---

> > > ### Author Response · Authors · 2024-09-05
> > > **Answer to Reviewer CP1D on Further verification**
> > >
> > > We thank the reviewer for their answer. In the final version of the manuscript, we will add the discussion that we carried out in the answer to Reviewer tTAz. In the meantime, we have added a caption to Table 3, in a similar fashion to what we did for all the other Tables in the document containing our additional experiments https://www.dropbox.com/scl/fi/rqne5xl80iwz4nt7mkfk3/CBDL_TMLR_Additional_Experiments_v2.pdf?rlkey=lgmqtmtdv92qnwhhzg1litddx&st=se8kkyfh&dl=0

---

> > > > ### Comment · Reviewer_tTAz · 2024-09-11
> > > > **Thank you!**
> > > >
> > > > Thanks for the additional experiments and updates. I'm very satisfied with the additional results.
> > > >
> > > > I would strongly suggest editing the paper for the camera-ready to make the "regular" BNNs the **baseline** as they are the most commonly used setup, and make the EBNNs an ablation that is a naive(?) extension to multiple BNNs to contrast them to your CBDL, which works much better. I say this because, again, EBNNs are barely used in practice. I haven't been able to find additional references/usages, and I have looked quite a bit now. This framing helps you guide readers better, and the positioning explains away that EBNNs perform worse than BNNs in the added experiments.
> > > >
> > > > The research question for this framing would be: how does naively ensembling BNNs to capture EU (which BNNs don't have according to your philosophical framework (which I still heavily disagree with)) fare compared to credal BNNs? The results then show that this is not trivial, and the credal approach makes a huge difference, providing further empirical evidence for its setup.
> > > >
> > > > Thanks for putting in all the extra work and effort to make this a stronger and empirically more convincing paper!

---

> > > > > ### Author Response · Authors · 2024-09-12
> > > > > **Thank you!**
> > > > >
> > > > > Dear Reviewer tTAz,
> > > > >
> > > > > Thank you so much for your kind words, and for your suggestion, which we wholeheartedly agree on. We will certainly implement it in the camera ready version, if the paper gets accepted.
> > > > >
> > > > > Yours,
> > > > >
> > > > > The Authors

---

### Public Comment · ~Calvin_McCarter1 · 2024-12-08
**Code availability?**

This method looks very interesting! The Github repository is currently empty -- will code for implementation and experiments be added in the near future? Thanks!

---

### Decision · Action_Editor_BjYT · 2024-09-12

**Recommendation:** Accept with minor revision

**Comment:**

Given the additional experiments provided throughout a detailed and fruitful discussion with the reviewers the paper has matured enough for me to recommend acceptance.

The restriction of a minor revision is to ensure that the additional results are properly included into the final version of the paper.

**Audience:**

Bayesian deep learning as well as uncertainty quantification in general are two popular and relevant areas of research that overlap with the research questions of a significant number of TMLR's audience.

**Claims And Evidence:**

After long detailed discussions with the reviewers, the additional experiments provided by the authors greatly strengthen the claims made in the submission.